# Native-state proteomics of Parvalbumin interneurons identifies unique molecular signatures and vulnerabilities to early Alzheimer's pathology

Prateek Kumar [1,2,3,9], Annie M. Goettemoeller[2,4,9], Claudia Espinosa-Garcia[1,3], Brendan R. Tobin [5], Ali Tfaily [3], Ruth S. Nelson[3], Aditya Natu[1], Eric B. Dammer [2,6], Juliet V. Santiago[1,2,4], Sneha Malepati[2,7], Lihong Cheng[1,2], Hailian Xiao[1,2], Duc D. Duong[2,6], Nicholas T. Seyfried [1,2,6], Levi B. Wood[5,8], Matthew J. M. Rowan [2,7,10] ✉ & Srikant Rangaraju [1,2,3,10] ✉

Dysfunction in fast-spiking parvalbumin interneurons (PV-INs) may represent an early pathophysiological perturbation in Alzheimer's Disease (AD). Defining early proteomic alterations in PV-INs can provide key biological and translationally-relevant insights. We used cell-type-specific in-vivo biotinylation of proteins (CIBOP) coupled with mass spectrometry to obtain native-state PV-IN proteomes. PV-IN proteomic signatures include high metabolic and translational activity, with over-representation of AD-risk and cognitive resilience-related proteins. In bulk proteomes, PV-IN proteins were associated with cognitive decline in humans, and with progressive neuropathology in humans and the 5xFAD mouse model of Aβ pathology. PV-IN CIBOP in early stages of Aβ pathology revealed signatures of increased mitochondria and metabolism, synaptic and cytoskeletal disruption and decreased mTOR signaling, not apparent in whole-brain proteomes. Furthermore, we demonstrated pre-synaptic defects in PV-to-excitatory neurotransmission, validating our proteomic findings. Overall, in this study we present native-state proteomes of PV-INs, revealing molecular insights into their unique roles in cognitive resiliency and AD pathogenesis.

A major goal in cellular neuroscience is to elucidate how the molecular signatures of unique neuronal subtypes translate to their functional diversity in intact circuits. Single-neuron transcriptomic studies have recently provided unparalleled access to the genetic diversity of dozens of unique brain cell classes[1,2]. Functional information is nonetheless limited in transcriptomic studies, due to substantial discordance between mRNA and protein levels, especially in neurons[3–5]. Proteomic studies relying on physical isolation of individual neuron types are also inadequate, as physical isolation of

individual neurons is poorly tolerated, and of those that do survive, the vast majority of their functional surface area (i.e., dendrites and axons) is lost[6,7]. To overcome these limitations, we recently developed an in vivo strategy called cell type-specific in vivo biotinylation of proteins (CIBOP). When coupled with mass spectrometry, CIBOP can resolve native state proteomes from physically unaltered cell subtypes in vivo[8]. Key technical advancements, especially relating to neuronal subtype-specific targeting across different disease models, are also necessary to fully realize the potential of this method via extension to

distinct classes of excitatory and inhibitory neurons. The recent discovery of highly versatile enhancer-AAVs[9] have the potential to fulfill these requirements, with tools targeting inhibitory interneurons receiving major initial development[10,11].

While inhibitory interneurons account for ~20% of neurons in the brain[12], these cells contact essentially every excitatory neuron in due to their extensive local axonal arborizations. Alterations in inhibitory interneuron function appear responsible for circuit and behavioral dysfunction in several neurological diseases. In particular, dysfunction of fast spiking, parvalbumin-expressing interneurons (PV-INs) are implicated in epilepsy, neurodevelopmental, and neurodegenerative diseases including Alzheimer's disease (AD)[13,14], a likely consequence of their role in maintaining circuit excitability locally, and brain state more generally, coupled with their substantial energy requirements[15]. Together, this cell class represents a promising locus for designer treatments across several major neurological disorders. Therapeutic failures are common in brain diseases, potentially due to unpredictable competing cell-type-specific responses. Thus, to enhance future therapeutic efficacy, high-resolution native state proteomic signatures of individual cell classes in wild type and disease models are required. Therefore, we implemented a versatile, systemic AAV-CIBOP intersectional approach[8,10,16] to characterize and compare native state in vivo PV-IN proteomes from both wild type mice and in a mouse model of early AD pathology. A recently-developed enhancer-AAV targeting method was used to express Cre recombinase specifically in PV neurons throughout the cortex and hippocampus of Rosa26^TurboID mice[8]. Upon Cre-mediated recombination, TurboID was expressed selectively in PV-INs, leading to robust cellular proteomic biotinylation. This PV-IN CIBOP approach identified over 600 proteins enriched in PV-INs, including canonical proteins as well as over 200 additional PV-IN proteins. The PV-IN proteome was enriched in mitochondrial, metabolic, ribosomal, synaptic, and many neurodegenerative genetic risk and cognitive resilience-related proteins, suggesting unique vulnerabilities of PV-INs in AD.

AD is arguably the most impactful and intractable neurodegenerative disease. Hyperexcitability is evident in distinct mouse models of familial and sporadic AD[17-20] including in prodromal disease stages[21,22]. Furthermore, abnormal brain activity is apparent in humans with mild cognitive impairment[23-25] and in early familial AD[26,27]. Increases in circuit activity also appear to directly drive circuit pathology, by accelerating toxic protein deposition[28-32]. Interestingly, selective alterations in PV-IN physiology are increasingly appreciated across several distinct in vivo models of AD pathology[19,33,34] which appear to causally contribute to prolonged cognitive dysfunction arising during the early stages of the disease[19,35-37]. Using network analyses of human post-mortem brain proteomes from controls and AD cases, we first identified a PV protein-enriched co-expression module (M33) strongly associated with cognitive resilience in longitudinal aging studies. Based on these lines of evidence, we next extended PV-IN CIBOP to a mouse model of hAPP/Aβ pathology[38,39]. We found that PV-INs in pre-plaque (3 month old) 5xFAD mice exhibited extensive alterations in their mitochondrial and metabolic, cytoskeletal, and synaptic proteins, coinciding with decreased Akt/mTOR signaling. Several of these changes were validated using optogenetics, patch clamp, and cell-type-specific biochemistry. Strikingly, many of the proteomic changes noted in PV-INs in response to early Aβ pathology, were not resolved in the bulk brain proteome, suggesting that these cell-type-specific alterations are largely non-overlapping in early AD.

Overall, our studies using the CIBOP approach reveal native-state proteomic signatures and identify potential molecular vulnerabilities of PV-INs to neurodegeneration in AD, and nominate potentially high-value targets otherwise hidden in the bulk proteome. This enhancer AAV-CIBOP strategy will also be broadly applicable to understanding molecular complexity of PV-INs and other neuronal subtypes, across any mouse model of health or disease.

## Results

### Native-state proteomics of PV interneurons in wild-type mouse brain

Proteomic biotinylation of native-state PV-INs (PV-CIBOP) was achieved by retro-orbital (RO) delivery of PV-IN-specific enhancer-targeting AAV (PHP.eB-E2-Cre-2A-GFP)[33,40,41] into Rosa26^TurboID/wt (PV-CIBOP group) or wild-type (WT) mice (Fig. 1A)[16]. For acute slice electrophysiology of PV-INs, we co-injected (RO) an AAV construct containing a floxed TdTomato sequence to fluorescently-label PV-INs due to non-visualization of GFP in ex vivo slices. After 3 weeks of Cre-recombination and 2 weeks of biotin supplementation[8], we performed acute slice current clamp recordings confirming selective targeting and unaltered physiology of fast-spiking PV-INs by PV-CIBOP (Fig. 1A, Supplementary Fig. 1). To assess potential impacts of PV-specific TurboID expression and proteomic biotinylation on PV neuron function and overall local circuit activity, we obtained voltage and current clamp recordings from layer 5 pyramidal neurons and PV interneurons, respectively, in both WT control and PV-CIBOP mice (Supplementary Fig. 1A, B, H). Fast, spontaneous excitatory and inhibitory synaptic events (EPSCs and sIPSCs) were isolated during pyramidal cell recordings (Supplementary Fig. 1B). The amplitude, frequency and kinetic properties of both sEPSCs and sIPSCs were unperturbed in PV-CIBOP brains (Supplementary Fig. 1C–G). These results indicate that PV-CIBOP does not affect basal circuit excitability, synaptic receptor distributions or synaptic properties. In the same slices, neighboring TdTomato+ neurons exhibited fast, non-accommodating firing with narrow action potentials (Supplementary Fig. 1H–J) and passive properties characteristic of fast-spiking PV-INs (Supplementary Fig. 1K–M). No differences in AP firing, various biophysical features, or passive properties were observed in PV-INs comparing PV-CIBOP with WT controls (Supplementary Fig. 1I–M). Immunohistochemical (IHC) studies from fixed PV-CIBOP cortices showed widespread biotinylation of PV-positive (PV + ) and GFP-positive neurons (Fig. 1B, C), in somatic and axo/dendritic compartments of PV-INs (Fig. 1D) without off-target biotinylation or reactive gliosis (Supplementary Fig. 2). Western blots (WB) showed strong biotinylation of a wide array of proteins in PV-CIBOP mice compared to few endogenously biotinylated proteins in WT control lysates (Fig. 1E)[8]. Using the N-terminal V5 (V5-TurboID) as a surrogate of Cre-mediated TurboID expression, we detected V5 in PV-CIBOP mice but not WT controls (Fig. 1E).

Biotinylated proteins from PV-CIBOP and WT control samples were enriched using streptavidin (SA) beads followed by silver stain and WB (Fig. 1E), confirming enrichment of biotinylated proteins from PV-CIBOP mice, mirroring patterns observed in bulk brain lysates (inputs). Label-free quantitative MS (LFQ-MS) of SA-enriched samples identified a PV-IN proteome of 628 proteins enriched in PV-CIBOP samples as compared to controls ( ≥ 2-fold enriched and unadjusted p ≤ 0.05; 192 proteins ≥2-fold enriched at the FDR ≤ 0.05 threshold) (Fig. 1F, Supplementary Data 1). These included canonical PV-IN proteins, including Kv3 channels (Kcnc1, Kcnc2, Kcnc3; which were expressed at similar levels relative to each other), Gria4, Syt2 and Ank1, while markers of excitatory neurons (Slc17a7), astrocytes (GFAP), microglia (Cst3), oligodendrocytes (Mbp, Plp1) were not enriched (Fig. 1G)[33,42–44]. Gene set enrichment (GSEA) (Fig. 1H) and SynGO (Fig. 1I)[45] analyses of the PV-IN proteome showed over-representation of gene ontologies (GO) including synaptodendritic and axonal localization, neurotransmission, vesicle function, synapse organization, ARF-GTPase signaling, growth factor receptor signaling, tauopathy/synucleinopathy, and included pre- and post-synaptic compartments. Some of the most abundant synaptic PV-IN proteins were involved in synaptic vesicle trafficking, fusion and exocytosis and included complexins (Cplx 1-3), synucleins alpha and beta (Snca, Sncb), Bin1 and amphiphysin (Fig. 1J)[46]. In contrast with bulk brain proteomes

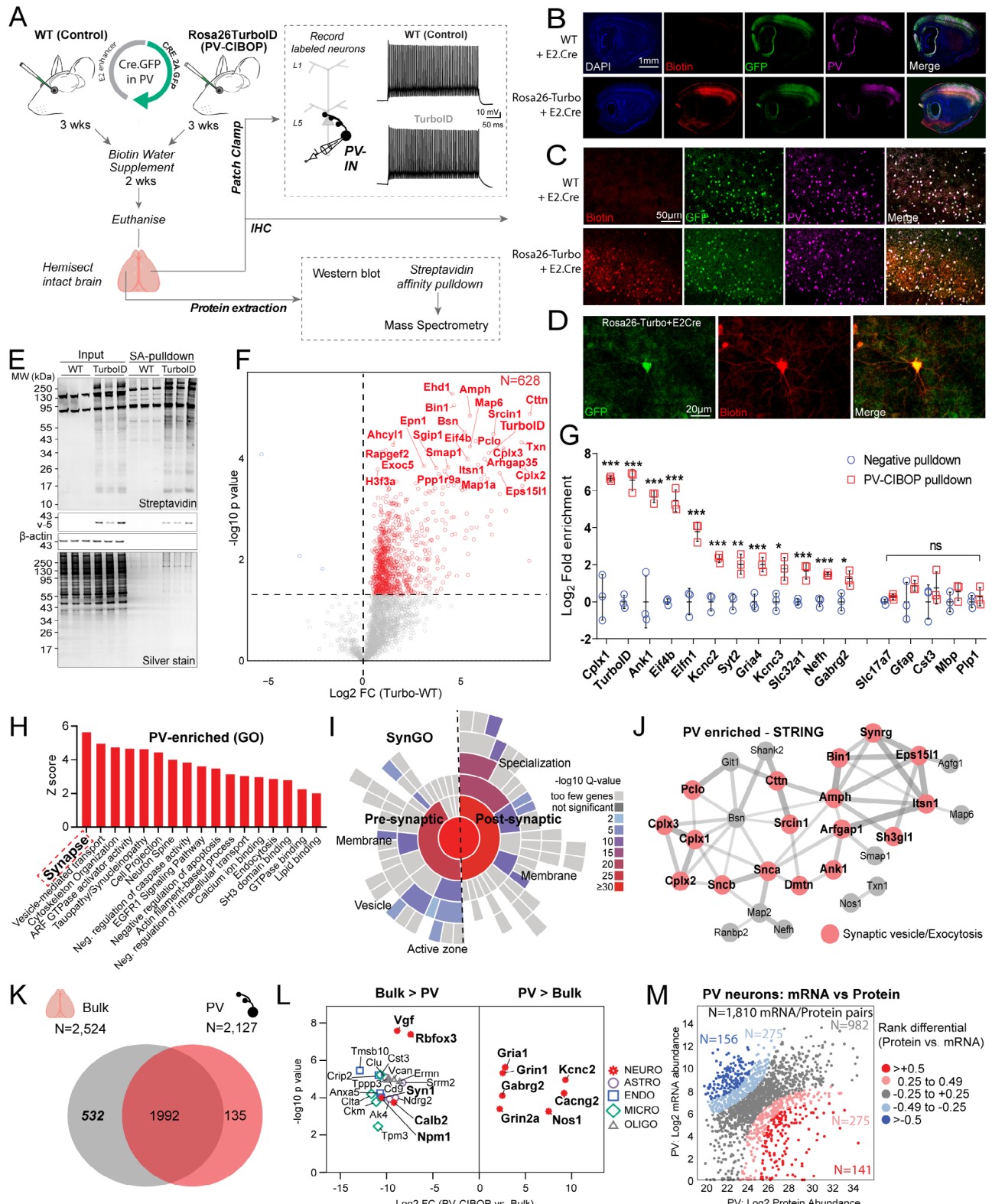

(Fig. 1K, L), PV-IN-specific proteomes including GABAergic and Kv3 channels, proteins involved in synaptic vesicle function, endocytic/endosomal pathways, GTPase binding proteins, cytoskeletal proteins (Ank1)[43,47] and ribosomal large subunit proteins (Rpl15, Rpl18, Rpl24). Non-PV-IN proteins and glial proteins were preferentially enriched in the bulk brain proteome (Fig. 1L). Using a reference dataset of brain protein half-lives generated by metabolic labeling[48], we found that the PV-CIBOP proteome had a similar distribution of short and long half-

life proteins (Supplementary Fig. 3). We also verified that the distribution of neuronal soma and axon/synaptic/dendrite-localized proteins in the PV-CIBOP proteome and bulk brain proteome, were similar (Supplementary Fig. 4). These analyses suggest that the PV-CIBOP proteome is indeed representative of the neuronal proteome, rather than being biased by protein turn-over or biased towards specific neuronal compartments. Lastly, we contrasted our PV-IN proteome with reference single cell/nuclear RNAseq PV-IN transcriptomes

**Fig. 1 | Native-state proteomics of PV-INs by CIBOP. A** Experimental outline to achieve native-state proteomics of PV-INs by CIBOP. **B–D** Immunohistochemistry of fixed brain sections confirmed biotinylation (red) of PV-INs (Pvalb: green) in PV-CIBOP but not in control mice (B: 4x and C: 20x magnification; D: Higher magnification (60x) images show biotinylation in PV-INs). **E** Top: Western Blot (WB) of input and streptavidin affinity pulldown samples confirms strong protein biotinylation in PV-CIBOP as compared to limited biotinylation in control animals. Bottom: Silver-stained gels of inputs and pulldown samples corresponding to WB images above. **F** Volcano plot representation of differential abundance of biotinylated protein, from PV-CIBOP and control mice. Red dots represent proteins biotinylated in PV-INs as compared to control mice (unpaired two-tailed *T*-test $p \leq 0.05$). **G** Top PV-enriched proteins are shown on the left (including TurboID, Cnk1, Kcnc2, Kcnc3, Erbb4, Slc32a1 and GABA-ergic proteins). In contrast, non-neuronal (Mbp, Gfap, Aldh1l1, Cotl1) and excitatory neuronal (Slc17a7) proteins were not enriched (Data are presented as mean + SD, *n* = 3/group; unpaired two-tailed *T*-test \**p* < 0.05, \*\**p* < 0.01, \*\*\**p* < 0.005). **H** Gene Ontology (GO) analyses of PV-enriched proteins compared to whole brain show enrichment of synaptic vesicle, GTPase binding, cytoskeletal and cell-projection related proteins. **I** SynGO analysis of PV-enriched proteins reveals labeling of proteins in pre- and post-synaptic compartments. **J** STRING analysis of PV-enriched synaptic proteins (>4-fold enriched over control) involved in synaptic vesicle and exocytosis, including complexins, ankyrins, synucleins. **K** Venn Diagram representing degree of overlap between proteins enriched in PV neurons, with whole brain proteomes from matched animals. **L** Top proteins differentially enriched in PV-INs as compared to the whole brain bulk proteome and those enriched in the bulk as compared to PV-INs, are highlighted. **M** Analysis of protein vs mRNA concordance in PV-INs, using PV-enriched proteins identified by PV-CIBOP and existing single nuclear transcriptomic data from mouse PV-INs. Based on differentials in rank abundances (protein vs. mRNA), discordant and concordant protein/mRNA pairs are highlighted. Also see Supplementary Figs. S1, S2, S3 and Supplementary Data 1 for related analyses and datasets. Source data are provided as a Source Data file. Image was created using BioRender.com.

from mouse brain (Fig. 1M, Supplementary Fig. 5)[49], revealing modest concordance between 1,810 mRNA-protein pairs (Spearman's Rho=0.27, *p* < 0.001) (Supplementary Data 1).

In summary, our PV-CIBOP studies successfully identified native-state proteomic signatures of PV-INs many of which are not captured by bulk brain proteomics and are discordant with mRNA-level findings.

## Proteomic signatures of PV-INs in contrast to Camk2a excitatory neurons reveal molecular signatures associated with vulnerability and cognitive resilience

We contrasted proteomic signatures of PV-INs with Camk2a-positive excitatory neurons using the CIBOP approach in two independent cellular contexts (Fig. 2A)[8]. Of 1,841 proteins enriched in either PV-CIBOP or Camk2a-CIBOP proteomes (Fig. 2A, Supplementary Data 2), 1,578 were enriched in Camk2a neurons and 1,408 proteins enriched in PV-INs, with 1,135 proteins enriched in both. 245 proteins were highly-enriched ( > 4-fold) in PV-INs (including Kv3 channel proteins) while 163 proteins were highly-enriched in Camk2a neurons (Fig. 2B). Ribosomal, GABA metabolism, ephrin B pathway, clathrin-coated vesicle, transport, cytoskeleton, endoplasmic reticulum, calcium binding, synaptic vesicle exocytosis and Akt/mTOR signaling were over-represented in PV-IN-enriched proteins (Fig. 2C, Supplementary Fig. 6). In contrast, cellular metabolism, fatty acid oxidation, NAD binding, lipid metabolism, proteasome complex, ER-phagosome and mitochondrial terms were over-represented in the Camk2a-CIBOP-enriched proteome (Fig. 2C, Supplementary Fig. 6). Upstream analyses identified potential microRNA (miRNA) regulators of PV-INs and Camk2a neurons, including enrichment of microRNAs 133a and 133b targets in the PV-IN proteome (Supplementary Fig. 6), in agreement with prior miRNA tagging and affinity purification (miRAP)[50] studies that identified miRNAs 133a and 133b as highly expressed in PV-INs. Two microRNAs recently identified as predictors of cognitive decline in humans (miR-29a and miR-132), were also predicted to regulate PV-IN proteomic signatures[51]. These analyses suggest that molecular signatures that define PV-INs may be regulated by distinct sets of miRNAs, some of which have known associations with cognitive decline in humans.

To identify neurodegeneration-relevant proteins in PV-INs, we cross-referenced PV-IN and Camk2a CIBOP proteomic markers with neurodegeneration-associated risk genes from Multi-marker Analysis of GenoMic Annotation (MAGMA) analyses (Fig. 2D, Supplementary Data 2)[52,53]. We identified 60 PV-IN AD-risk proteins related to synaptic vesicle fusion, docking and recycling (Bin1, Picalm, Dnm2, Ap1g1, Ap2a2, Sgip1), cytoskeleton and microtubules (Ank1, Actb, Tubb2a, Mapt), mitochondria (Mtch2, Ndufs3, Ndufb9, Slc25a11), and SNARE complex (Syn2, Stx1b, Vamp1, Nsf, Stxb1) (Fig. 2E). In comparison, 24 Camk2a neuron-enriched AD-risk proteins were identified, including

oxidoreductases (Sdhb, Idh2, Aldh5a1, Etfb and Acadl), serine/threonine kinase Akt3 and TAR DNA binding protein (Tardbp).

We also leveraged data from recent protein-wide association studies of post-mortem human brains from participants in the Religious Orders Study and the Rush Memory and Aging Project (ROSMAP) longitudinal study in which post-mortem brain proteins correlated with rate of cognitive decline, were identified. Using cognitive slope as the outcome, proteins positively correlated with cognitive stability (a positive cognitive slope) represented pro-resilience proteins (n = 645). Conversely, proteins that were negatively correlated with cognitive stability (a negative cognitive slope), represented anti-resilience proteins (N = 575)[54,55]. As compared to the Camk2a-CIBOP proteome, the PV-IN proteome was significantly enriched in pro-resilience proteins, including complexins (Cplx1, Cplx2), Ank1, highly-abundant PV-IN proteins (e.g. Aak1, Cttn, Bin1, Elfn1, Bsn) as well as ribosomal, mitochondrial, GTP binding, synaptic compartment and vesicle fusion proteins (Fig. 2F). While PV-CIBOP is selective to the PV-INs, Camk2a-CIBOP broadly labels several classes of excitatory neurons in the cortex[56,57], which limits the ability to detect molecular characteristics of distinct excitatory neuronal sub-classes. Therefore, future studies should apply more selective AAV or transgenic approaches to contrast distinct excitatory neuronal sub-classes.

This evaluation of PV-IN proteomes, in contrast to Camk2a neurons, reveal a generalized molecular phenotype of high translational, synaptic vesicle transport and fusion (neurotransmission), GTP binding and signaling (Akt/mTOR) activities in PV-INs, including many AD-related genetic risk factors and proteins associated with cognitive resilience.

## Network analyses of human post-mortem brain proteomes identify unique associations of PV-IN markers with neuropathology and cognitive dysfunction in AD

Supporting evidence for unique vulnerability of PV-INs to neurodegenerative disease pathology in AD may be revealed by analyses of bulk brain proteomes. We interrogated published human post-mortem bulk brain proteomic studies[58–67] which included >500 post-mortem dorsolateral pre-frontal cortex samples (non-disease controls, Asymptomatic AD, and AD with dementia cases) from the ROSMAP and Banner Sun Health cohorts[66,67]. Over 8,000 proteins quantified in this previous study coalesced into 44 groups of highly co-expressed proteins (modules, annotated as M1-M44)[68], some of which were enriched in markers of distinct brain cell types (eg. M1, M5, M10, M33 for neurons, M11 and M12 for astrocytes, M11 and M21 for microglia, M3 for oligodendrocytes), as well as distinct molecular mechanisms (eg. MAPK signaling in module M7)[68]. These modules also have unique correlations to amyloid burden (CERAD), neurofibrillary tangles (Braak stage), and cognitive function (MMSE or global cognitive function)

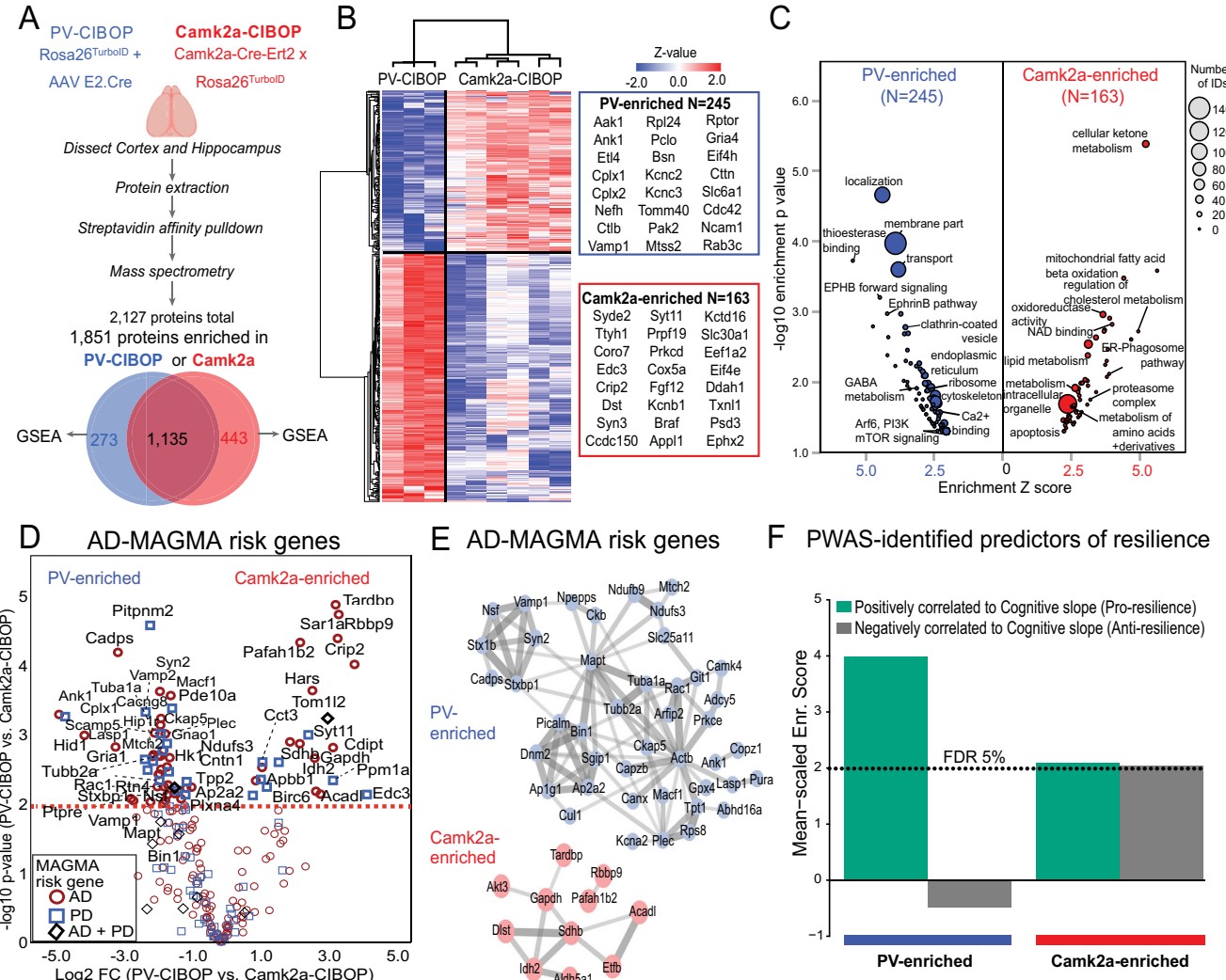

**Fig. 2 | Distinct proteomic signatures and disease vulnerabilities of PV-INs and Camk2a excitatory neurons revealed by CIBOP. A** Experimental outline for comparative analysis of CIBOP-based proteomics of PV-INs and Camk2a excitatory neurons from the mouse cortex by label-free quantitation MS analyses. 1851 proteins were quantified above negative samples in either PV-INs or Camk2a neurons. **B** DEA comparing PV-CIBOP (n = 3) and Camk2a-CIBOP (n = 6) cortex proteomes identified proteins >3-fold differential enrichment (signature proteins of each neuronal class), which were then hierarchically clustered. Top proteins (based on fold-change) are shown alongside the heatmap. **C** GSEA of PV-IN (blue) and Camk2a (red) signature proteins identified over-represented terms (GO, KEGG, Wikipathways, Reactome, Pathway commons) for PV-IN and Camk2a neurons. *X*-axis represents enrichment Z score for a given term, and *Y*-axis represents level of statistical significance of enrichment (Fisher exact test). Size of each data point indicates the number of protein IDs in that enrichment term. **D** Volcano plot representation of PV-IN and Camk2a neuron signature proteins which have known genetic risk associations in Alzheimer's disease (AD) and Parkinson's disease (PD)

based on MAGMA. Some proteins have shared genetic risk associations with AD and PD (Y-axis: Camk2a-CIBOP vs. PV-IN-CIBOP unpaired two-tailed *T*-test *\*p* < 0.05). **E** Protein-protein interaction network (STRING) of AD-associated MAGMA risk genes with differential enrichment in PV-INs and Camk2a neurons. Clusters of mitochondrial, synaptic vesicle and endocytosis related proteins were revealed in PV-IN AD MAGMA risk genes. **F** Enrichment of PWAS-identified proteins associated with cognitive slope in PV-enriched and Camk2a-enriched proteomic signatures. Cognitive slope was estimated in ROSMAP cases. Positive slope indicates cognitive stability or resilience while a negative slope indicates cognitive decline. Proteins positively correlated with cognitive slope are referred to as pro-resilience proteins while those negative correlated with cognitive slope are anti-resilience proteins. Enrichment of pro-resilience and anti-resilience proteins in PV-enriched and Camk2a-enriched proteins identified by CIBOP was assessed after weighting based on strength of association between proteins and cognitive slope. FDR 5% threshold is shown. Also see Supplementary Fig. 4 and Supplementary Data 2 for related analyses and datasets. Image was created using BioRender.com.

(Fig. 3A)[58,67]. Therefore, these modules can be treated as groups of proteins representative of distinct disease mechanisms, some of which may be cell type-specific.

We analyzed this human post-mortem brain proteomic dataset, now using sets of neuron sub-class-specific markers, to identify protein modules that may represent disease mechanisms associated with distinct neuronal classes. We derived 1,040 genes with ≥4-fold enrichment in specific neuronal classes (glutamatergic pyramidal neurons and GABAergic neurons including PV, Sst, VIP subtypes) from the Allen brain single cell/nuclear RNAseq (sc/snRNAseq) atlas (Supplementary Data 3). Pan-excitatory/glutamatergic markers (Camk2a, Slc17a7) were

enriched in modules M1, M5, M22, M10 and M4. Pan-inhibitory markers (Gad1-2, Slc32a1) were enriched in modules M33 and M23. Among the inhibitory neuron modules, M33 showed enrichment in PV-IN markers, including Pvalb and Kcnc2 [Kv3.2]), while VIP interneuron markers were enriched in M23 (Fig. 3B). Therefore, M1 (as well as M22, M5, M10 and M4) is enriched in cellular mechanisms of AD in excitatory neurons, while M33 represents AD patho-mechanisms impacting PV-INs. Both pan-excitatory M1 and PV-IN M33 module abundances were lower in AD cases (Fig. 3C, D) and were associated with cognitive function (MMSE and global cognitive function) at last clinical visit prior to death, and negatively associated with severity of dementia and

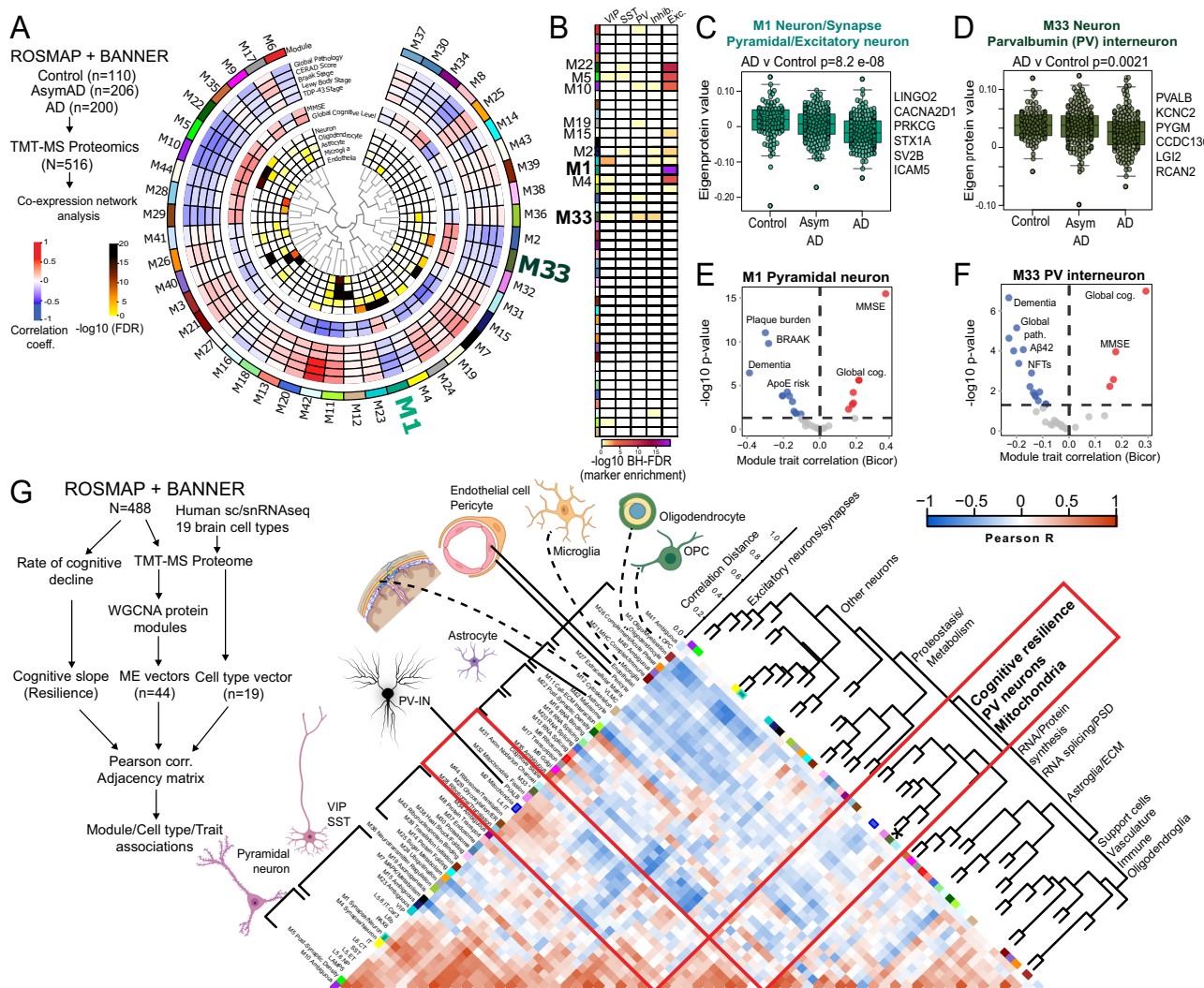

**Fig. 3 | PV-IN molecular signatures are associated with neuropathology and cognitive resilience in humans. A** Summary of network-based analysis of human bulk brain proteomes derived from post-mortem frontal cortex samples from controls, AsymAD and AD cases, from ROSMAP and BANNER cohorts (adapted from Johnson et. al.)[70]. Protein co-expression modules (M1-42) are arranged in a circular manner. Dendrogram indicates module inter-relatedness. Module trait associations are arranged in layers: Cell type signatures with cell type-enrichment statistical significance (-log10 FDR; module-cognitive trait associations (MMSE and Global cognitive score); module-neuropathological trait associations (red: positive, blue: negative). **B** Modules that showed over-representation distinct neuronal markers (pan-excitatory, pan-inhibitory and IN classes, namely PV-IN, SST-IN and VIP-IN) are shown. Color indicates level of statistical significance of enrichment. **C, D** Comparisons of module eigenprotein abundances of M1 (C, pan-excitatory neuronal module) and M33 (D, PV-IN module) across controls, AsymAD and AD cases. Overall ANOVA *p*-value is shown and top neuronal class-specific proteins representative of M1 and M33 are highlighted. **E, F** Volcano plot representations of Module-trait correlations (*X*-axis: Bicor, *Y*-axis: -log10 *p*-value of correlation) for M1 (**E**) and M33 (**F**). Top correlated traits (including ApoE genetic risk based on allelic combinations of ApoE ε2, 3 and 4) are labeled. **G** Adjacency matrix analysis based on correlations between protein co-expression modules (ME vectors), Cell type abundance vectors (based on selected markers of 19 distinct brain cell types identified by sc/snRNAseq) and cognitive slope till time of death (Left: overall outline of the analytical plan). Right: Heatmap representation of Pearson's correlation-based adjacency matrix. Cell type vectors and module ME vectors with representative ontologies of each ME, are shown on the left of the heatmap. Dendrogram on the right indicates relatedness among ME vectors, cell type vectors and cognitive slope, and revealed a cluster of PV-IN module M33, PV-IN vector, mitochondrial module M2 and M32, and cognitive slope (see Supplementary Data 3 for additional details). See Supplementary Data 3 and Supplementary Fig. 7 for related analyses. Image was created using BioRender.com.

neuropathology (both Aβ and tau) (Fig. 3E, F) although APOE genetic risk was associated with M1, but not M33. This analysis shows that markers of distinct neuronal sub-classes are indeed enriched in specific protein modules, some of which have correlations with clinical and pathological traits and are therefore likely to be pathologically relevant.

We next assessed whether excitatory and inhibitory neuronal signatures obtained from bulk brain proteomes, are associated with rate of cognitive decline (cognitive slope) in longitudinal studies of aging and AD[55,69] (Fig. 3G and Supplementary Fig. 7A, B). Using consensus marker lists of cell types from reference human brain single

cell/nuclear RNAseq studies, we applied single-sample GSEA (ssGSEA) to proteomic data from 488 post-mortem human dorsolateral prefrontal cortex (DLPFC) samples from ROSMAP/BANNER studies, to estimate abundances of 19 classes of excitatory and inhibitory neurons, glia and vascular cells (Fig. 3G)[1,70–72]. We also assessed whether protein co-expression modules, some of which are specific to neuronal sub-classes, also correlated with cognitive slope. The associations between different cell type abundances, protein modules and rate of cognitive decline were assessed using a correlation-based adjacency matrix. We found that the PV-IN cell type abundance, modules M33 (enriched in PV-IN markers), M2 and M32 (mitochondria) as well as

cognitive slope (cognitive resiliency) were highly inter-correlated (Fig. 3G). In addition to PV-INs, the layer 4 IT neuronal estimate was also positively correlated with cognitive slope (Supplementary Fig. 7A, B). The abundances of these two cell types are therefore linked to cognitive resilience. The association between PV-IN module M33 and cognitive resilience remained significant after adjusting for several co-existent neuropathologies (including amyloid, tau, synuclein), suggesting that the relationship between PV-IN abundance and cognitive resilience may be independent of neuropathology (Supplementary Fig. 7C)[55].

Overall, these network-based and correlative analyses of bulk brain proteomic data indirectly suggest that distinct neuronal sub-classes may be differentially impacted in AD pathology, and support the idea that integrity of PV-INs and/or their proteins, may be determinants of cognitive resilience in AD.

## Neuron-specific molecular signatures resolve vulnerability of PV neurons to progressive Aβ pathology in the 5xFAD mouse model

The cross-sectional nature and post-mortem sampling of tissue in human post-mortem brain proteomic studies, limit the ability to resolve the impact of aging and disease progression on neuronal protein changes in AD. Since these can be better assessed in disease models, we examined bulk brain proteomes from a mouse model of Aβ pathology. We analyzed TMT-MS data from 43 WT and 43 5xFAD mice (age span 1.8-14.4 mo., 50% male, 50% female), in which over 8,500 proteins were quantified (Fig. 4A, Supplementary Data 4). As expected in 5xFAD mice, age-dependent increase in Aβ pathology was associated with concomitant increase in levels of Apoe, microglial proteins (Trem2, Msn, C1qb) and astrocyte proteins (Gfap) (Fig. 4A). Using 10 and 14 month timepoints, we assessed whether proteomic changes at more established and advanced stages of Aβ pathology in 5xFAD mice, are representative of changes occurring in post-mortem human AD brain using proteomes (control, AD and AsymAD) from the published resource also used in Fig. 3[68]. We observed proteome-wide moderate positive correlations in AD pathological changes occurring across species (Supplementary Fig. 8A–C). Mechanisms involving splicing, proteasome, innate immunity, actin-cytoskeleton and MAPK signaling were increased in human AD and mouse 5xFAD brains (Supplementary Fig. 8D). Conversely, mitochondrial respiration, mitochondrial translation, endocytosis and synaptic signaling were decreased in human AD and mouse 5xFAD brains (Supplementary Fig. 8E).

Based on this evidence of concordance of pathological changes across species based on our and recent proteomics studies[73], despite evident inter-species differences between humans and mice, we focused our analysis to canonical transcriptomic markers (Supplementary Data 4) of different neuronal classes[49], and identified distinct patterns of change of neuronal proteins with aging and genotype (5xFAD vs WT), as well as biological interaction between aging and genotype. Excitatory neuronal proteins (Camk2a, Slc17a7) showed an age-dependent decrease without an impact of genotype. In contrast, Pvalb and Sst proteins (found in PV-INs and SST GABAergic INs, respectively) increased with age in WT mice, but this trend was significantly blunted by 5xFAD genotype, particularly at 6 months of age (Fig. 4B). Kcnc3, which encodes a Kv3 channel highly expressed by PV-INs, showed an age-dependent decrease in 5xFAD mice while Vip, a marker of VIP-INs, did not show changes related to either age or genotype. We also observed a strong positive correlation between PV-IN proteins (Pvalb and Kcnc3) and myelin proteins (Plp1 and Mbp) (Fig. 4A) indicative of cell-cell interactions between PV-INs and oligodendrocytes[74].

Using lists of markers enriched in pan-excitatory, pan-inhibitory neurons, as well as in sub-classes of PV-INs, SST-INs and VIP-INs[49], we calculated composite cell-type-specific abundance scores of neuronal classes, to identify age and genotype (5xFAD vs. WT) effects in the bulk proteomic data (Fig. 4C). Pan-excitatory markers decreased with aging, and were minimally higher in abundance in 5xFAD brains as compared to WT, regardless of age. Pan-inhibitory markers also showed an age-dependent decrease in abundance which became more pronounced in 5xFAD mice. Among major IN classes, PV-IN markers showed a gradual increase with age in WT mice, although this was suppressed in 5xFAD brain (Fig. 4B). In contrast, SST-IN and VIP-IN markers were unaffected by genotype (Fig. 4C).

To determine whether PV-IN protein changes in 3-6 month-old 5xFAD mice are related to changes in PV-IN cell numbers or PV-IN proteins[75], we performed IHC studies on an independent set of 3 and 6 month-old WT (n = 7-8) and 5xFAD (n = 7-8) brains, and assessed Pvalb protein levels along with detection of peri-neuronal nets (PNNs) by Wisteria floribunda agglutinin (WFA) in the cortex and subiculum (Fig. 4D–G). PNNs are known to disproportionately encapsulate PV-INs in the brain and are key regulators of PV-IN excitability[76–78]. The number of Pvalb protein-positive neurons in the cortex and subiculum were not impacted by genotype (5xFAD vs WT) at 3 or 6 months of age (Fig. 4D, E, Supplementary Fig. 9) although the proportion of PNN-positive PV-INs was significantly lower in 5xFAD mice (Fig. 4G).

Overall, our analyses of mouse brain proteomic data identify unique age and Aβ-dependent changes that appear to differentially impact neuronal sub-classes, where PV-INs may be selectively vulnerable in early stages of pathology in 5xFAD mice[13]. Our histological studies suggest that observed proteomic changes in PV-IN proteins in 5xFAD brain is not explained by changes in the absolute numbers of PV-INs[79], although the health of PV-INs may be perturbed, as indicated by loss of PNNs around PV-INs in early stages of Aβ pathology and Aβ-induced changes in PV-IN proteins abundances. To better understand the molecular basis for differential vulnerability of PV-INs to AD pathology with spatiotemporal resolution, neuron class-specific native state proteomic investigations using CIBOP are warranted.

## Unique molecular signatures of PV-IN proteome in early stages of Aβ pathology in 3 month-old 5xFAD mice

To identify molecular events specifically occurring in PV-INs because of early AD pathology, we applied the PV-CIBOP strategy to achieve PV-IN-specific TurboID expression, biotinylation and fluorescent labeling in WT and 5xFAD mice (Fig. 5A). Three weeks after RO AAV injection and 2 weeks of biotinylation, mice were euthanized at 3 months of age. This early stage of Aβ pathology (i.e., significant Aβ burden with minimal plaque formation restricted to the subiculum) was chosen to capture potentially-modifiable disease mechanisms. IHC confirmed PV-IN-specific biotinylation in the cortex of WT and 5xFAD PV-CIBOP mice (Fig. 5B). Flow cytometry of enzymatically-dissociated cortex[7] confirmed equal efficiency of PV-IN targeting across all groups (Fig. 5C). Consistent with early stage of pathology, we observed minimal extracellular Aβ plaque pathology in the subiculum and cortex of 5xFAD mice, while total Aβ42 levels measured by ELISA were substantially increased in 3 month-old 5xFAD mice (Fig. 5D). WBs of cortical lysates showed robust biotinylation and V5 protein signals in all PV-CIBOP mice (Fig. 5E). WB of SA-enriched samples showed enrichment of biotinylated proteins in PV-CIBOP animals compared to controls, regardless of 5xFAD or WT genotype (Fig. 5F, Supplementary Fig. 10A).

LFQ-MS of bulk brain samples (inputs) and SA-enriched (i.e., PV-specific) samples quantified 3,086 proteins and 2,149 proteins respectively (Supplementary Data 5). PCA of bulk proteomes indicated minimal effects of biotinylation on the overall brain proteome (Supplementary Fig. 10B). 1,973 proteins were enriched in the PV-IN proteome from both WT and 5xFAD PV-CIBOP groups as compared to negative control SA-enriched proteomes. Top PV-IN enriched proteins identified in WT/PV-CIBOP tissues in this experiment, including equal levels of enrichment of three Kv3 (3.1, 3.2, 3.3) channel proteins, agreed with findings in the first PV-CIBOP study in Fig. 1 (Supplementary Fig. 10C). PCA resolved differences between PV-CIBOP proteomes

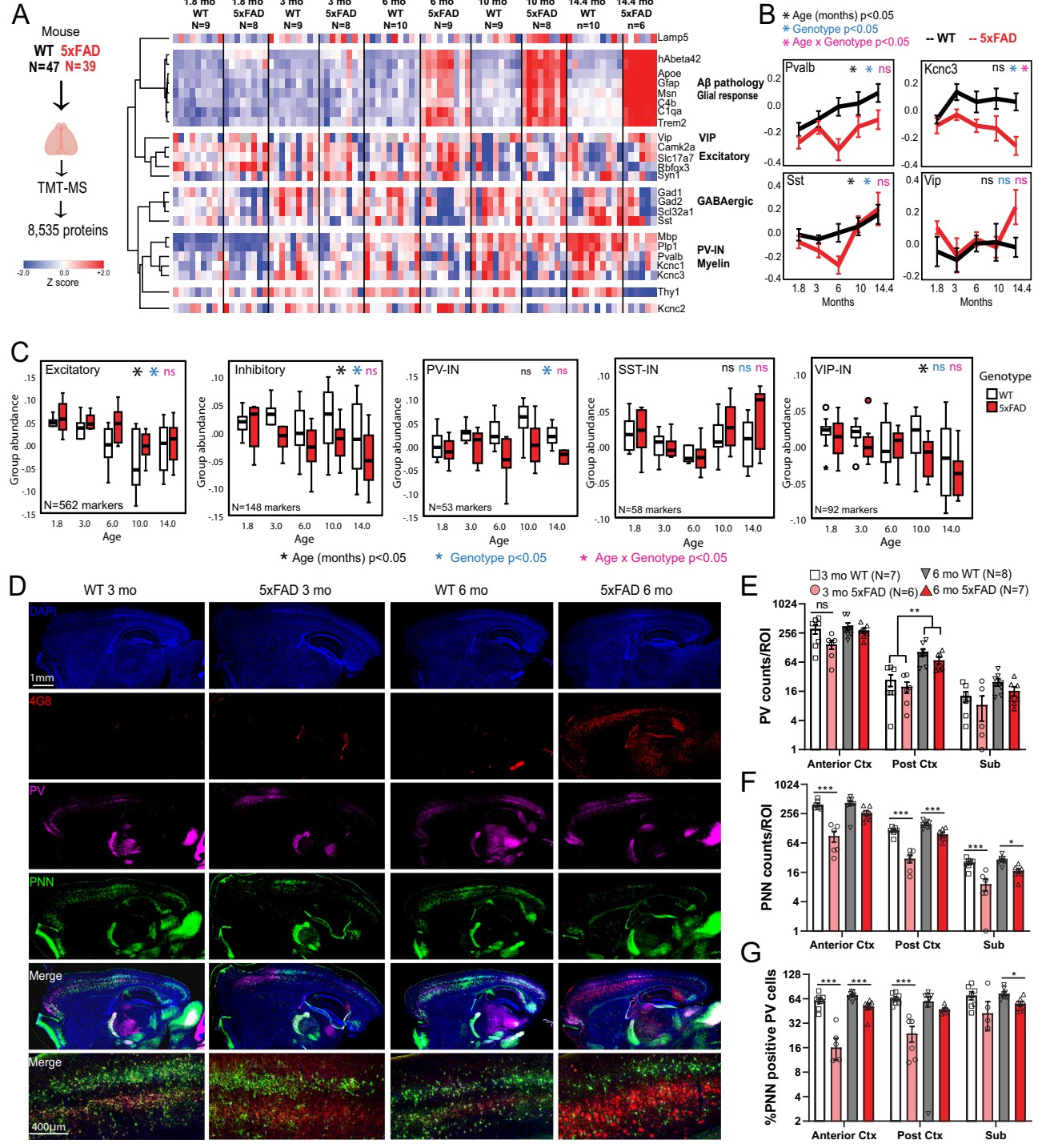

**Fig. 4 | Bulk tissue proteomics of mouse brain reveals differential effects of Aβ pathology and aging on PV-INs and their peri-neuronal nets. A** Study outline for analysis of mouse bulk brain (cortex) TMT-MS proteomics data. 8,535 proteins were quantified by TMT-MS from 47 WT and 39 5xFAD mouse brains. From these, selected proteins reflective of Aβ pathology (hAβ42 peptide), were visualized as a heatmap after hierarchical clustering based on protein IDs (N/group for panels **A**–**C** are indicated). **B** Trajectories of change in levels of PV-IN proteins (Pvalb, Kcnc3), SST-IN (Sst) and VIP-IN (Vip) based on age and genotype. Error bars represent SEM. Statistical tests included linear regression analyses with age, genotype and 'age x genotype' interaction terms as covariates. Levels of significance of each are indicated. **C** Trajectories of change in overall levels of pan-excitatory, pan-inhibitory, as well as PV-IN, SST-IN, and VIP-IN proteins, based on age and genotype. We used lists of transcriptomic markers of these classes of neurons (from sc/snRNAseq datasets) that were at least 4-fold enriched in the class of interest over all other neuronal types. After normalizing and z-transforming proteomic data, neuronal class-based

group abundance scores were calculated and compared across ages and genotypes. Linear regression analyses were performed using age, genotype and age x genotype interaction term as covariates. Levels of significance of each, are indicated (Box plots: Median, inter-quartile range, min and max are shown).
**D** Representative images from immunofluorescence microscopy studies of mouse brain (sagittal sections, WT and 5xFAD, ages 3 and 6 months, animals used for TMT-MS studies in **A**), to detect PV-INs (Pvalb protein), perineuronal nets (WFA lectin), Aβ pathology (4G8) and DAPI. 4x tiled images and 20x images from cortex are shown. **E**–**G** Quantitative analysis of Pvalb protein, PNNs, and proportion of Pvalb+ INs that have PNNs in the cortex and subiculum of WT and 5xFAD mice at 3- and 6-mo of age. Y-axes are log2 transformed. Error bars represent SEM (Post-hoc Tukey HSD pairwise comparisons; *p < 0.05, **p < 0.01, ***p < 0.005). See Supplementary Fig. 5 and Supplementary Data 4. Source data are provided as a Source Data file. Image was created using BioRender.com.

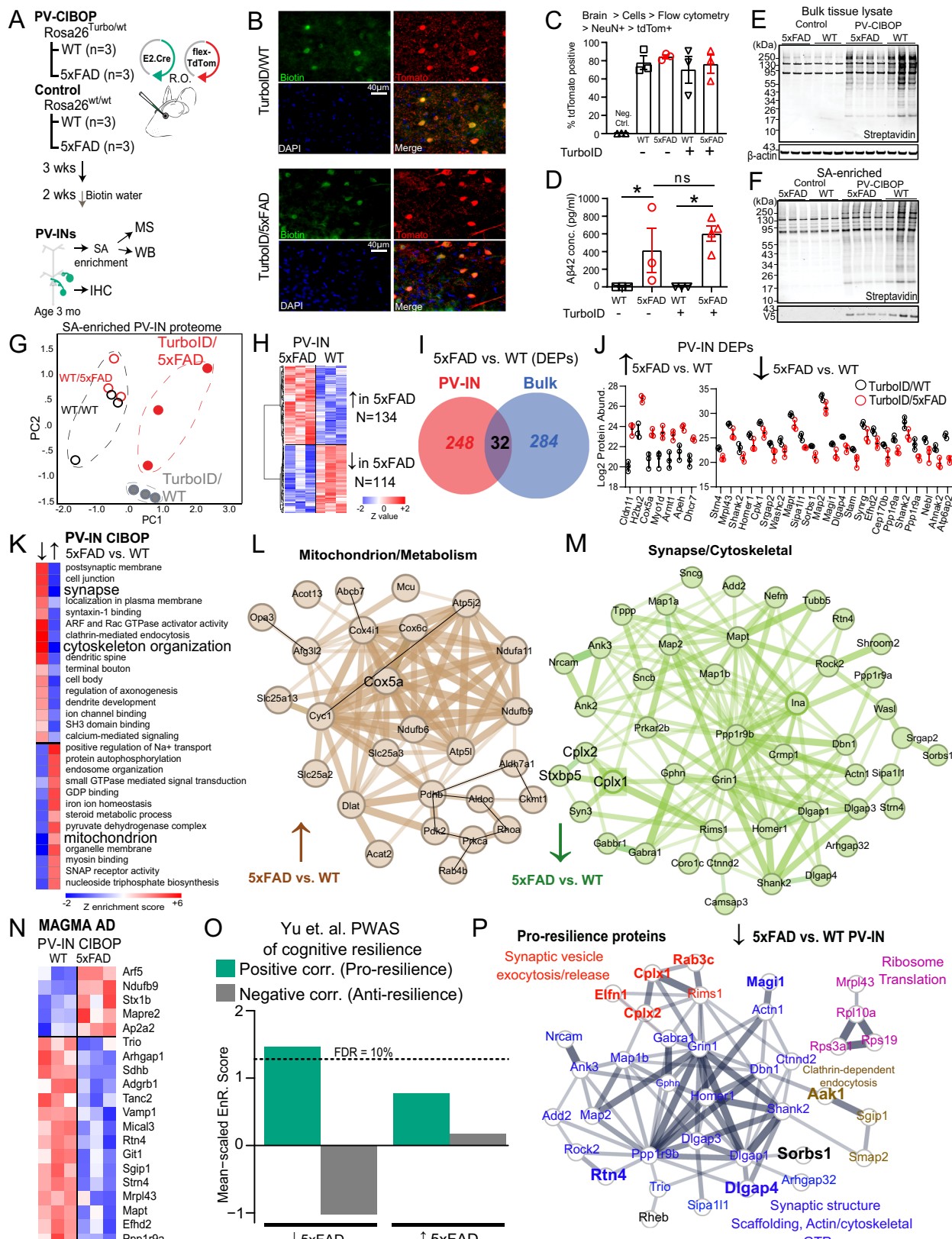

from WT and 5xFAD mice (Fig. 5G). We identified 248 differentially enriched proteins (DEPs) in PV-CIBOP proteomes from Turbo-expressing WT and 5xFAD mice, which included 134 proteins that were increased and 114 decreased in 5xFAD PV-INs (Fig. 5H, Supplementary Data 5). In alignment with our physiology studies[33], Kv3 protein levels did not vary in young 5xFAD mice (p = 0.75, 0.10, 0.25 for

Kv3.1, 3.2, and 3.3, respectively; unpaired T-tests). Overall, the PV-IN-specific DEPs in 5xFAD mice had very little overlap with DEPs identified in analyses of bulk brain proteomes from the same mice (Fig. 5I, Supplementary Data 5, Supplementary Fig. 10D). At the level of gene ontologies identified using gene set variation analyses (GSVA), the ontologies differentially enriched in the PV-IN proteome also showed

**Fig. 5 | PV-IN proteomic alterations in early stages of Aβ pathology in the 5xFAD model. A** Experimental outline for PV-CIBOP in 3-month-old 5xFAD mice. **B** IHC studies confirming PV-IN-specific biotinylation in WT and 5xFAD PV-CIBOP mice. **C** Flow cytometry analyses showing AAV-mediated targeting efficiency of PV-INs across experimental animals (n = 3/group; One-way ANOVA, p = 0.78; Data shown as mean ± SEM). **D** Aβ42 ELISA measurements from bulk cortex homogenates, confirming comparable Aβ42 levels across groups (n = 3 each for WT, FAD and Turbo WT, n = 4 Turbo 5xFAD, Data shown as mean ± SEM *p < 0.05, unpaired two-tailed T-test). **E, F** WB from bulk cortical tissue lysates and from SA-enriched samples showing robust biotinylation in PV-CIBOP compared to non-CIBOP mice. **G** PCA of MS data from SA-enriched proteomes: All PV-IN proteomes clustered away from control samples, and further distinction was observed between 5xFAD and WT PV-IN proteomes. **H** Heatmap representation of DEPs comparing WT/PV-CIBOP and 5xFAD/PV-CIBOP SA-enriched proteins. **I** PV-IN-specific DEPs minimally overlap with bulk tissue DEPs in 5xFAD and WT mice. **J** Top DEPs (showing at least 4-fold differential enrichment) comparing 5xFAD to WT PV-IN proteomes are shown

(n = 3/group, mean ± SEM shown). **K** GSEA of DEPs comparing 5xFAD to WT PV-IN proteomes. **L, M** STRING protein-protein-interactions (PPI) within DEPs identified in Mitochondrial (**L**, increased in 5xFAD PV-IN) and Synaptic/Dendritic/Cytoskeletal (**M**, Decreased in 5xFAD PV-IN) ontologies. Thickness of edges indicates strength of known interactions. **N** Heatmap representation of DEPs comparing 5xFAD to WT PV-IN proteomes, limited to proteins encoded by genes with known genetic risk associations in AD (AD-MAGMA significance p < 0.05). **O** Enrichment of pro-resilience and anti-resilience proteins (from Yu et. al. PWAS study) within lists of DEPs (5xFAD vs. WT PV-IN proteomes). FDR 10% threshold is shown. **P** STRING PPIs of PWAS-nominated proteins positively associated with cognitive resilience (pro-resilience) that are decreased in 5xFAD PV-INs based on PV-CIBOP studies. Colors indicate shared functions and/or ontologies. Of these, proteins that are also selectively enriched in PV-INs as compared to Camk2a neurons (from CIBOP studies in Fig. 3) are highlighted (larger font, and bold). See Supplementary Fig. 6 and Supplementary Data 5 for related analyses. Source data provided as a Source Data file.

minimal agreement with changes occurring in the bulk proteome (Supplementary Data 5). One example of stark discordance between bulk and PV-IN proteomic changes was the oxidative phosphorylation/aerobic respiration gene set, which was increased in PV-INs but decreased in the bulk proteome (5xFAD vs. WT). In contrast, synaptic proteins were decreased in both bulk and PV-IN proteomes, but the magnitude of this decrease was larger in PV-INs compared to bulk proteomes (Supplementary Data 5). This observed lack of concordance between the effects of 5xFAD genotype on the PV-IN proteome as compared to the bulk brain proteome, suggests unique metabolic/respiratory perturbations in PV-INs, that are most likely diluted in bulk tissue analyses.

Among DEPs identified in the PV-IN proteome, top proteins showing at least 4-fold increase in 5xFAD PV-INs included Cox5a, Dhcr7 and Apeh (Fig. 5J). Cox5a is a Complex IV mitochondrial protein involved in ATP synthesis[80]. Dhcr7 encodes 7-dehydrocholesterol reductase that catalyzes final rate limiting steps of cholesterol biosynthesis[81]. Apeh encodes acylaminoacyl-peptide hydrolase that hydrolyses terminal acetylated residues in small acetylated peptides, including degradation of monomeric and oligomeric Aβ[82]. Synaptic structural proteins including Shank2, Homer1, Map2 were conversely decreased by at least 4-fold in 5xFAD PV-INs (Fig. 5J). GSEA and GSVA identified in PV-IN proteomes showed that mitochondrial function, steroid biosynthesis, small GTPase signaling, and GDP binding were increased in 5xFAD PV-INs while structural/cytoskeletal, synaptic, axonal and dendritic ontologies were decreased in 5xFAD PV-INs (Fig. 5K, L, Supplementary Data 5).

We identified 36 post-synaptic proteins (post-synapse GO:0098794) that showed decreased levels in 5xFAD PV-INs, including structural constituents of the post-synapse (Dlgap1, Homer1, Gphn, Ina, Shank2, Git), enzymes with kinase activity or binding (Ppp1r9b, Bcr, Rtn4, Rheb, Prkar2b, Map2), neurotransmitter receptors (Grin1, Gabra1, Gabbr1), dendritic spine proteins (Grin1, Homer1, Dlgap3, Shank2, Dbn1, Bai1, Tanc2, Ncam1) and ribosomal subunits (Rps19, Rpl10a). Pre-synaptic proteins involved in synaptic vesicle fusion and exocytosis, including Complexins (Cplx1, 2), showed decreased levels in 5xFAD PV-INs (Fig. 5M). Of note, Cplx1 and 2 (but not Cplx3) were more also abundant in PV-INs as compared to Camk2a neurons in our CIBOP studies (Supplementary Fig. 10), in line with a recent proteomic study comparing excitatory and inhibitory synaptosomal fractions[83]. Several MAGMA-identified AD genetic risk factors also showed differential abundances in 5xFAD PV-INs as compared to WT PV-INs (e.g., decreased- Ppp1r9a, Mapt, Git1; increased- Arf5, Ndufb9, Stx1b (Fig. 5N).

To predict whether changes in the PV-IN synaptic protein landscape would result in detrimental or protective consequences, we cross-referenced the 5xFAD vs. WT PV-IN DEP list against pro-resilience and anti-resilience brain proteins identified in human studies

(Supplementary Data 5)[36]. We found that pro-resilience proteins were over-represented while anti-resilience factors were under-represented in proteins that decreased in 5xFAD PV-INs (Fig. 5O). The pro-resilience proteins which were decreased specifically in PV-INs in early 5xFAD pathology included synaptic structural, synaptic scaffolding, actin/cytoskeleton (Dlgap13/4, Shank2, Homer1, Dbn1, Map1b, Map2, Ank3), ribosome (Rpl10a, Rps19, Mrpl43), mTOR-C1 regulating protein (Rheb), clathrin-dependent endocytic (Aak1, Sgip1, Smap2) and synaptic vesicle fusion/exocytosis/release related proteins (Cplx1, Cplx2, Elfn1, Rab3c, Rims1) (Fig. 5P). Many of these pro-resilience proteins that were decreased in 5xFAD PV-INs (e.g. Cplx1, Cplx2, Elfn1, Rab3c, Rtn4, Dlgap4, Sorbs1, Magi1) were also highly-enriched in PV-INs as compared to Camk2a neurons (Fig. 5P, Supplementary Fig. 10E, F), indicating that the changes in early AD pathology may indeed be related to PV-IN dysfunction.

Overall, these mouse-human integrative analyses indicate that altered levels of PV-IN bouton and dendritic proteins at early stages of Aβ pathology may result in detrimental synaptic or network effects, and thus represent proteomic changes potentially related to cognitive resilience with AD pathology. Thus, we next aimed to understand the functional impact of these PV-specific synaptic proteomic alterations.

### Early Aβ pathology impacts PV-pyramidal cell neurotransmission and network activity
We leveraged the 'E2' enhancer to express optogenetic actuators in PV-INs (Fig. 6, Supplementary Fig. 11) to measure PV-specific neurotransmission properties, where postsynaptic pyramidal cell recordings represent an integrated response to PV-specific, action potential-evoked neurotransmission. C1V1 (AAV.E2.C1V1) was injected in both WT and 5xFAD mice simultaneously with AAV.Camk2.YFP, which served to confirm accurate viral targeting (Fig. 6A). Acute slices were taken from two separate age cohorts (~2 or ~3 months old, 7 or 14 weeks respectively) and voltage clamp recordings were obtained from postsynaptic L5 pyramidal neurons (Fig. 6B). Short (~0.2 ms) LED-light pulses (590 nm) reliably evoked IPSCs every trial (0.1 Hz inter-trial interval) in both WT and 5xFAD recordings with minimal temporal jitter. The amplitude of the first IPSC amplitude was unchanged in both 2- and 3-month-old 5xFAD mice (e.g., 2-month-old WT: 133.5 ± 25.90 pA, n = 10; 3-month-old 5xFAD: 163.7 ± 30.55 pA, n = 7; p = 0.46, unpaired t-test), suggesting that GABA_A receptor availability was similar in the postsynaptic pyramidal cells. As several synaptic vesicle fusion/exocytosis/release related proteins were altered in our 5xFAD PV-IN proteome, we more closely evaluated presynaptic properties (i.e., release probability) of PV-pyramidal synapses.

To examine whether modification of vesicle fusion and association proteins affected release probability and presynaptic dynamics at PV-pyramidal synapses, we measured the paired pulse ratio (PPR)[84] and

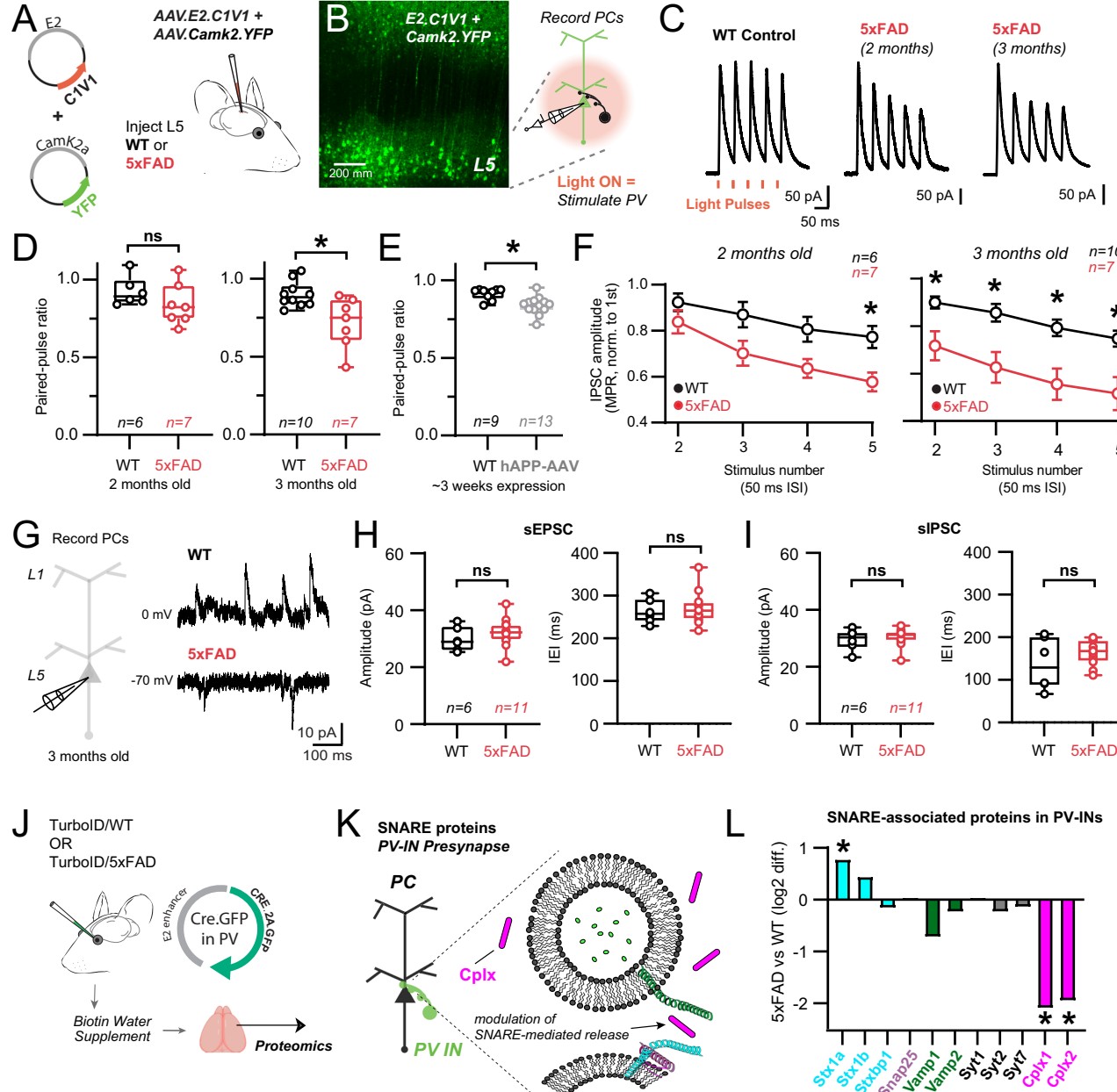

**Fig. 6 | Progressive dysfunction of PV-pyramidal cell neurotransmission in young 5xFAD mice. A** E2.AAV-strategy for optogenetic activation of S1 PV-INs in WT and 5xFAD mice and co-labeling of Camk2a pyramidal neurons. **B** 2-photon image showing successful targeting of L5 following ~1 week after stereotactic surgery. Cartoon: Experimental workflow to stimulate PV-INs and record their synaptic properties in post-synaptic pyramidal cells (*n* = 3 mice/condition). **C** Averaged voltage-clamp traces from postsynaptic WT and 5xFAD pyramidal cells in layer 5 in response to short-amber light pulses. PV-IN IPSCs shown as time-locked to amber light pulses in a 20 Hz train. **D** Quantification of paired-pulse ratios (PPR) of C1V1-evoked IPSCs in pyramidal cell recordings for 2- and 3-month-old WT and 5xFAD mice (2 months: *p* = 0.21; 3 months: *p* = 0.008). **E** Quantification of PPR of AAV1.-DIO.ChETA-evoked IPSCs in pyramidal cell recordings from PV-Cre mice-3 weeks following hAPP-AAV injections in L5 (*p* = 0.001). **F** Quantification of changes in PV synapse dynamics (multiple pulse ratio/MPR) in WT and 5xFAD mice (2 months: (left to right) *p* = 0.63, *p* = 0.07, *p* = 0.07, *p* = 0.03; 3 months: (left to right) *p* = 0.04, *p* = 0.005, *p* = 0.003, *p* = 0.004; Two-way ANOVA with Sidak's posthoc

comparisons for each stimulus in WT and 5xFAD experiments). Data are displayed as mean + /- SEM. **G** Voltage-clamp experiments were performed examining spontaneous synaptic activity in 3-month-old WT and 5xFAD mice. Holding voltage was interleaved between -70 and 0 mV to resolve spontaneous EPSCs and IPSCs, respectively. **H** Quantification of spontaneous EPSC amplitude and frequency in 3-month-old 5xFAD and WT mice. Data points indicate average from all spontaneous events from individual recordings (sEPSC Amplitude: *p* = 0.36); (sEPSC IEI: *p* = 0.69). **I** Quantification of spontaneous IPSC amplitude and frequency in the same recordings as in (H). Data points indicate an average value from all spontaneous events from individual recordings (sIPSC Amplitude: *p* = 0.67); (sIPSC IEI: *p* = 0.25). For box plots D-I: Median, inter-quartile range, min and max are shown. **J–L** Pre-synaptic PV-IN SNARE proteins identified by CIBOP, that show differential enrichment in 5xFAD vs. WT mice (Average log2FC enrichment is shown, *\*p* < 0.05). Unpaired two-tailed t-tests used for all comparisons, except panel F. See Supplementary Fig. 7 and Source data files. Image was created using BioRender.com.

multiple pulse ratio (MPR) of optogenetically-evoked IPSCs at 20 Hz using 1.5 mM external $Ca^{2+}$ (Fig. 6C)[85,86]. Evaluation of the paired-pulse ratio showed modest depression (PPR ~ 0.9) in WT mice at both age timepoints. In 2-month-old 5xFAD mice, no difference in PPR was observed (Fig. 6D) although synaptic depression did intensify in 5xFAD mice during repetitive stimuli (MPR) at this age (Fig. 6F). However, in 3-month-old 5xFAD mice, a decrease in PPR emerged (Fig. 6D) and this more robust depression was now maintained throughout the stimulus train (Fig. 6F). Together these results show progressive presynaptic dysfunction in PV-pyramidal synapses following early Aβ pathology, consistent with changes in proteins regulating vesicular release probability (Fig. 5). Other mechanisms, such as proteins involved with vesicular docking and replenishment, axonal action potential signaling, and $Ca^{2+}$ dynamics could also contribute. For example, changes in presynaptic parvalbumin expression may affect short-term plasticity and release probability[87]. However, we did not observe a change in bulk parvalbumin protein by MS (Fig. 4B) or IHC (Supplementary Fig. 9) at 3 months of age.

We next asked whether the signature of synaptic dysfunction observed in 5xFAD mice could be recapitulated in an independent, adult-onset model of APP/Aβ pathology. We packaged the human APP gene (variant NM_000484.4) into an AAV (AAV.Ef1a.hAPP). This particular APP isoform was chosen as it has been shown to proportionally increase with aging and is associated with increased AD risk[88,89]. While overexpression of hAPP is expected to produce a significant increase in Aβ[90], this approach is distinct in that APP/Ab production is limited to the mature circuit, as in sporadic AD. 5-11 week old PV-Cre mice were co-injected with hAPP-AAV and AAV.DIO.CAG.ChETA for PV-specific optogenetic control Supplementary Fig. 11A. Control mice only received AAV.DIO.CAG.ChETA with saline to normalize overall viral titer across experiments. After 2-3 weeks expression, voltage clamp recordings were obtained from postsynaptic pyramidal neurons using 1.5 mM external $Ca^{2+}$. Brief (~4 ms) LED-light pulses (470 nm) could reliably evoke IPSCs on every trial (0.1 Hz inter-trial interval) in both control and hAPP-AAV groups, with minimal temporal jitter Supplementary Fig. 11B,C. Similar to 3-month-old 5xFAD mice, the amplitude of the first PV-PC IPSC was unchanged after hAPP expression (Control: $68.18 \pm 38.37$ pA, $n = 9$; hAPP-AAV: $72.70 \pm 44.49$ pA, $n = 13$; $p = 0.82$, unpaired t-test). Optogenetic stimulation with ChETA also showed modest PV-PC synaptic depression in controls (Fig. 6D vs 6E). While PV-pyramidal synapses are often regarded as strongly depressing, their PPR is often observed as only slightly depressing when external $Ca^{2+}$ is set at naturalistic levels[91–94], as we found here. Indeed, after elevation of external $Ca^{2+}$, we found a far more strongly depressing PPR (Supplementary Fig. 11E–G). Furthermore, basal synaptic properties were not attributable to direct $Ca^{2+}$ entry through the rhodopsin pore locally at the presynapse (Supplementary Fig. 11G). Overall, similar to experiments in 5xFAD, synaptic depression measured via PPR and MPR was again enhanced following adult-onset hAPP-AAV expression (Fig. 6E; Supplementary Fig. 11D). Together these results complement our findings in 5xFAD mice, highlighting the early emergence of presynaptic dysfunction at PV-pyramidal synapses following early Aβ pathology.

Alterations in evoked release in PV-INs may affect basal network excitability by disrupting excitatory/inhibitory balance. Thus, we next examined whether changes in the amplitude and frequency of spontaneous EPSCs and IPSCs were apparent in pyramidal cell recordings from 3 month old 5xFAD mice (Fig. 6G). Interestingly, no changes were observed in either the amplitude or frequency of excitatory and inhibitory spontaneous synaptic events (Fig. 6 H, I) echoing recent work from 3-month-old 5xFAD mice in hippocampal slices, where local circuit behavior and oscillations were also largely resilient to change[95]. This may be due to several factors, including the relatively low AP frequency of evoked PV firing in the slice. APP/Aβ pathology could also preferentially affect evoked release, but spare stochastic vesicular

fusion, as these two processes may be mechanistically independent in the same synapse[96].

## Potential molecular mechanisms of APP/Aβ pathology on altered neurotransmission

Our optogenetic findings indicate a selective disruption in evoked release at PV-pyramidal synapses. Thus, we next examined relevant SNARE-associated protein changes in 5xFAD mice using PV-CIBOP (Fig. 6J, K). Despite the fact that CIBOP does not explicitly target or isolate synaptic proteins, such as with more formal synaptosomal fractionation approaches, we still identified major SNARE-associated proteins in our proteomic datasets in both WT and 5xFAD mice (Fig. 6L) including Syntaxins, Snap25, Vamp, Synaptotagmins, and Cplx. Among this group of PV-IN SNARE associated proteins, only Cplx1 and 2 were significantly reduced in 3 month old 5xFAD. Interestingly, previous work has found that the loss of Cplx1 and 2 can alter evoked release without affecting spontaneous release, analogous to our findings in 5xFAD above[97].

## Evidence for extensive mitochondrial protein changes in PV-INs in response to early Aβ pathology

The extensive cell-type-specific proteomic alterations we observed in PV-CIBOP and bulk proteomes thus may be reflective of early homeostatic responses to maintain overall circuit functionality[98,99], or potentially related to metabolic stress[100], either directly or indirectly relating to Aβ pathology. Therefore, we next sought to explore the extent of changes to mitochondrial proteins and associated metabolic pathways specifically in PV-INs in early stages of Aβ pathology. Of 300 mitochondrial proteins (MitoCarta 3.0) biotinylated in PV-INs[101], 30 proteins were increased (e.g. Cox5a, Mpst, Ndufa11, Ckmt1) and 4 proteins that were decreased (Mrpl43, Septin4, Sdhb, Bphl) in 5xFAD compared to WT PV-IN proteomes (Fig. 7A). Proteins involved in complex III, complex IV, complex V, amino acid metabolism and protein homeostasis were differentially enriched in 5xFAD PV-IN proteomes, while mitochondrial central dogma (mitochondria-specific DNA, RNA and translation-related elements), complex II and detoxification related proteins were unaffected (Fig. 7B). In contrast to the PV-IN proteome, only 26 mitochondrial proteins were differentially enriched in the bulk brain proteome, which included only 4 shared DEPs (Fig. 7C). Furthermore, the overall level of concordance between bulk brain and PV-IN mitochondrial protein levels was negligible ($R^2 = 0.0001$). The increase of Cox5a levels in 5xFAD PV-INs but not in the bulk brain tissue was validated by Western Blot, using both bulk brain homogenates and PV-specific enriched proteins (Fig. 7D). In bulk brain MS proteomes, Cox5a showed age-dependent decrease in 5xFAD mice (Fig. 7E), a pattern in stark contrast to increased levels in PV-INs at 3 months. This overall pattern of increased abundances of mitochondrial proteins belonging to most mitochondrial compartments, may represent increased mitochondrial biogenesis to meet increased energy demands needed to sustain PV-IN functionality, particularly in the setting of emerging synaptic defects in early Aβ pathology.

## Evidence for decreased mTOR-C1 signaling in PV-INs in early Aβ pathology

The mitochondrial and synaptic derangements occurring in PV-INs in 5xFAD mice suggest that upstream signaling pathways may be dysregulated in PV-INs. Metabolic signaling pathways, including Akt/mTOR, are important regulators of mitochondrial biogenesis and turnover, as well as synaptic function in neurons while several MAPKs (ERK, p38 MAPK, Jnk) impact cell proliferation, synaptic function and survival[102–105]. We found that PV-CIBOP labeled 75 proteins involved in Akt/mTOR (eg. Mtor, Rptor, Eif4b) and MAPK (eg. Map2k1, Ras proteins, Pak2, Akt3, Mapk3, Mapk10) signaling pathways (Fig. 8A). Of these, few proteins showed increased (Rhoa, Prkca, Hras, Cacna2d1) and decreased levels (Eif4b, Mapt, Rheb) in 5xFAD PV-INs (Fig. 8B).

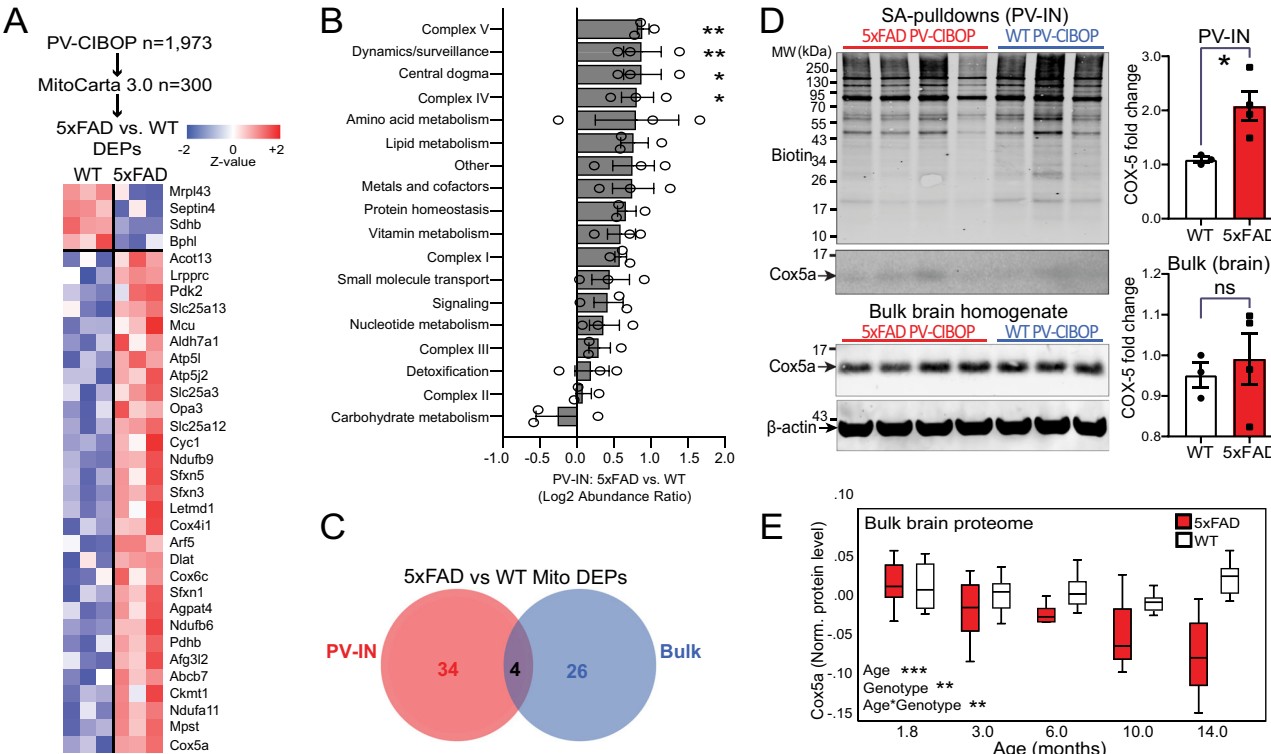

**Fig. 7 | Distinct mitochondrial alterations in PV-INs at early stages of Aβ pathology. A** Heatmap representation of mitochondria-localized proteins that were also identified as DEPs comparing 5xFAD to WT PV-IN proteomes. **B** Differential abundance analysis of distinct mitochondrial functional groups, comparing 5xFAD to WT PV-IN proteomes. The 300 mitochondrial proteins identified in PV-INs were categorized based on known functional and localization-related annotations (from MitoCarta 3.0). Protein levels were normalized and them group-wise abundances were estimated and compared across WT and 5xFAD genotypes ($n = 3$ (WT), 4 (5xFAD)) *$p < 0.05$, **$p < 0.01$ for unpaired two-tailed T-test, error bars represent SEM. **C** Venn diagram of mitochondrial proteins that were identified as DEPs in either PV-IN proteomes or bulk brain cortical proteomes, comparing 5xFAD and WT mice. Minimal overlap in DEPs were observed, highlighting unique mitochondrial effects of Aβ pathology in PV-INs, not visible at the bulk tissue level. **D** WB verification of increased Cox5a protein levels in PV-INs in

5xFAD as compared to WT mice. SA-enriched pulldowns were independently performed from samples used for LFQ-MS studies. Cox5a protein band intensity was normalized to total biotinylation signal in the SA-enriched pulldowns, and to beta-actin in the bulk brain homogenates, and then compared across genotype (5xFAD vs. WT) ($n = 3$ (WT), 4 (5xFAD); Data shown as mean ± SEM; *p < 0.05, unpaired two-tailed T-test). **E** Cox5a protein levels, quantified by TMT-MS, from an independent set of cortical brain homogenates obtained from WT and 5xFAD mice (from Fig. 4). Using linear regression modeling, age, genotype and age x genotype interaction terms were tested for associations with Cox5a protein levels. As compared to WT brain where Cox5a levels were relatively constant with aging, Cox5a levels in 5xFAD brain showed age-dependent decrease after 6 months of age. This pattern was discordant with increased Cox5a in PV-INs in 5xFAD mice at 3 months. *$p < 0.05$, **$p < 0.01$, ***$p < 0.005$. Source data are provided as a Source Data file.

Interestingly, these differential effects of Aβ pathology were only observed in PV-IN proteomes, and not in the bulk brain proteome with the exception of Rheb, highlighting the specificity of these alterations in PV-INs in early AD pathology (Fig. 8C).

Based on the ability of CIBOP to biotinylate signaling proteins in PV-INs, we performed adapted Luminex assays to detect MAPK (Erk, P38 Mapk and Jnk) and Akt/mTOR signaling phospho-proteins, specifically derived from PV neurons[8]. In this approach, the biotinylated phospho-protein is immobilized on beads using capture antibodies, and then their biotinylation status is detected by streptavidin-fluorophore, to directly measure PV-IN-derived phospho-proteins from brain homogenates (Fig. 8D, Supplementary Data 5). We found that mTOR signaling (via phosphorylation of mTOR and down-stream target p70 S6K), was decreased in 5xFAD PV-INs while MAPK pathway activation was not altered (Fig. 8E). This pattern of decreased mTOR signaling was consistent with lower levels of Rheb (a direct activator of mTOR-C1 function), higher levels of RhoA (a known inhibitor of Rheb function) and lower levels of Eif4b (involved in translation initiation) in 5xFAD PV-INs (Fig. 8F)[106]. Collectively, our MS PV-IN CIBOP and adapted Luminex analyses indicate decreased mTOR-C1 activity in 5xFAD PV-INs.

To assess functional relevance of decreased mTOR-C1 in 5xFAD PV-INs, we assessed three composite measures of mTOR-C1 signaling,

including autophagy (mTOR-C1 inhibits autophagy)[105], translational efficiency/protein degradation (mTOR-C1 increases translational efficiency and decreases protein degradation)[107,108] and synaptic plasticity (mTOR-C1 facilitates synaptic plasticity) (Fig. 8F)[104]. 23 GO-annotated positive regulators of autophagy (GO-0010508) were labeled in PV-INs and collectively, this group showed increased levels in 5xFAD PV-INs (Fig. 8G). Western blot analyses of biotinylated proteins from WT and 5xFAD PV-CIBOP mice found increased LC3-II (relative to LC3-I) in 5xFAD PV-INs as compared to WT PV-INs, consistent with increased autophagy in 5xFAD PV-INs (Fig. 8H). To determine whether decreased mTOR-C1 signaling in 5xFAD PV-INs impacts translational efficiency and/or increased protein degradation, we assessed the relative abundances of long-lived and short-lived proteins in 5xFAD and WT PV-IN proteomes. Using a reference dataset of protein half-life estimates derived from in vivo isotopic labeling studies in adult mice[48], we found that DEPs that were increased in 5xFAD PV-INs were biased towards longer-lived proteins, as compared to non-DEPs as well as compared to DEPs with decreased levels in 5xFAD PV-INs (Fig. 8I, Supplementary Data 6). This molecular footprint of relatively-increased abundance of longer-lived proteins in 5xFAD PV-INs is consistent with decreased translational efficiency and/or increased protein degradation in 5xFAD PV-INs. Lastly, we found that proteins involved in regulation of synaptic plasticity (GO-0048167, n = 102 proteins) were also decreased

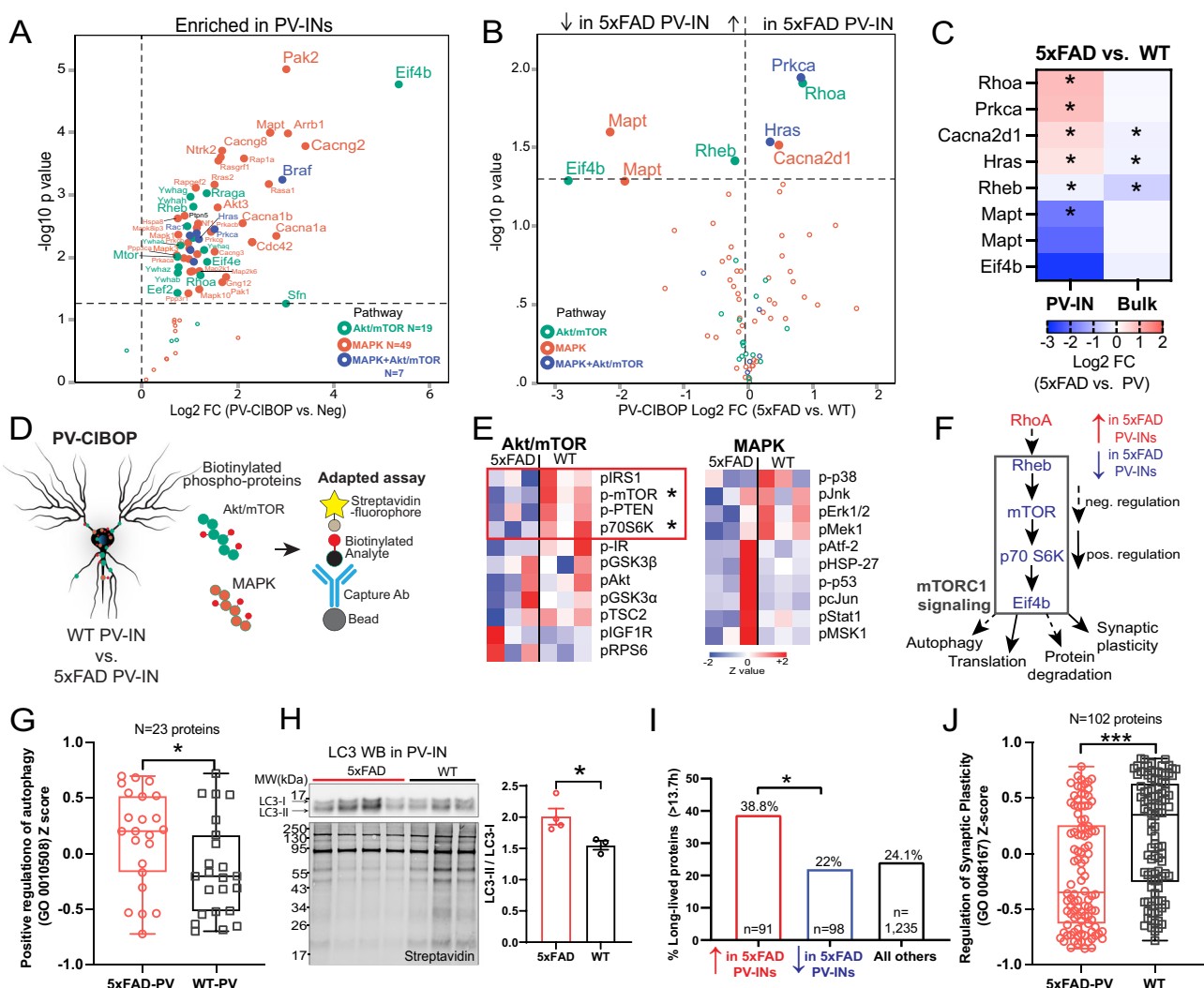

**Fig. 8 | PV-IN-specific decrease in mTOR signaling in early Aβ pathology. A.** Akt/mTOR and/or MAPK signaling proteins biotinylated in PV-IN CIBOP proteomes (as compared to non-CIBOP mice). **B.** Akt/mTOR and MAPK proteins identified as DEPs comparing 5xFAD to WT PV-IN proteomes. **C.** Heatmap representation of Akt/mTOR and MAPK DEPs in PV-IN proteomes and their corresponding bulk brain proteomes (*p < 0.05, two-tailed unpaired T-test). **D.** Cartoon representation of adapted Luminex immunoassay to measure levels of PV-IN-derived phospho-proteins belonging to Akt/mTOR and MAPK proteins from bulk tissue. **E.** Heatmap visualization of Akt/mTOR and MAPK phospho-proteins in PV-INs measured by adapted Luminex assay from WT and 5xFAD mice (n = 3 mice/group, p < 0.05 unpaired two-tailed T-test). **F.** Summary: Decreased activity in mTOR signaling in 5xFAD PV-INs as compared to WT PV-INs, based on total protein levels estimated by PV-CIBOP MS, and phospho-protein levels by the adapted Luminex approaches. **G.** Comparison of proteins that positively regulate autophagy (GO:0010508), in 5xFAD and WT PV-IN proteomes (protein levels were normalized, z-transformed and then group-averaged across biological replicates before group comparisons;

unpaired two-tailed T-test,*p < 0.05). **H.** Top: WB of PV-IN (SA-enriched) samples from 5xFAD and WT PV-CIBOP brain. LC3-II/I ratio was compared across the two groups. Bottom: Biotinylated protein from samples corresponding to WB images above. Data are displayed as mean values +/- SEM. (n = 3 (WT), 4 (5xFAD), *p < 0.05, independent two-tailed T-test). **I.** Analysis of DEPs (5xFAD vs. WT PV-IN proteomes) based on published protein half-lives in mouse brain. Proteins with increased levels in 5xFAD PV-INs were skewed towards proteins with longer half-lives (>13.7 days which represents the 75th percentile of protein half-lives in brain). This pattern is consistent with decreased translational efficiency and/or increased protein degradation, which would disproportionately impact the relative abundances of short-lived proteins. **J.** Comparison of proteins that regulate synaptic plasticity (GO:0048167) as a group, in 5xFAD and WT PV-IN proteomes (levels of 102 proteins were normalized, z-transformed and then averaged across biological replicates before group comparisons using unpaired two-tailed T-test (***p < 0.005). See Supplementary Data 6 for related analyses. Source data are provided as a Source Data file.

in 5xFAD PV-IN proteomes (Fig. 8J), consistent with observed synaptic protein changes and physiological defects presented earlier (Figs. 5,6). These analyses provide congruent lines of evidence for decreased mTOR signaling at various levels of the signaling axis (upstream and downstream of mTOR-C1 function), that are associated with increased autophagic flux, decreased translational efficiency and decreased synaptic plasticity in PV-INs at early stages of Aβ pathology.

## Discussion

We describe a versatile intersectional method[8,10,16] allowing quantitative in vivo neuron-type-specific proteomics. We leveraged this

approach to isolate the native-state PV-IN proteome, and examine changes in this cell type in an early-stage mouse model of Aβ pathology. In principle, our approaches are readily adaptable to other neuron types, due to the recent expansion of cis-element-directed AAV approaches[109,110] with unparalleled cell-type-specificity. A secondary advantage to our workflow is the ability to seamlessly translate the technique across different mouse models of disease, without the need for complex and expensive cross-breeding strategies.

Comprehensive molecular characterization of neuronal subtypes can provide critical insights into the function and modifiable pathogenic mechanisms of neurological diseases. Ideally, this should be

performed at the proteomic level, while retaining the native state of neuronal proteome (i.e., protein samples from intact soma, dendrites, and axons), overcoming limitations of transcriptomic studies which rely on isolation of intact neurons or neuronal nuclei. Despite the modest concordance between mRNA and functionally-relevant protein particularly in neurons[3–5,111], proteomic studies of neurons and neuronal subtypes in the in vivo context have lagged behind transcriptomic advances due to several technical barriers. However, recent advances in cell type-specific in vivo proteomic labeling approaches, involving proximity-dependent biotinylation by biotin ligases included TurboID (CIBOP), and bio-orthogonal amino acid tagging approaches[112–114], provide exciting opportunities to investigate neuronal subtype-specific molecular signatures and disease mechanisms. For the CIBOP approach described in this study, proximity-based biotinylation of proteins in the cytosol and several cellular compartments (e.g., synaptic boutons or postsynaptic densities) can be achieved by TurboID expression selectively in brain cell types of interest, by driving Cre recombinase expression via transgenic or AAV approaches, in the Rosa26$^{TurboID}$ mouse line[8]. CIBOP with TurboID mice has thus far been successfully applied in specific brain cell types by crossing with cell-selective Cre mouse lines (Camk2a-Cre for excitatory neurons and Aldh1l1-Cre for astrocytes) and is thus well positioned to be extended to other neuronal subtypes and glial cells[8,114,115]. Universal extension of cell-specific CIBOP across mouse models of disease would provide unparalleled resolution to pathological mechanisms. However, the need for further time-consuming and expensive cross-breeding represents a significant barrier. Thus, we developed an enhancer-AAV method to deploy CIBOP more rapidly in PV-INs of both WT and disease model mice. Further development of intersectional AAV approaches should also allow CIBOP to be expanded in other model species as well as to cell types outside the brain[116].

Among neuronal subtypes, PV-INs represent a unique class of inhibitory interneurons with fast-firing properties and high metabolic activity[13,98]. Selective dysfunction of PV-INs contributes to a variety of neurological insults, including in neurodegenerative diseases such as AD, neurodevelopmental disorders, and catastrophic early-life epilepsy[117–119]. We applied CIBOP to PV-INs throughout the forebrain by subcloning a Cre-expressing cassette into a PV-specific enhancer-AAV[10] (AAV.E2.Cre) delivered systemically using the PHP.eB serotype, which readily enters the brain[16]. Delivery of AAV.E2.Cre in Rosa26$^{TurboID}$ mice lead to PV-specific proteomic biotinylation. When coupled with MS of affinity-purified biotinylated proteins, we obtained the PV-IN-specific proteome while retaining their native functional state without need for cell isolation and without physiological disruption. Taken together, the PV-CIBOP approach identified unique proteomic signatures of PV-INs complementing existing transcriptomic data, serving as an important resource to the neuroscience research community.

The PV-IN proteomic signature includes hundreds of proteins that are either exclusive to, or highly-enriched in PV-INs over the bulk brain proteome, including canonical PV-IN markers (e.g. Kv3.1-3.3, Erbb4, Ank1, Syt2)[12] as well as 200 proteins not predicted by PV-IN scRNAseq databases (eg. Tnpo3, Htt, Synrg, Cplx3, Mtor, Gria1). PV-CIBOP also labeled proteins with predicted subcellular targets in the cell body, axons and dendrites of PV-INs, including pre- and post-synaptic compartments. The PV-IN proteome was suggestive of high metabolic activity, protein translation, signaling and vesicle functions, consistent with the fast-firing and rapid neurotransmission[15] properties of these cells. The PV-CIBOP proteome was also highly-enriched in proteins with causal associations to neurodegenerative disease risk, including AD (BIN1)[120], pure tauopathies (MAPT)[121], and synucleinopathies (SNCA, SNCB)[122,123]. Several lines of evidence now suggest that interneuron dysfunction leads to altered circuit excitability in early-stage models of APP/Aβ pathology[124,125]. Stx1b and Gat1, among others, are examples of proteins enriched in the PV-IN dataset linked to both epileptiform activity and AD[126]. Furthermore, other neurodevelopmental and autism implicated molecules (Shank2, Syngap1, ErbB4)[127–131] were also highly enriched in PV-INs. When contrasted to CIBOP-derived proteomes from the (predominantly pyramidal neuron) Camk2a-Cre mice, PV-IN proteomes exhibited clearly higher levels of ribosomal, endocytic, Akt/mTOR signaling, synaptic cytoskeletal, endocytic and synaptic vesicle-related proteins. The proteomic contrast between PV-IN and Camk2a neurons in mice, integrated with recently-identified proteomic correlates of cognitive resilience in humans, revealed a disproportionate enrichment of pro-resilience proteins in the PV-IN proteome, suggesting a link between cognitive resilience and PV-INs.

To look for evidence of PV-IN-associated vulnerability in human AD, we assessed existing human post-mortem bulk brain proteomic data from controls and AD cases (with and without dementia prior to death)[70]. We identified a module of co-expressed proteins (M33) that was enriched in PV-IN markers (e.g. Pvalb, Kcnc2), that was distinct from modules enriched in excitatory neuronal (pyramidal cell) markers. The PV-IN proteomic module was less abundant in AD cases, compared to both asymptomatic AD and control patients, and this decrease in M33 was also correlated with AD neuropathological features (Aβ pathology, neurofibrillary tangles), severity of cognitive dysfunction, and rate of cognitive decline although independent of APOE genetic risk. After accounting for neuropathological hallmarks of neurodegenerative diseases, M33 still remained associated with cognitive resilience, suggesting that links between PV-INs and cognitive resilience may be independent of neurodegenerative pathology. Using a complementary approach to estimate abundances of different neuronal classes, we found that PV-IN cell type abundance was strongly associated with cognitive resiliency in longitudinal studies of aging. Our findings suggest that preservation of PV-IN function in the brain may be generally protective in AD, and link PV-INs with cognitive resilience and vulnerability in AD.

To directly capture longitudinal impacts of aging and disease progression, we analyzed mouse bulk brain proteomes from WT and 5xFAD mice spanning a wide age range and found that PV-IN proteins (but not excitatory neuronal proteins) showed unique age-dependent increases in expression although this was suppressed in 5xFAD mice, starting as early as 3 months of age. Albeit a snapshot from bulk tissue, this suggests that of PV-IN protein levels, but not other neuronal markers, may change in early stages of APP/Aβ pathology. Somatostatin (Sst), a protein primarily expressed in cortical dendrite-targeting (non-PV fast spiking)[132,133] inhibitory interneurons, was similarly reduced starting at 3 months of age. As both Pvalb and Sst expression are linked to the level of circuit activity[134,135] these changes may reflect a differential dysregulation of interneuron activity levels at a stage where substantial plaque formations are just arising in 5xFAD mice. At the histological level, no measurable differences in PV-IN density were observed between 3 month-old WT and 5xFAD mice, arguing against early overall cell loss of PV-INs at this early stage, but rather suggesting changes to their proteomic profile.

To evaluate PV-IN proteomic changes in response to early Aβ pathology, we compared PV-CIBOP-derived proteomes from 3 month-old Rosa26$^{TurboID}$ WT and 5xFAD mice. Over 450 DEPs were found. Proteins involved in mitochondrial function, cholesterol biosynthesis (e.g., Dhcr7), and metabolism were generally increased in PV-INs. In contrast, cytoskeletal, structural, and synapse-associated proteins were generally decreased in PV-INs. Surprisingly, alterations found in the PV-IN proteome were almost completely non-overlapping with those changes resolved from bulk brain. Since the majority of intra-neuronal APP/Aβ was detected in non-PV-INs at this stage of pathology, the observed changes in PV-INs are most likely due to Aβ from other non-PV neurons rather than dysfunctional Aβ processing in PV-INs. Based on these specific effects of Aβ pathology on PV-INs, extending CIBOP to other interneuron and excitatory neuron subclasses, and capturing the effects of brain region and age in future studies will be

necessary to resolve whether PV-IN protein levels are profoundly affected in early AD models, or rather, are part of a continuum of emerging cell-type autonomous alterations across different brain regions.

Initial PV-CIBOP studies in WT mice found substantial enrichment of proteins encoded by MAGMA-identified AD genetic risk factors, as well of pro-resilience proteins in the PV-IN proteome in contrast to Camk2a neurons[54,55,70]. Therefore, we asked whether MAGMA AD proteins would also be disrupted in our early AD model PV proteome, and indeed, cross-referenced DEPs in 5xFAD matched with 20 MAGMA AD genes. Furthermore, proteins associated with cognitive resilience were systematically reduced in 5xFAD PV-INs, particularly proteins involved in presynaptic vesicle fusion/exocytosis/release (Cplx1, Cplx2, Stx1b, Elfn1, Rab3c, Rims1)[55]. To examine the functional implications of this pre-synaptic signature, we used PV-IN-specific optogenetic approaches in two independent models of early APP/Aβ pathology. At cortical PV-to-pyramidal synapses, both studies clearly point to disturbances in presynaptic function. In particular, changes in vesicular release probability appear likely. Beyond pre-synaptic dysfunction, several studies have also shown an emergence of inhibitory post-synaptic dysfunction across a number of APP/Aβ models[136–139]. Future mechanistic investigations are warranted to examine the roles of resiliency-related pre-synaptic and post-synaptic proteins in PV-INs and other inhibitory cell synaptic mechanisms in AD models, for example, whether their levels shift in response to direct interactions with Aβ or rather change in response to other cellular or circuit stressors. Importantly, these early alterations identified at PV synapses may represent opportunities for early therapeutic intervention.

Despite minimal plaque burden in 3 month-old 5xFAD mice[38], the significant shifts in the 5xFAD PV-IN proteome may represent a homeostatic response to prior changes in neuronal and circuit behavior and organization[100,140,141] known to occur in young, pre-plaque APP/Aβ models, including in younger (<3 month-old) 5xFAD mice[33,125,142,143]. Relatedly, a signature of circuit instability is also present in human patients with mild cognitive impairment and AD[23,35,144]. To compensate for this early circuit dysfunction, PV-INs are well-suited to homeostatically respond[145], but this process could impose a higher metabolic demand to sustain this compensation. Indeed, mitochondrial impairments have been observed prior to extensive pathology in APP/Aβ model mice[146,147]. In our PV-CIBOP proteomes, we found a signature of stress-responsive proteins (Armt1, Rhob, Gstm1, RhoA, Tmco1, Akr1b3, Gcn1, Hras, Cul3, Pdk2, Rap2a, Flot1) in 5xFAD as compared to WT. Of note, RhoA activation increases Aβ and tau pathology and co-localizes with NFTs in human brain[148,149]. In contrast to the overall synaptic effects of early AD pathology in PV-INs, we observed a marked increase in mitochondrial and metabolic proteins in PV-INs. This increase could be reflective of a protective, compensatory (via increased mitochondrial biogenesis to sustain higher metabolic demand), or on the other hand, via direct interactions with Aβ. Other potentially compensatory changes observed in 5xFAD PV-INs included increased Dhcr7 for de-novo cholesterol biosynthesis in neurons, increased Apeh to process Aβ oligomers along with increased autophagy as supported by increased levels of positive regulators of autophagy and increased lipidated form of LC3 (LC3-II). Conversely, a detrimental/dysfunctional response (e.g., accumulation of dysfunction mitochondria) is also possible. We noted that mitochondrial functional proteins and Complex I, III, IV, V proteins were selectively increased in 5xFAD PV-INs while a smaller group of mitochondrial structural, dogma, and Complex II proteins were not. Therefore, follow-up studies focusing on mitochondrial structure and function specifically in PV-INs are warranted to better understand the basis and consequences of these mitochondrial alterations. Taken together, the molecular phenotype of 5xFAD PV-INs is indicative of a significant cellular stress response occurring in 3 month old PV-INs, comprising both compensatory and maladaptive events, which is not evident in the bulk proteome at this age. Furthermore, we present several lines of evidence from bulk brain and PV-IN-specific experiments, and human brain proteomic analyses, that PV-IN proteomic signatures and cognitive resiliency are linked. Therefore, understanding the mechanisms for this compensation could provide therapeutic insights for future studies.

Metabolic shifts and mitochondrial biosynthesis are regulated by signaling pathways such as Akt/mTOR and MAPK[102–105]. We also observed high levels of proteins involved in both Akt/mTOR (e.g. Mtor, Eif4b) and MAPK (e.g. Erk and Mek proteins) signaling pathways in PV-INs. Therefore, we hypothesized that mTOR signaling may be altered in 5xFAD PV-INs. We directly measured biotinylated phospho-proteins indicative of levels of activity of these pathways specifically in PV-INs by leveraging an adapted Luminex immunoassay method recently validated for CIBOP-based studies[8]. Our MS-based and Luminex-based analyses provide evidence of decreased mTOR (mTOR-C1) signaling in PV-INs, but not in bulk brain tissue, that appear to augment autophagic flux, decrease translational efficiency or increased protein degradation, and impair synaptic plasticity. Collectively, our results indicate early dysregulation of mTOR signaling in PV-INs as a potential upstream mechanism for mitochondrial and metabolic alterations as well as synaptic dysfunction occurring selectively in PV-INs in early stages of AD pathology in 5xFAD mice.

Limitations of our work relate to technical considerations of both AAV-based PV-IN targeting, and potential proteomic biases of the CIBOP approach. Currently, CIBOP leads to cell type-specific expression of TurboID-NES, which contains a nuclear export sequence for preferential proteomic labeling outside the nucleus. This may bias the PV-IN proteome away from nuclear proteins as well as from proteins present within the lumen of organelles (e.g. ER/Golgi, mitochondria, lysosomes)[8,115]. Whether removal of the NES impacts the nature of the PV-IN proteome, remains to be determined. Another consideration is that our AAV.E2.Cre strategy targets PV-INs as a whole, although several PV-IN subtypes have been identified by scRNAseq studies[1] (i.e., chandelier cells and several basket cell PV types). It is therefore possible that the proteomic signatures of these different PV-IN subtypes are non-uniform, for example, of those residing in different cortical layers. Thus, our initial PV-CIBOP derived proteome may not accurately describe the proteomic granularity which may further exist within PV-IN interneuron classes. Further studies with increased PV subtype-specificity or physiological and morphological studies using CRISPR or related methods to examine individual proteins may be useful in this regard. While we contrasted PV-IN proteomes against Camk2a-CIBOP proteomes in WT mice, the PV-CIBOP approach used AAV while the Camk2a-CIBOP strategy used transgenic crosses to achieve cell type-specific labeling. Given the immense diversity within excitatory neurons[2] and the low-level expression of Camk2a in non-excitatory neurons[56,57], future studies should apply neuronal class-specific approaches in mouse models of AD and tau pathology to examine cellular proteomic alterations occurring in AD pathology. A key finding of our work is that PV-IN-specific changes, revealed by CIBOP, were not apparent at the level of the bulk brain proteome. We suspect that the reason for this disagreement is likely because PV-INs are a small fraction of brain cells, therefore, their proteomic signatures and proteome-level changes in AD pathology are not captured at the bulk tissue level. Also, majority of PV-IN proteins are not exclusive to PV-INs, although AD pathology may exert cell type-specific effects on these common neuronal proteins as well, further limiting the interpretation of bulk brain proteomics data. Another contributor to this disagreement is that the inherent biases of the CIBOP approach may over- or under-sample specific proteomic changes, exaggerating discrepancies with the bulk proteome. To address this potential methodological confound, future studies should consider using the CIBOP approach simultaneously to in a pan-cellular manner or in the context of a different neuronal class.

In summary, our integrative PV-CIBOP approach revealed a native-state proteomic signature for a single, highly-important interneuron class in the mouse brain. Comparison of PV-CIBOP proteomic signatures with human post-mortem data suggests selective synaptic and metabolic PV-IN vulnerabilities in early AD pathogenesis that may be linked to cognitive dysfunction. These findings provide a strong rationale to investigate and target early proteomic changes occurring in PV-INs and other inhibitory neuron types in mouse models of AD and other neurological diseases.

## Methods

### Ethics approval and inclusion to participate

All experiments involving animal use were performed in accordance with the ethical guidelines of Emory University Institutional Animal Care and Use Committee (IACUC, PROTO201700821) and ARRIVE guidelines.

### Reagents

A detailed list of reagents used in these studies, including antibodies, is provided in Supplementary Data 7.

### Animals

C57BL/6 J (wildtype[WT]) mice (JAX #000664), Rosa26-TurboID (C57BL/6-Gt(ROSA)26Sortm1(birA)Srgj/J, JAX #037890)[8], Camk2a-Cre-ert2 (B6;129S6-Tg(Camk2a-cre/ERT2)1Aibs/J, JAX #012362)[150] and 5xFAD (B6.Cg-Tg(APPSwFlLon,PSEN1*M146L*L286V)6799Vas/Mmjax, JAX #034848)[19,38] mouse lines were used for experiments in this study and genotyping was performed using primers and polymerase chain reaction (PCR) conditions listed on the vendor website (Jackson labs). All animals were maintained on the C57BL6/L background, following at least 10 serial backcrosses if originally derived from a different or mixed background. Male and female mice were used for all experiments with data collected from ≥ 3 mice per experimental condition for all experiments. Animals were housed in the Department of Animal Resources at Emory University under a 12 h light/12 h dark cycle with ad libitum access to food and water and kept under environmentally controlled and pathogen-free conditions. All experiments involving animal procedures were approved by the Emory University Institutional Animal Care and Use Committee (IACUC, PROTO201700821) and were in accordance with the ARRIVE guidelines.

### Retro-Orbital AAV injections

The same AAV retro-orbital injection was given to male and female mice of each genotype as previously described[16]. Rosa26$^{TurboID/wt}$ and WT control mice were briefly (~2 min) anesthetized with 1.5–2% isoflurane to perform the injection. AAV(PHP.eB).E2.Cre.2 A.GFP virus (2.4E + 11 vector genomes) and AAV(PHP.eB).Flex.Tdtom (3.15E + 11 vector genomes) were co-injected (final volume; 65 µl in sterile saline) to target and label PV interneurons throughout the cortex. Injections into the retro-orbital sinus of the left eye were performed with a 31 G x 5/16 TW (0.25 mm) needle using an insulin syringe. Fresh syringes were used for each mouse. Mice were kept on a heating pad for the duration of the procedure until recovery (< 5 min) and then returned to their home cage. After 3 weeks post-injection, mice were provided with biotin water continuously. Biotin water was administrated for 2 weeks until acute slice sample collection (total of 5 weeks post-RO injection).

### Tissue collection

Mice were fully anesthetized with isoflurane and euthanized by decapitation. Mice brains were immediately removed by dissection. The right hemisphere of the brain was snap frozen on dry ice either intact or after regional dissection (cerebellum, brain stem, cortex, hippocampus, and striatum/thalamus) for downstream proteomics analysis while left hemisphere were used for electrophysiological recording and immunostaining.

### Acute Slice Preparation

Brains were immediately removed by dissection in ice-cold cutting solution (in mM) 87 NaCl, 25 NaHO3, 2.5 KCl, 1.25 NaH2PO4, 7 MgCl2, 0.5 CaCl2, 10 glucose, and 7 sucrose. The left hemisphere of the brain was used for electrophysiological recording and immunostaining. For electrophysiology, brain slices (300 µm) were immediately prepared in the coronal or sagittal plane using a vibrating blade microtome (VT1200S, Leica Biosystems) in the same solution. Slices were transferred to an incubation chamber and maintained at 34 °C for ~30 min and then 23-24 °C thereafter for patch clamp recordings. Solutions were equilibrated and maintained with carbogen gas (95% O2/5% CO2) throughout the entire day. Remaining tissue from the left hemisphere was transferred to 4% paraformaldehyde (PFA) in 0.1 M phosphate-buffered saline (PBS)PFA in PBS at 4 °C for immuno-histological analysis.

### CIBOP studies

PV-CIBOP studies in WT mice were performed by single retro-orbital injections of AAV (AAV(PHP.eB).E2.Cre.2 A.GFP) to Rosa26$^{TurboID/wt}$ mice. As our previous studies including controls herein used the PHP.eB serotype, this was utilized throughout for E2 enhancer-AAV experiments. Rosa26$^{TurboID/wt}$ (PV-CIBOP) were also crossed with 5xFAD (hemizygous) to derive 5xFAD (hemi)/Rosa26$^{TurboID/wt}$ (5xFAD/PV-CIBOP) and littermate WT/PV-CIBOP animals. Camk2a-CIBOP experiments were performed in Camk2a-Cre-ert2$^{het}$/Rosa26$^{TurboID/wt}$ mice. Tamoxifen was injected (intraperitoneally(i.p.), 75 mg/kg/dose in corn oil x for 5 consecutive days), followed by 3 weeks to allow Cre-mediated recombination and TurboID expression after which biotinylation (37.5 mg/L in drinking water) was performed for 2 weeks[8]. In the case of PV-CIBOP (on WT or 5xFAD backgrounds), AAV injections were followed by 3 weeks to allow recombination after which biotinylation was performed as above. For all CIBOP studies, control animals included Cre-only (in the case of Camk2a-CIBOP experiments) or Rosa26$^{TurboID/wt}$ mice (for PV-CIBOP experiments). Control groups in PV-CIBOP studies also received AAV E2.Cre injections for fair comparisons. In Camk2a-CIBOP studies, all experimental groups received tamoxifen to account for tamoxifen mediated effects. No tamoxifen was needed for PV-CIBOP experiments (non-inducible Cre). Recombination period (after inducing Cre via tamoxifen or delivery of AAV E2.Cre) and biotinylation period after recombination were kept constant across all studies. CIBOP studies were completed at 12 to 13 weeks of age.

### Electrophysiology

For whole-cell current clamp recordings, acute slices were continuously perfused in a recording chamber (Warner Instruments) with (in mM) 128 NaCl, 26.2 NaHO3, 2.5 KCl, 1 NaH2PO4, 1.5 CaCl2, 1.5 MgCl2 and 11 glucose, maintained at 30.0 ± 0.5 °C. The recording solutions were equilibrated and maintained with carbogen gas (95% O2/5% CO2) in a closed loop system for the entire experiment. PV neurons were targeted for whole-cell recordings in layer 5 somatosensory cortex as previously described using combined gradient-contrast video microscopy and epifluorescent illumination either custom-built or commercial (Olympus) upright microscopes[33]. Although our AAV expressed a GFP downstream of a 2 A linker (Fig. 1), GFP fluorescence was too dim, hence co-injection with a floxed tdTom for fluorescence-targeted patch-clamp of PV interneurons. Recordings were obtained using Multiclamp 700B amplifiers (Molecular Devices). Signals were low pass filtered at 4–10 kHz and sampled at 50 kHz with the Digidata 1440B digitizer (Molecular Devices). Borosilicate patch pipettes (World Precision Instruments) were filled with an intracellular solution containing (in mM) 124 potassium gluconate, 2 KCl, 9 HEPES, 4 MgCl2, 4 NaATP, 3 l-ascorbic acid, and 0.5 NaGTP. Pipette capacitance was neutralized in all recordings and electrode series resistance compensated using bridge balance in current clamp. Liquid junction potentials were uncorrected.

For PV interneuron current clamp experiments, membrane potentials were maintained at −70 mV using a constant current bias. AP trains were then initiated by somatic current injection (300 ms) normalized to the cellular capacitance in each recording measured immediately in voltage clamp after breakthrough[33,151]. For quantification of individual AP parameters, the first AP in a spike train was analyzed for all cells. Passive properties were determined by averaging the responses of several 100 ms long, −20 pA steps during each recording.

For voltage clamp recordings, cells were filled with an intracellular solution containing (in mM) 120 CsMeSO4, 10 HEPES, 5 TEA.Cl, 4 Na2ATP, 0.5 Na2GTP, 2 MgCl2, 10 L-Ascorbic Acid, and 3 Qx314. Spontaneous excitatory and inhibitory postsynaptic currents (sEPSCs and sIPSCs) were recorded at a holding voltage of −70 and 0 mV interleaved for one second each for 3-5 min. Signals were filtered with a Bessel 10 kHz low-pass filter and sampled at 50 kHz. All recordings had a series resistance of <20 MΩ. Event detection was carried out using Clampfit (Molecular Devices) using a template matching algorithm and were manually curated following posthoc 4 kHz low-pass filtering. Events were compared against the template within a match threshold of 2.5. Events with a rise time of <0.15 ms and a decay time less than double the rise time were excluded. All statistical tests were completed in Prism (GraphPad) by using unpaired t-tests for cell averages, and p < 0.05 was considered statistically significant. All group statistics are presented as cumulative frequency distribution or mean ± SEM (standard error of mean).

For optogenetic slice experiments, C1V1 was activated using an unfiltered amber LED (M590L3; Thorlabs) centered on $l = 596$ nm ($\pm 15$ nm). ChETA was excited using blue light from an unfiltered LED (M470L3; Thorlabs) centered on $l = 461$ nm ($\pm 20$ nm). LEDs were rapidly modulated with time-locked TTL pulses from the electrophysiology software using short pulses with a current controller (LEDD1B; Thorlabs). Due to kinetics differences between ChETA and C1V1, 0.15-0.3 and 4 ms pulses (respectively) were found to be optimal to reliably elicit PV-pyramidal cell IPSCs every trial with minimal jitter. To eliminate potential sources of variation between experiments, these parameters and the amber/blue light power at the objective remained unaltered for all optogenetic control and test experiments.

## Stereotactic surgery for optogenetics

PV-IN-specific optogenetic studies were performed in separate cohorts of WT and 5xFAD mice, or in PV-Cre (Jax strain #:017320) mice for hAPP-AAV studies. For C1V1 experiments in 5xFAD (or WT controls), AAV(PHP.eB)-E2-C1V1-eYFP (addgene 135633) was co-injected with AAV1.CamKII(1.3).eYFP.WPRE.hGH (addgene 105622) at a 1:1 ratio. For AAV1-Ef1a-DIO-ChETA-EYFP experiments (addgene 26968), co-injections of an hAPP-expressing virus AAV(PHP.eB).EF1a.hAPP.oPRE (hAPP RefSeq NM_000484.4) or an equivalent amount of sterile saline in controls were performed. In another subset of experiments, AAV(PHP.eB)-E2-ChETA-eYFP was utilized. Injections were all performed in the SBFI region of S1 cortex. For injections, mice were head-fixed in a stereotactic platform under continuous isoflurane anesthesia (1.8–2.0%). Thermoregulation was provided by a heat plate with a rectal thermocouple for biofeedback, to maintain core body temperature near 37 °C. A small incision was made and a craniotomy cut in the skull (<0.5 μm in diameter) to allow access for a glass microinjection pipette. Coordinates (in mm from Bregma) for microinjection were $X = \pm 3.10$–3.50; Y = −2.1; α = 0°; Z = 0.85–0.95. Viral solutions (stock titers were ~1.0 × 10^13 vg/mL, except for AAV.EF1a.hAPP, which was 1.0 × 10^12 vg/ml) were injected slowly (~0.02 μL min⁻¹) using Picospritzer-directed short pulses (~0.3 μL total). After ejection of virus, the micropipette was held in place (~5 min) before withdrawal. The scalp was closed first with surgical sutures and Vetbond (3 M) tissue adhesive thereafter and the animal was allowed to recover under analgesia (carprofen and buprenorphine SR). After allowing for onset

of expression (1 or 3 weeks for C1V1/YFP or ChETA/hAPP, respectively), animals were sacrificed, and acute slices harvested for patch clamp studies as detailed above.

**Flow Cytometry.** Flow cytometry was performed to examine the efficiency of AAV(PHP.eB).E2-Cre virus targeting efficiency across WT and 5X Fad mice. In brief, mice (Turboid WT, Turbo id 5XFad, and 3 negative controls; $n = 3$ each) were anesthetized in a glass induction chamber with isoflurane until the mouse stopped breathing and showed no sensory response to tail pinch. Mice were decapitated and brain were dissected in ice-cold cutting solution (in mM) 87 NaCl, 25 NaHO3, 2.5 KCl, 1.25 NaH2PO4, 7 MgCl2, 0.5 CaCl2, 10 glucose, and 7 sucrose. 250 μm acute slices of the brain were obtained and then placed into a cutting solution with 0.5 mg/mL protease (P5147–100MG, Sigma-Aldrich) for 60 min with continuous carbogen gas bubbling. Tissue samples were then manually triturated in 300 μL of 1% PBS into a single-cell suspension. Cells were first fixed in 200 μL of 1x fixation buffer (00822249; eBioscience) for 30 min on ice, then washed with 1 ml of 1X PBS three times. Cells were then permeabilized with 200 μL permeabilization buffer (eBioscience; cat# 00-8333-56) for 30 min on ice. To determine the proportion of TdTomato positive cells within neurons, fixed and permeabilized cells were incubated with anti-NeuN (ABN90; Biolegend) and anti-RFP/TdTomato (600401379; Biolegend) antibody at dilution 1:250 for 1 h on ice. Cells were then washed with 1 mL of permeabilization buffer 3 times and incubated for 30 min with a secondary antibody anti-rabbit AF 594 (A21207; Thermofisher) and anti-guinea pig AF488 (A11073; Thermofisher) at dilution 1:500. Cells were finally washed 3 times with 1X permeabilization buffer, as mentioned above. After the last wash, 250 μL of PBS was added, vortexed and kept on ice in the dark until flow cytometry was performed. Compensation was performed prior to the experiment using single fluorophore labelled OneComp eBeads™ (984 01-1111-41; Thermofisher). OneComp eBeads (50-112-9031; Thermofisher) stained with appropriate fluorophore-conjugated antibodies and unstained beads were used for compensation controls. Data were collected on a BD LSR II flow cytometer. Flow data were further analyzed using FlowJo v10. After gating for single mononuclear cells (FSC and SSC), events were sub-gated for NeuN positive neurons and finally, proportion of TdTomato positive cells among NeuN+ neurons were estimated, and compared across groups. Of note, the isolation method used is inherently biased towards isolation of interneurons as compared to other neuronal populations.

**Enzyme-linked immunosorbent assay (ELISA).** ELISA was performed to measure Aβ42 levels in WT and 5xFAD brains to determine whether biotinylation impacts the level of Aβ42 pathology in non-turbo 5xFAD and PV-CIBOP/5xFAD animal groups ($n = 3$ each for WT, non-turbo 5x FAD, Turbo-WT, and 4 for Turbo-FAD). The ELISA was performed by using Human Amyloid beta (aa1-42) Quantikine ELISA Kit (DAB142; RnDsystems). In brief the original tissue lysates were diluted 1:50 in PBS and BCA was reperformed for the accurate measurement of the protein across the samples. The concentration was adjusted across the samples and ELISA was performed on 0.3 μg of the protein for each sample in duplicate according to the manufacturer protocol (DAB142; RnDsystems).

## Immunohistochemistry (IHC) and immunofluorescence microscopy

IHC was performed on brain slices obtained from CIBOP studies. Tissue was transferred immediately after euthanizing the animals in 4% PFA or after completion of electrophysiological recordings. Brain tissue/slices were post-fixed in 4% PFA in PBS at 4 °C overnight then subsequently transferred into 30% sucrose solution until cryosectioning. After embedding in optimal cutting temperature (OCT) compound, the slices were further cut coronally or sagittally into

40-μm-thick sections on a cryostat (Leica Biosystems). IHC and immunofluorescence protocols used were as previously published[8]. The tissue sections were transferred into the cryoprotectant or directly mounted on the charged glass slides and stored at −20 °C until use. For immunofluorescence (IF) staining, 40 μm thick free-floating brain sections or mounted on glass slides were washed, blocked and permeabilized by incubating in TBS containing 0.30% Triton X-100 and 5% horse serum for 1 h at room temperature. If desired, then the antigen retrieval was performed in Tris-EDTA buffer (10 mM Tris base, 1 mM EDTA, 0.05% tween-20) pH9 for 30 min at 65 °C before the blocking and permeabilization step. Primary antibodies were diluted in TBS containing 0.30% Triton TX-100 and 1% horse serum. After overnight incubation at 4 °C with primary antibodies, the sections were rinsed 3x in TBS containing 1% horse serum at room temperature for 10 min each. Then, the sections were incubated in the appropriate fluorophore-conjugated secondary antibody at room temperature for 2 h in the dark. The sections were rinsed once and incubated with DAPI (1 μg/ml) for 5 min, washed 3x in TBS for 10 min, dried, and cover slipped with ProLong Diamond Antifade Mountant (P36965; Thermo-Fisher). All the primary and secondary antibody detail including dilution used are listed in Supplementary Data 7. We optimized several existing antibodies to detect Pvalb protein by IHC (Supplementary Data 7). Since the AAV E2.Cre.GFP labels all PV-INs with very high efficiency, cells expressing GFP were used as the reference standard for PV-INs and Pvalb antibodies with highest levels of agreement with GFP-positive PV-INs were used interchangeably for IHC studies, to allow species compatibility of primary antibodies. In addition, the guinea pig Pvalb antibody (195004; Synaptic Systems) preferentially labeled the synapto-dendritic compartment of PV-INs. IHC studies were also performed on experimental animals from CIBOP studies, as well as from non-CIBOP WT and 5xFAD mice using sagittal sections of entire hemispheres.

Images of the same region across all samples were captured as z-stacks using the Keyence BZ-X810, except for images in Supplementary Fig. 5 (Parvalbumin IHC quantification). Some of the z-stacked images of entire brain were stitched together to allow regional comparison based on level of biotinylation. Images for quantification of Parvalbumin staining were obtained with a two-photon laser scanning microscope (2pLSM) using a commercial scan head (Ultima; Bruker Corp) fitted with galvanometer mirrors (Cambridge Technology) using a 60x, 1.0 NA objective. Parvalbumin levels were quantified in an analogous fashion to that described previously[152], but at higher magnification to resolve potential differences across different cortical layers. All image processing was performed either using the Keyence BZ-X810 Analyzer or Image J software (FIJI Version 1.53).

### Tissue processing for protein-based analysis, including Western Blot (WB)

Tissue processing for proteomic studies, including MS studies, were performed similar to the previous CIBOP study[8]. Frozen brain tissues (whole brain homogenate excluding the cerebellum for studies in Fig. 1 and whole cortical samples for studies in Figs. 2 and 5) either intact or dissected cortex, was weighed and added to 1.5 mL Rino tubes (TUBE1R5-S; Next Advance) containing stainless-steel beads (0.9–2 mm in diameter, SSB14B; Next Advance) and six volumes of the tissue weight of urea lysis buffer (8 M urea, 10 mM Tris, 100 mM NaH2PO4, pH 8.5) containing 1X HALT protease inhibitor cocktail without EDTA (78425, ThermoFisher). Tissues were homogenized in a Bullet Blender (Next Advance) twice for 5 min cycles at 4 °C. Tissue were further sonicated consisting of 5 s of active sonication at 20% amplitude with 5 s incubation periods on ice. Homogenates were let sit for 10 min on ice and then centrifuged for 5 min at 13,523-G and the supernatants were transferred to a new tube. Protein concentration was determined by BCA assay using Pierce™ BCA Protein Assay Kit (23225, Thermofisher scientific). For WB analyses, 10 μg of protein

from brain lysates were used to verify TurboID expression (anti-V5) and biotinylation (streptavidin fluorophore conjugate). Standard WB protocols, as previously published, were followed[8].

Other proteins detected by WB also included beta actin, Pvalb and LC3 (see Supplementary Data 7 for antibodies/dilutions). All blots were imaged using Odyssey Infrared Imaging System (LI-COR Biosciences) or by ChemiDoc Imaging System (Bio-Rad) and densitometry was performed using ImageJ software.

### Enrichment of biotinylated proteins from CIBOP brain

As per CIBOP protocols previously optimized by our group[8], biotinylated proteins were captured by streptavidin magnetic beads (88817; Thermofisher Scientific) in 1.5 mL Eppendorf LoBind tubes using 83 μL beads per 1 mg of protein in a 500 μL RIPA lysis buffer (RLB) (50 mM Tris, 150 mM NaCl, 0.1% SDS, 0.5% sodium deoxycholate, 1% Triton X-100). In brief, the beads were washed twice with 1 ml of RLB and 1 mg of protein were incubated in 500 μl of total RPL. After incubation at 4 °C for 1 h with rotation, beads were serially washed at room temperature (twice with 1 mL RIPA lysis buffer for 8 min, once with 1 mL 1 M KCl for 8 min, once with 1 mL 0.1 M sodium carbonate (Na2CO3) for ~10 s, once with 1 mL 2 M urea in 10 mM Tris-HCl (pH 8.0) for ~10 s, and twice with 1 mL RIPA lysis buffer for 8 min), followed by 1 RIPA lysis buffer wash and 4 final PBS washes. Finally, after placing the tubes on the magnetic rack, PBS was removed completely, then the beads were further diluted in 100 μl of PBS. The beads were mixed and 10% of this biotinylated protein coated beads were used for quality control studies to verify enrichment of biotinylated proteins (including WB and silver stain of proteins eluted from the beads). Elution of biotinylated protein was performed by heating the beads in 30 μL of 2X protein loading buffer (1610737; BioRad) supplemented with 2 mM biotin + 20 mM dithiothreitol (DTT) at 95 °C for 10 min. The remaining 90% of enriched biotinylated proteins sample attached on beads were stored at −20*C for western blot or mass spectrometric analysis of biotinylated protein.

### Western blotting

To confirm protein biotinylation, 10 μg of tissue lysates were resolved on a 4–12% Bris-Tris gel (M00668, GenScript) and transferred onto a nitrocellulose membrane. The membranes were washed once with TBS-T (0.1% tween-20) and then blocked with Start Block (37543, Thermofisher Scientific) and probed with streptavidin-Alexa 680 (S32358; Invitrogen, dilution 1:15000) diluted in Start Block for 1 h at room temperature. After 3 washes with TBS-T (0.1% tween-20), blots were imaged using Odyssey Infrared Imaging System (LI-COR Biosciences). For detection of V5, β-actin, Cox 5a, LC3 and Tubulin, in whole cell lysate or in streptavidin enriched biotinylated protein, an equal amount of each tissue sample was resolved on a Bris-Tris gel and transferred onto a nitrocellulose membrane. After blocking, incubation with primary antibodies was performed overnight at 4 °C. Further, the membranes were washed 3 times with TBS-T (0.1% tween-20), 10 min each and incubated with an appropriate secondary antibody conjugated with horseradish peroxidase-conjugated or another fluorophore. For HRP-conjugated secondary antibody proteins were detected using the enhanced chemiluminescence method (ECL) (1705060; BioRad). The membrane was imaged on either Odyssey Infrared Imaging System (LI-COR Biosciences) or ChemiDoc Imaging System (Bio-Rad). Validation of MS enriched protein were performed with 100 ug equivalent of protein eluted with beads. The quantification of each band was performed by densitometric measurement.

### Protein digestion, MS, protein identification and quantification

On-bead digestion of proteins (including reduction, alkylation followed by enzymatic digestion by Trypsin and Lys-C) from SA-enriched

pulldown samples (1 mg protein used as input) and digestion of bulk brain (input) samples (50 μg protein), were performed as previously described with no protocol alterations[8]. In brief, after removal of PBS from remaining 90% of streptavidin beads (10% used for quality control using western blot and silver stain) were resuspended in 225 uL of 50 mM ammonium bicarbonate (NH4HCO3) buffer. Biotinylated proteins were then reduced with 1 mM DTT and further alkylated with 5 mM iodoacetamide (IAA) in the dark for 30 min each on shaker. Proteins were digested overnight with 0.5 μg of lysyl (Lys-C) endopeptidase (127-06621; Wako) at RT on shaker followed by further overnight digestion with 1 μg trypsin (90058; ThermoFisher Scientific) at RT on shaker. The resulting peptide solutions were acidified to a final concentration of 1% formic acid (FA) and 0.1% trifluoracetic acid (TFA), desalted with a HLB columns (Cat#186003908; Waters). The resulting protein solution was dried in a vacuum centrifuge (SpeedVac Vacuum Concentrator). Detailed methods for this protocol have been previously published[8]. Lyophilized peptides were resuspended followed by liquid chromatography and MS (Q-Exactive Plus, Thermo, data dependent acquisition mode) as per previously published protocols[8]. MS raw data files were searched using Andromeda, integrated into MaxQuant using the mouse Uniprot 2020 database as reference (91,441 target sequences including V5-TurboID). All raw MS data as well as searched MaxQuant data before and after processing to handle missing values, and uploaded to the ProteomeXchange Consortium via the PRIDE repository with the dataset identifier PXD043963[153]. As previously published, methionine oxidation (+15.9949 Da) and protein N-terminal acetylation (+42.0106 Da) were included as variable modifications (up to 5 allowed per peptide), and cysteine was assigned as a fixed carbamidomethyl modification (+57.0215 Da). Only fully tryptic peptides with up to 2 missed cleavages were included in the database search. A precursor mass tolerance of ±20 ppm was applied prior to mass accuracy calibration and ±4.5 ppm after internal MaxQuant calibration. Other search parameters included a maximum peptide mass of 4.6 kDa, minimum peptide length of 6 residues, 0.05 Da tolerance for orbitrap and 0.6 Da tolerance for ion trap MS/MS scans. The false discovery rate (FDR) for peptide spectral matches, proteins, and site decoy fraction were 1%. Other quantification settings were similar to prior CIBOP studies[8]. Quantitation of proteins was performed using summed peptide intensities given by MaxQuant. We used razor plus unique peptides for protein level quantitation. The MaxQuant output data were uploaded into Perseus (Version 1.6.15) and intensity values were log2 transformed, after which data were filtered so that >50% of samples in a given CIBOP group expected to contain biotinylated proteins, were non-missing values. Protein intensities from SA-enriched pulldown samples (expected to have biotinylated proteins by TurboID) were normalized to sum column intensities prior to comparisons across groups. This was done to account for any variability in level of biotinylation as a result of variable Cre-mediated recombination, TurboID expression and/or biotinylation[8].

### Analyses of MS data and bioinformatics analyses

Within each MS study, we compared bulk proteomes to SA-enriched proteomes to confirm that expected proteins (from either PV-INs or Camk2a neurons) were indeed enriched while non-neuronal proteins (e.g. glial proteins) were de-enriched as compared to bulk brain proteomes. We also identified proteins unique to bulk or SA-enriched pulldown samples. Within SA-enriched biotinylated proteins, we restricted our analyses to those proteins that were confidently biotinylated and enriched (based on statistical significance unadj. $P < 0.05$ as well as 2-fold enrichment in biotinylated vs. non-biotinylated samples). This allowed us to exclude proteins that were non-specifically enriched by streptavidin beads. Within biotinylated proteins, group comparisons were performed using a combination of approaches,

including differential abundance analysis, hierarchical clustering analysis (Broad Institute, Morpheus, https://software.broadinstitute.org/morpheus), as well as PCA, (in SPSS Ver 26.0 or R). Differential abundance analyses were performed on log2 transformed and normalized intensity values using two-tailed unpaired T-test for 2 groups assuming equal variance across groups or one-way ANOVA + post-hoc Tukey HSD tests for >2 groups). Unadjusted and FDR-corrected comparisons were performed, although we relied on unadjusted p-values along with effect size (fold-enrichment) to improve stringency of analyses. After curating lists of differentially enriched proteins, gene set enrichment analyses (GSEA) were performed (AltAnalyze Ver 2.1.4.3) using all proteins identified across bulk and pulldown proteomes as the reference (background list). Ontologies included GO, Wikipathways, KEGG, Pathway Commons, as well as prediction of upstream transcriptional and micro RNA regulators (all included in AltAnalyze Ver 2.1.4.3). Ontologies representative of a given group were selected based on enrichment scores (Fisher test $p < 0.05$). We used SynGO to identify the types of known synaptic proteins (in pre- as well as post-synaptic compartments, and different functional classes) identified in CIBOP studies[154]. Protein-protein-interactions between proteins within lists of interest were examined using STRING[154].

We also performed GSVA of proteins identified in bulk as well as PV-IN proteomes from WT and 5xFAD PV-CIBOP mice to complement GSEA[155,156]. GSVA was performed using the R package GSVA (v1.46.0). As previously published, statistical differences in enrichment scores for each ontology comparing two groups, were computed by comparing the true differences in means against a null distribution which was obtained by 1000 random permutations of gene labels. Benjamini & Hochberg false discovery rate adjusted p values < 0.05 were considered significant. The reference gene sets for GSVA were the M5 (Mouse) Ontology Gene Sets from MSigDB (https://www.gsea-msigdb.org/gsea/msigdb/mouse/collections.jsp?targetSpeciesDB=Mouse#M5).

### Luminex immunoassay for signaling Phospho-protein quantification from mouse brain

Multiplexed Luminex immunoassays were used to measure phospho-proteins in the MAPK (Millipore 48-660MAG) and PI3/Akt/mTOR pathways (Millipore 48-612MAG). The PI3/Akt/mTOR panel included pGSK3α (Ser21), pIGF1R (Tyr1135/Tyr1136), pIRS1 (Ser636), pAkt (Ser473), p-mTOR (Ser2448), p70S6K (Thr412), pIR (Tyr1162/Tyr1163), pPTEN (Ser380), pGSK3β (Ser9), pTSC2 (Ser939) and RPS6 (Ser235/Ser236). The MAPK panel detected pATF2 (Thr71), pErk (Thr185/Tyr187), pHSP27 (Ser78), pJNK (Thr183/Tyr185), p-c-Jun (Ser73), pMEK1 (Ser222), pMSK1 (Ser212), p38 (Thr180/Tyr182), p53 (Ser15) and pSTAT1 (Tyr701). We performed adapted Luminex assays as previously described[8] to directly quantify biotinylated proteins in PV-CIBOP samples (from WT and 5xFAD CIBOP animals), whereby the biotinylated hosphor-protein of interest is first immobilized on a bead using capture antibodies, and then their biotinylation status is detected using a streptavidin fluorophore (Streptavidin-PE), to directly quantify biotinylated PV-IN-derived hosphor-proteins from the bulk homogenate. Luminex assays were read on a MAGPIX instrument (Luminex). As per published protocols, we performed linear ranging for every experiment and sample type prior to full assay runs[8]. Using this approach, any signal arising from non-biotinylated (non-CIBOP) control samples is the background/noise level, which was subtracted from signals derived from CIBOP animals. We also additionally normalized these background-subtracted signals based on TurboID protein levels quantified by MS, to account for any unequal biotinylation across samples. Data were analyzed with and without this TurboID normalization, and no meaningful differences were observed between approaches, therefore the TurboID-normalized data were statistically analyzed and presented in the results.

## Analysis of existing mouse brain TMT-MS data

We used a subset of the data from a larger mouse brain TMT-MS study of aging and 5xFAD disease pathology, and a complete description of this mouse TMT-MS study including expression data after batch correction are available online (https://www.synapse.org/#!Synapse:syn27023828); and data relevant to this study are included in the Supplementary data (Supplementary Data 4). Briefly, TMT-MS was performed on whole cortical brain homogenates from 43 WT and 43 5xFAD mice (ages 1.8 mo. To 14.4 months, n = 8, equally balanced based on sex). Standard tissue processing and TMT-MS pipelines were used, as we have previously published[70]. Brain samples were homogenized using a bullet blender with additional sonication in 8 M Urea lysis buffer containing HALT protease and phosphatase inhibitor (78425, ThermoFisher). Proteins were reduced, alkylated and digested (Lysyl endopeptidase and Trypsin), followed by peptide cleanup and TMT (16-plex kit) peptide labeling as per manufacturer's instructions. We included one global internal standard (GIS) per TMT plex batch to facilitate normalization across batches. All samples in a given batch were randomized across six TMT batches, while maintaining nearly-equal representation of age, sex and genotype across all six batches. A complete description of the TMT mass spectrometry study, including methods for sample preparation, mass spectrometry methodology and data processing, are available online (https://www.synapse.org/#!Synapse:syn27023828). Mass spectrometry raw data were processed in Proteome Discover (Ver 2.1) and then searched against Uniprot mouse database (version 2020), and then processed downstream as described for human brain TMT mass spectrometry studies above. Batch effect was adjusted using bootstrap regression which modelled genotype, age, sex and batch, but covariance with batch only was removed[157]. From the 8,535 proteins identified in this mouse brain proteome, we analyzed data related to known markers of distinct classes of mouse neurons and glial subtypes, based on published bulk and single cell RNAseq studies, as well as markers of AD-associated pathology (including hAbeta42 peptide and Apoe).

## Analysis of human brain proteomic data, brain cell type estimates and association with neuropathological and cognitive traits

Single nucleus Allen brain atlas snRNA data was downloaded from https://cells.ucsc.edu/?ds=allen-celltypes+human-cortex+various-cortical-areas&meta=class_label and processed in R to generate a counts per million normalized reference matrix with 47,509 non-excluded cell nuclei assigned to any of 19 cell type clusters[1,72]. The 598 sample Banner+ROSMAP consensus proteome protein profiles of bulk dorsolateral prefrontal cortex (BA-9) from postmortem human donors was the bulk brain data for deconvolution, or ultimately, across-sample, within cell type relative abundance estimation[70]. EnsDeconv was run with some adjustment per: https://randel.github.io/EnsDeconv/reference/get_params.html and https://randel.github.io/EnsDeconv/[158]. Briefly, the 5 marker identification methods used to get the top 50 markers by each method were t, wilcox, combined, "none" (i.e., all genes in the snRNA reference as a profile), and regression. All methods were run for both untransformed and log2-transformed data. CIBERSORT was used as the most efficient deconvolution method with a low profile for RAM use and CPU time, and estimates from 9 of 10 successful combinations of the above marker selection and transformation methods with CIBERSORT estimation. The nine individual marker selection methods produced a redundant total of 350 marker genes, and of these, genes present in all 9 of the lists for each respective cell type were kept as a consensus list of markers. These consensus lists (Supplementary Data 3) were used as input into the GSVA R package implementation of the ssGSEA algorithm[71]. Finally, ssGSEA estimates of within-cell type relative abundances across the 488/598 samples in the published consensus protein network[70] were correlated to the 44 module eigenproteins

(Mes), which are the first principal components of each module in the network, in addition to the ROSMAP cohort specific trait of slope of cognitive decline, a Z score-scaled measure indicating degree of cognitive resilience of an individual compared to the mean age-dependent cognitive decline of the full ROSMAP cohort population[54,55]. Correlation was performed using the WGCNA R package (v1.72-1) function plotEigengeneNetworks. For resilience PWAS enrichment of significance among PV-IN or CAMK2A neuron-enriched protein gene products (Fig. 2F), and for PV-IN 5xFAD DEPs (Fig. 5O), permutation-based enrichment of pooled significance from the PWAS was computed as previously published ()[58], Software for this is available from https://www.github.com/edammer/MAGMA.SPA.

## Other sources of data used for analyses in this manuscript

MicroRNA affinity purification (miRAP) data from studies of PV-IN and Camk2a neurons was downloaded from Supplementary information associated with the original miRAP publication[50] and miRNA species with PV-IN vs. Camk2a neuronal enrichment patterns, were cross-referenced with predicted miRNA regulators in our PV-CIBOP and Camk2a-CIBOP studies.

## Statistics and reproducibility

Specific statistical tests used for individual experiments are detailed in the figure legends. Generally, all continuous variables were analyzed using parametric tests (two-tailed unpaired T-test assuming equal variances when comparing 2 groups, or one-way ANOVA and post-hoc Tukey HSD tests for >2 group comparisons). For proteomic analyses, we used un-adjusted p values + log2FC thresholds to prioritize differentially-abundant proteins. Power calculations were not performed for individual experiments. In the proteomics study related to PV-CIBOP in 5xFAD mice, 1 out of 4 labeled mice was excluded because of unexpected protein loss during technical sample preparation for MS studies.

## Reporting summary

Further information on research design is available in the Nature Portfolio Reporting Summary linked to this article.

## Data availability

The mass spectrometry proteomics data generated by PV-CIBOP studies have been deposited to the ProteomeXchange Consortium via the PRIDE partner repository with the dataset identifier PXD043963, and processed data have been provided as Supplementary Data files as well. Camk2a-CIBOP data can be obtained using dataset identifiers PXD027488 and PXD032161. The 2020 mouse Uniprot database (downloaded from https://www.uniprot.org/help/reference_proteome). Source Data is provided as a Source Data file. Source data are provided with this paper.

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

## Acknowledgements

SR: R01 NS114130, RF1 AG071587, R01 AG075820. MJR: R56AG072473, RF1AG079269, Emory ADRC grant 00100569. NTS: U01AG061357, RF1 AG071587 and R01 AG075820. LBW: AG075820, NSF CAREER 1944053. JVS: F31NS127530, AMG: F31AG076289. Panels in Fig. 1, Fig. 2, Fig. 3, Fig. 4 and Fig. 6 were created with licensed version of Adobe Illustrator or with BioRender.com, as indicated in figure legends.

## Author contributions

P.K., A.M.G., M.J.R. and S.R. conceptualized the experiments. P.K., A.M.G., C.E.G., B.R.T., J.V.S., A.T., R.S.N., E.B.D., S.M., L.C., H.X., D.D., L.B.W., N.T.S., M.J.R. and S.R. performed the experiments and analyzed and interpreted the results. A.N. analyzed mass spectrometry data. P.K., A.M.G., M.J.R. and S.R. wrote the original draft of the manuscript. All authors were reviewed and edited subsequent versions of the manuscript over several iterations. A.M.G., N.T.S., J.V.S., M.J.R. and S.R. acquired funding for the studies. N.T.S., M.J.R. and S.R. provided laboratory resources for the experiments.

## Competing interests

The authors declare no competing interests.

## Additional information

[1]Department of Neurology, Emory University School of Medicine, Atlanta, GA 30322, USA. [2]Center for Neurodegenerative Disease, Emory University
School of Medicine, Atlanta, USA. [3]3 Department of Neurology, Yale University School of Medicine, New Haven, CT 06510, USA. [4]Neuroscience Graduate
Program, Laney Graduate School, Emory University, Atlanta, USA. [5]Georgia W. Woodruff School of Mechanical Engineering, Parker H. Petit Institute for
Bioengineering and Bioscience, and Wallace H. Coulter Department of Biomedical Engineering, Georgia Institute of Technology, Atlanta, GA 30322, USA.
[6]Department of Biochemistry, Emory University, Atlanta, GA 30322, USA. [7]Department of Cell Biology, Emory University School of Medicine, Atlanta,
GA 30322, USA. [8]School of Chemical and Biological Engineering, GeoInsrgia titute of Technology, Atlanta, GA 30322, USA. [9]These authors contributed
equally: Prateek Kumar, Annie M. Goettemoeller. [15]These authors jointly supervised this work: Matthew J. M Rowan, Srikant Rangaraju.
✉e-mail: mjrowan@emory.edu; srikant.rangaraju@yale.edu

