## [Peer Review File · Nature Communications]

Native-state proteomics of Parvalbumin interneurons identifies unique molecular signatures and vulnerabilities to early Alzheimer's pathologyEditorial Note: Parts of this Peer Review File have been redacted as indicated to remove third-party material where no permission to publish could be obtained.

REVIEWER COMMENTS

Reviewer #1 (Remarks to the Author):

This manuscript by Kumar and Goettemoeller uses a very interesting intersectional strategy to express TurboID in PV+ interneurons. Rather than a PV-Cre mouse, they use a viral-based enhancer strategy to target Cre to PV+ interneurons in a Rosa26-TurboID transgenic mouse developed and previously described by the Rangaraju group. With TurboID expressed in this specific class of neurons, they perform a proteomic analysis of these neurons. They get a robust data set that is quite different from transcriptomic data sets, emphasizing the different information each -omic strategy yields. The authors then perform a differential comparison between these PV+ interneurons and Camk2a neurons, the latter using a Cre-expressing mouse. This yields an interesting set of differentially expressed genes between the two cell types. Together, these analyses illustrate the power and utility of the proximity labeling strategy. The authors also use this strategy to compare changes in PV+ interneurons in a mouse model of AD. The authors are to be commended for the large number of animals analyzed across many ages. Finally, the authors examine changes in mitochondrial metabolism and they perform physiology using optogenetic stimulation. These experiments suggest changes in synapse function and mitochondrial function. Overall, I really liked the concept of using this technology to identify proteomics-level changes in PV+ interneurons as a consequence of disease. However, there were some things I did not appreciate or thought that were distractions. I list some of these below. Nevertheless, I think the quality of the results are high and this should be of broad interest both from a technical perspective as well as the data sets generated. The implications of the work remain to be validated and tested experimentally.

Concerns:

1. There was a huge amount of bioinformatic comparison with prior data sets, GO analyses, etc. I personally found these comparisons to be a distraction from the results of their studies and comparisons frequently felt forced and conclusions over-interpreted. For example, on line
2. I did not understand the analyses described in Fig. 3. 'Modules' are not well described and the significance of any of these comparisons were not clear to me. I finished reading from lines 209-253 and asked 'so what?' I don't understand how these comparisons "...resolve AD mechanisms at the level of individual neuronal classes.." I think this is a big overstatement. There are no mechanistic studies performed here, only correlations.
3. Line 374 – pro-resilience and anti-resilience proteins were introduced without any introduction. What do these mean?
4. Line 418: "...likely due to changes in proteins...". I suggest a much softer statement such as "...consistent with changes in proteins...". The statement of likely is too strong. Similarly, line 471 "... is likely to represent..." is too strong in the absence of more direct evidence. These are inferences based on correlation.
5. Line 461 – what does (central dogma) refer to?
6. Line 661 – I don't understand the argument why homeostatic plasticity imposes a higher metabolic demand on neurons.

Reviewer #2 (Remarks to the Author):

This manuscript builds on prior work from the authors to apply cell type-specific biotin labeling (called Cell type-specific In vivo Biotinylation Of Proteins, or CIBOP) coupled with mass spec to the study of an important subtype of neuron – the parvalbumin positive fast-spiking GABAergic inhibitory interneuron -- in cerebral cortex of wild-type vs. FxFAD mice (a mouse model of Alzheimer's disease). These groups previously published the TurboID mouse to drive Cre-inducible expression of a V5-tagged biotin ligase sequence (Rayaprolu et al. 2022. Nat Commun), and also showed alterations in Kv3 channel biophysics in cortical PV-INs (which mediates fast-spiking by these cells) in FxFAD mice. Results of the present work provide a rich dataset of differentially expressed proteins in PV-INs vs. bulk tissue and pyramidal cells, in wild-type mice and in FxFAD mice, revealing PV-IN specific signatures that are hidden via other approaches such as sc/snRNAseq and proteomics of bulk tissue. Of note, the authors find evidence for a compensatory or homeostatic signature in PV-INs in FxFAD mice that might reflect a cell-type specific response to increased metabolic demand in these cells. Overall, this is an ambitious project; the approach is rigorous and the amount of data produced is vast. Novelty is deemed to be high. Results will be of interest to scientists in a range of fields including proteomics, neuroscience, neurology, and aging.

Many of the proteins shown to be differentially expressed by PV-INs show a specific subcellular localization or compartment-specific expression. This important point is mentioned in the Discussion. But, what can CIBOP say about relative expression at soma vs. axon/synapse/dendrite? Is it possible that CIBOP is preferentially labeling proteins from a particular subcellular compartment and hence biasing results towards such proteins? What about proteins with different half-life?

PV-INs are known to express Kv3 channels which regulate the high-frequency discharge pattern of these cells. Prior work using immunohistochemistry has shown high-level protein expression of Kv3.1 and Kv3.2 in neocortical PV-INs, with Kv3.1 at relatively higher levels in superficial layers and Kv3.2 in deeper layers. There is Kv3.3 in neocortical PV-INs, but the levels appear to be much lower at the level of mRNA and protein (Chang Zagha et al 2007 JCN). Hence, it is somewhat curious that the authors find highest levels of Kv3.3 (Kcnc3), followed by Kv3.2 (Kcnc2), and not Kcnc1 (Kv3.1). What is the explanation for this? This is an important issue because the authors reference this finding as support for the veracity of the approach and a kind of gold-standard that the results indeed reflect the PV-IN proteome.

Line 396-397. What AAV serotype is being used? It appears from Figure 6A that the virus is injected directly into the brain, rather than via retro-orbital injection as before. It is known that many cell type-specific enhancers such as E2 do not retain cell type specificity when injected directly into the brain and confirming this to be the case would be important.

Measurements of PPR using optogenetics are confounded by the fact that ChR2 is permeable to calcium. This is clear simply from the fact that data from multiple labs have shown that the PV-IN to pyramidal cell synapse has a PPR of about 0.65, whereas this optogenetic-induced response is much higher. The authors allude to this issue; but, minimally, the authors should discuss this issue more directly and may not want to refer to this response as PPR per se. If there is a lower release probability at PV-IN to pyramidal cell synapses and fewer PV-INs, then wouldn't one expect to see a decreased frequency of spontaneous IPSCs in pyramidal cells (Figure 6I)? Prior work from multiple labs including that of the

author (Rowan) has shown that axonal Kv3 channels regulate action potential width. Might the difference in “PPR” be due to increased rate of failures of synaptic release? If Kv3 channels keep spikes brief and limit spike-evoked calcium influx at the PV-IN terminal, then shouldn't less Kv3 protein in FxFAD mice (as shown by the authors) lead to an increased PPR? Alternatively, might any difference in PPR be attributable to decreased PNN (see Wingert and Worg 2021 Front Synaptic Neurosci for a review and discussion of this issue) or A-beta accumulation in pyramidal cells? Finally, a PPR of 0.9 might suggest that a mixture of PV-INs and other interneurons are labeled via this approach (see previous point re: specificity of E2). The authors do temper their claims somewhat, but the limitations of this approach could be discussed further. Please clarify.

A main issue encountered in the study is determining if observed changes in PV-INs are cause, effect, or epiphenomenon. The idea that the observed patterns of proteomic changes in PV-INs in FxFAD mice might reflect increases in mitochondrial biogenesis to meet increased energy demands of PV-INs to “maintain circuit homeostasis” still seems speculative. Is this because of seizures/epilepsy? Is there evidence that PV-INs require more ATP or fire at higher rates early in AD pathogenesis? The slice physiology presented here shows that sIPSC frequency and amplitude is unchanged.

Minor: on Line 311-312, “chosen capture” should perhaps be “chosen to capture.”

Reviewer #3 (Remarks to the Author):

The author attempted to identify significant protein targets related to AD pathogenesis in parvalbumin interneurons (PV-INs) using cell-type-specific in-vivo biotinylation of proteins (CIBOP) coupled with mass spectrometry. They proposed that PV-IN in early amyloid disease had increased mitochondrial protein and metabolism, synaptic and cytoskeletal disruption, and reduced Akt/mTOR signaling. Furthermore, a presynaptic impairment in PV INs-to-excitatory neurotransmission was discovered. The study is intriguing and significant, and the author showed the important role of PV INs in early AD through several tests and analysis. Here are some queries for the author.

1. The author's rationale for selecting PV-INs for this study appears to be inadequate, despite the author's suggestion that PV-INs are significant to cognitive impairment. There are many other cell types that are relevant for cognitive impairment and AD pathogenesis, and they must explain why they chose PV INs and provide background information on previous investigations of cell type specific proteomes.
2. PV-INs account for less than 10-20% of all neurons in the brain, therefore I'm curious how much impact this cell-type-specific proteome alteration could have during early AD pathology, compared to the universal change caused by amyloid in the majority of neurons that could be observed in the bulk brain proteome.
3. The author compares PV-INs to Camk2a in figure 2F and suggests that PV INs enriched more neurodegeneration-relevant proteins, AD risk genes, and pro-resilience proteins than Camk2a, implying that PV INs are vulnerable or significant during AD pathogenesis. However, it is difficult to assess the significance based on the number of elevated proteins until they demonstrate level change or

malfunction in the context of AD. To demonstrate the significance of these proteins, the author must compare DEP of PV-INs to DEP of Camk2a using the wt and AD models, respectively.

4. The author conducted a bulk proteomic investigation on 5XFAD mice of various ages because human post-mortem brain proteomics analysis has limitations in studying protein changes in aging and illness progression (Figure 3, 4). To justify the extension of human proteome analysis to the mouse model, the author should present a comparison or correlation analysis of proteomics data between the human post mortem study and the late stage of AD animal model.

In Figures 4C, D, and E, it is unclear why there is a disparity in results between PV protein change, PV IN number change, and PNN-positive PV INs by age and genotype, despite the author's attempt to argue that the lower proportion of PNN-positive PV-INs in 5XFAD is a health concern. Furthermore, the number of PV INs appears to be altered by phenotype, particularly in the later stages of disease (fig 3C), despite the author's suggestion that PV INs are influenced in the early stages of amyloid beta pathology.

5. It would be more plausible if the results in figure 5JP DEP list were confirmed or integrated using the postmortem bulk proteome data given in figure 3.

6. In Figures 5 and 6, how does early amyloid disease specifically affect presynaptic dysfunction of the PV-pyramidal synapse? What would the mechanism be?

7. In figure 6, the author introduced hAPP-AV19E as a 5XFAD compatible model for functional studies. However, there is no proof that the hAPP-AV19E model (expressed for 3 weeks) has similar PV INs proteome features to the 3 months 5XFAD. The authors must clarify how these two Alzheimer's disease models are equivalent.

8. The authors propose that the proteome modification of PV INs in early amyloid disease does not extend to functional alteration and that the lack of network disruption is due to an early homeostatic response to keep the circuit functional as resilience. The authors proposed metabolic stress and mitochondrial protein change as reasons for this; however, mitochondrial proteins may be modified by amyloid not only in PV INs but also in other excitatory neurons, microglia, astrocytes, and so on. Furthermore, compensating mechanisms for early circuit disruption caused by amyloid disease may occur in a variety of brain cell types. How could the author argue that PV INs mitochondrial protein change or compensatory process is the fundamental mechanism of resilience in early amyloid disease without first verifying it in other cell types like Camk2a?

9. The findings of this study imply that maintaining PV-IN function is protective in AD and that PV-INs are associated with cognitive resilience and vulnerability in AD (page 20). However, this AD vulnerability of PV INs appears to be ApoE genotype-independent, and I'm curious about the author's explanation for this.

We wish to thank the reviewers for their thorough and helpful reviews. In response to reviewer comments, we have now updated our manuscript with new experiments and additional discussion, all marked in the revised manuscript as highlighted text. We believe the manuscript to be much improved after responding to reviewer comments.

REVIEWER COMMENTS

Reviewer #1 (Remarks to the Author):

This manuscript by Kumar and Goettemoeller uses a very interesting intersectional strategy to express TurboID in PV+ interneurons. Rather than a PV-Cre mouse, they use a viral-based enhancer strategy to target Cre to PV+ interneurons in a Rosa26-TurboID transgenic mouse developed and previously described by the Rangaraju group. With TurboID expressed in this specific class of neurons, they perform a proteomic analysis of these neurons. They get a robust data set that is quite different from transcriptomic data sets, emphasizing the different information each -omic strategy yields. The authors then perform a differential comparison between these PV+ interneurons and Camk2a neurons, the latter using a Cre-expressing mouse. This yields an interesting set of differentially expressed genes between the two cell types. Together, these analyses illustrate the power and utility of the proximity labeling strategy. The authors also use this strategy to compare changes in PV+ interneurons in a mouse model of AD. The authors are to be commended for the large number of animals analyzed across many ages. Finally, the authors examine changes in mitochondrial metabolism and they perform physiology using optogenetic stimulation. These experiments suggest changes in synapse function and mitochondrial function. Overall, I really liked the concept of using this technology to identify proteomics-level changes in PV+ interneurons as a consequence of disease. However, there were some things I did not appreciate or thought that were distractions. I list some of these below. Nevertheless, I think the quality of the results are high and this should be of broad interest both from a technical perspective as well as the data sets generated. The implications of the work remain to be validated and tested experimentally.

Concerns:

1. There was a huge amount of bioinformatic comparison with prior data sets, GO analyses, etc. I personally found these comparisons to be a distraction from the results of their studies and comparisons frequently felt forced and conclusions over-interpreted. For example, on line

Response: We appreciate the reviewer's comment. Text in the results section has been modified to clarify the purpose of each analysis, remove redundancies, limit the reliance on GO analyses, and highlight the main take-away message from these supportive analyses, while being careful not to over-interpret the results. We have also made a concerted effort to emphasize the correlative/indirect nature of these analyses in Figures 2 and 3, and avoided any forced conclusions, but rather, use the human-based and bulk brain-proteomic findings, to provide a foundation for the PV-IN-specific studies in 5xFAD mice. Figure 3 has also been simplified, by moving ancillary correlative analyses to a Supplemental Figure. The modifications have been made in the Results and Discussion sections, highlighted in yellow.

2. I did not understand the analyses described in Fig. 3. 'Modules' are not well described and the significance of any of these comparisons were not clear to me. I finished reading from lines 209-253 and asked, 'so what?' I don't understand how these comparisons "...resolve AD mechanisms at the level of individual neuronal classes.." I think this is a big overstatement. There are no mechanistic studies performed here, only correlations.

Response: As addressed in our response to Comment 1, we have provided more clarity in the section related to Figure 3. We have now defined what modules mean and have provided a higher-level summary of the significance of each comparison, particularly to highlight the “so what” conclusions. The modifications have been made in the Results and Discussion sections, highlighted in yellow.

3. Line 374 – pro-resilience and anti-resilience proteins were introduced without any introduction. What do these mean?

Response: We have tried to better define how lists of pro-resilience and anti-resilience proteins were derived from human brain, both in the results (where these terms are first introduced), as well as in the legend for Figure 2. Please note that these lists were derived from prior proteomic studies of post-mortem brain samples from the ROSMAP study, where each participant had a trajectory of cognitive change from enrolment, till death [1, 2]. Cognitive slope was then estimated, such that an individual who has stable cognitive function over time, has a positive cognitive slope, while an individual who experiences faster cognitive decline, has a more negative cognitive slope. Bulk brain proteins that were positively correlated with a positive cognitive slope were labeled as “pro-resilience” because higher levels of these proteins correlated with cognitive stability/resilience to cognitive decline. Conversely, proteins negatively correlated with cognitive slope were labeled as “anti-resilience” because higher levels of these proteins were associated with more rapid cognitive decline. The lists of proteins related to cognitive resilience, allowed us to better interpret the proteomic differences between PV-IN and Camk2a excitatory neurons. We found that pro-resilience proteins were preferentially enriched in PV-IN proteomes as compared to Camk2a neuronal proteomes. This suggests that protective/pro-resilience factors are enriched more in PV-INs. The human brain analysis in Fig 3 also suggests that estimates of PV-IN abundance (as measured by the PV-IN module, or using in silico estimates of PV-INs using known marker lists), are the strongest correlates of cognitive trajectory. Lastly, the association between PV-IN modules and cognitive resilience remains even after adjusting for co-existent neuropathology in human brain. Collectively, these findings from mouse and human, indicate that PV-INs and their proteins may have key roles in regulating cognitive resilience in the brain.

The lists of pro- and anti-resilience proteins also allowed us to interpret the proteomic changes occurring in PV-IN in early AD pathology (shown in Figure 5). For example, we found that some of the earliest proteomic changes occurring in PV-INs, but not in the bulk brain, including decreased levels of several pro-resilience proteins, including Complexins. While not confirmatory, this allows us to infer that loss of pro-resilience proteins in PV-INs is a feature of early AD pathology and may provide a basis for the selective vulnerability of PV-INs in early AD pathology. To make our line of thought clearer in the manuscript, we have attempted to modify results and discussion sections accordingly and highlighted these changes in Yellow.

4. Line 418: “...likely due to changes in proteins...”. I suggest a much softer statement such as “...consistent with changes in proteins...”. The statement of likely is too strong. Similarly, line 471 “... is likely to represent...” is too strong in the absence of more direct evidence. These are inferences based on correlation.

Response: We have modified these statements, highlighting the correlative nature of these specific analyses. We have similarly edited other parts of the manuscripts to avoid such strong statements that are based on correlation.

5. Line 461 – what does (central dogma) refer to?

Response: MitoCarta3.0 categories mitochondrial proteins into 7 functional categories: (i) mitochondrial central dogma, (ii) protein import, sorting and homeostasis, (iii) oxidative phosphorylation (OXPHOS), (iv) metabolism, (v) small molecule transport, (vi) mitochondrial dynamics and surveillance, and (vii) signaling [3]. Mitochondrial central dogma here refers to proteins involved in maintenance/metabolism of mitochondria-specific DNA, RNA and translational elements. We have modified this sentence for clarity, accordingly.

6. Line 661 – I don't understand the argument why homeostatic plasticity imposes a higher metabolic demand on neurons.

Response: We understand the reviewer's concern and have now modified this language. Please see related point (5 of Reviewer #2's concerns) below for a more detailed response.

Reviewer #2 (Remarks to the Author):

This manuscript builds on prior work from the authors to apply cell type-specific biotin labeling (called Cell type-specific In vivo Biotinylation Of Proteins, or CIBOP) coupled with mass spec to the study of an important subtype of neuron – the parvalbumin positive fast-spiking GABAergic inhibitory interneuron -- in cerebral cortex of wild-type vs. FxFAD mice (a mouse model of Alzheimer's disease). These groups previously published the TurboID mouse to drive Cre-inducible expression of a V5-tagged biotin ligase sequence (Rayaprolu et al. 2022. Nat Commun), and also showed alterations in Kv3 channel biophysics in cortical PV-INs (which mediates fast-spiking by these cells) in FxFAD mice. Results of the present work provide a rich dataset of differentially expressed proteins in PV-INs vs. bulk tissue and pyramidal cells, in wild-type mice and in FxFAD mice, revealing PV-IN specific signatures that are hidden via other approaches such as sc/snRNAseq and proteomics of bulk tissue. Of note, the authors find evidence for a compensatory or homeostatic signature in PV-INs in FxFAD mice that might reflect a cell-type specific response to increased metabolic demand in these cells. Overall, this is an ambitious project; the approach is rigorous and the amount of data produced is vast. Novelty is deemed to be high. Results will be of interest to scientists in a range of fields including proteomics, neuroscience, neurology, and aging.

Concerns:

1. Many of the proteins shown to be differentially expressed by PV-INs show a specific subcellular localization or compartment-specific expression. This important point is mentioned in the Discussion. But, what can CIBOP say about relative expression at soma vs. axon/synapse/dendrite? Is it possible that CIBOP is preferentially labeling proteins from a particular subcellular compartment and hence biasing results towards such proteins? What about proteins with different half-life?

Response: We completely agree with the reviewer's comments. It is true that PV-IN CIBOP, as used in our studies using the TurboID-NES model, will bias the proteome towards the cytosol with relative under-sampling of nuclear, mitochondrial and some intra-luminal/secreted proteins. Evidence for this bias was recently published using in vitro TurboID-NES models [4]. Despite this bias, >60% of all cellular proteins were captured by the TurboID-NES method in two mammalian cell lines. Without selectively directing TurboID to specific subcellular compartments in neurons, this approach cannot distinguish between molecular changes occurring in the soma vs. axon/dendrites/synapse. Whether there is a labeling bias of TurboID-NES towards somatic vs axon/synapse/dendritic compartments, can be generally assessed if we compare the distributions

of our data against known atlases of somatic vs axon/synapse/dendritic proteins. To accomplish this, we turned to a dataset published by Glock et. (PNAS 2021) where Ribo-seq was applied to neurons to identify transcripts preferentially translated in the somatic vs. neuropil compartment (translatomes) [5]. The assumption here is that mRNAs that are actively translated in specific compartments of neurons define the proteome of that sub-cellular compartment. For example, the translatome of the soma should represent soma-enriched proteome, while the translatome of neuropil (axon/synapse/dendrites), should represent the proteome of the non-somatic compartment of neurons. Using this reference dataset, we compared the distribution of proteins based on Ribo-seq-benchmarked localizations (Soma, Neuropil or Other). The summary results of this analysis are shown below, and as a supplemental figure (**Supp Figure S4**). The proportion of neuropil-enriched transcripts (equivalent of non-somatic proteins) were approximately 10-11% in our PV-IN proteomes, bulk brain proteomes and the reference Glock et. al. dataset of all 7,350 measured mRNAs. Somatic proteins were more frequently quantified in both PV-IN and bulk brain protein datasets (60%) (Both MS datasets) as compared to the reference Ribo-seq translatome (37.5%) (a RNAseq dataset). This is most likely representative of differences in proteomic (LFQ-MS) methods used by us, as compared to RNAseq approaches used by Glock et. al. This analysis indicates that TurboID-labeled CIBOP proteomes of PV-INS do not seem to be biased towards somatic or neuropil (axon/dendrite/synapse) compartments.

**Summary Table: Distribution of proteins based on RiboSeq_Neuropil vs. Soma
(Glock et. al. PNAS 2021)**

[REDACTED]

The other question raised by the reviewer is related to a potential labeling bias of TurboID based on protein half-life/turn-over. Since TurboID-NES labels proteins regardless of their turnover, we expect to not have a bias towards short-lived or long-lived proteins. Alternatively, it could also be possible that TurboID labeling occurs preferentially in specific compartments involved in translation or protein degradation, leading to biases related to protein half-life. To address this important question, we cross-referenced the PV-IN proteome and the corresponding bulk brain proteomes from our study, against measured protein half-lives in the brain that were derived using metabolic amino acid labeling in vivo by Fornasiero et. al.[6], a reference dataset we also used for analyses presented in **Figure 6** and **Supp Datasheet 6**). We analyzed 1,243 bulk brain proteins and 880 PV-IN enriched proteins (by CIBOP) with corresponding half-lives from the Fornasiero et. al. dataset. The median half-life of proteins measured in the bulk brain

proteome was 8.7 days while the median half-life of proteins labeled by TurboID in the PV-IN proteomes was 8.4 days (Mann Whitney $p=0.16$, two-tailed T-test $p=0.20$), indicating that TurboID labeling by CIBOP does not preferentially label proteins based on their half-life in brain.

We have included these two analyses as a supplemental Figures (**Supp Fig S3 and Supp Fig S4**) highlighting these results and have appropriately added sentences in the results section as well.

2. PV-INs are known to express Kv3 channels which regulate the high-frequency discharge pattern of these cells. Prior work using immunohistochemistry has shown high-level protein expression of Kv3.1 and Kv3.2 in neocortical PV-INs, with Kv3.1 at relatively higher levels in superficial layers and Kv3.2 in deeper layers. There is Kv3.3 in neocortical PV-INs, but the levels appear to be much lower at the level of mRNA and protein (Chang Zagha et al 2007 JCN). Hence, it is somewhat curious that the authors find highest levels of Kv3.3 (Kcnc3), followed by Kv3.2 (Kcnc2), and not Kv3.1 (Kcnc1). What is the explanation for this? This is an important issue because the authors reference this finding as support for the veracity of the approach and a kind of gold-standard that the results indeed reflect the PV-IN proteome.

Response: We thank the reviewer for bringing up this important point. In the manuscript, we stated that **1)** Kcnc2 and Kcnc3 were among the *most enriched* in the PV proteome *with respect* to background levels, that **2)** Kcnc2 was *only* identified in the PV proteome (but not the bulk), and lastly that **3)** Kcnc2 and Kcnc3 were both highly enriched (>4-fold) with respect to the CaMKII-derived native-state proteome. That these Kv3 proteins were highly enriched in our PV-IN proteomes is in line with the previous literature regarding the requirements for the fast-firing phenotype of this neuron subtype.

However, these analyses did not make an explicit comparison of the expression level for all 3 identified subunits in the native state PV proteomes against one another. Thus, we now compare the relative abundance of these 3 subunits after background subtraction in both the WT and 5xFAD conditions. We found that Kcnc1 (Kv3.1), Kcnc2 (Kv3.2), and Kcnc3 (Kv3.3) protein levels in PV cells were actually quite similar, and also unchanged in 5xFAD mice (despite a downward

trend for Kv3.2, in line with our somatic channel recordings in 5xFAD mice in a previous publication (Olah et al. 2022)). We now show a summary of these results below:

- We now report this comparison in the results section (p. 5 & 12).

Nonetheless, to the reviewer's point, our method cannot discriminate between proteins derived from PV interneurons in specific layers, and thus the PV proteome depicted here represents the cellular average from different layers of the cortex. We think this is an important consideration for Kv3 and all proteins in the dataset and thus:

- We now emphasize these limitations in the discussion/limitations section (p. 24).

While our Kv3 findings were in line with what was expected from a putative PV-IN proteome, this was not our only 'gold-standard'. Other expected PV proteins were highly enriched using the PV-CIBOP method including Gria4, Syt2, and Ank1, while markers of excitatory neurons (Slc17a7), astrocytes (GFAP), microglia (Cst3), oligodendrocytes (Mbp, Plp1) were not enriched. Also, a recently published synaptosomal fractionation proteomics study found significant enrichment of Cplx1 in inhibitory synapses [7]- further validating our findings here, where we found very strong enrichment of Cplx1 in PV interneurons.

Furthermore, we used additional methods to validate our proteomic findings throughout the manuscript, including but not limited to **1**) verified increase of Cox5a levels in 5xFAD PV-INs (but not in the bulk brain) by Western Blot as predicted from PV-CIBOP and **2**) presynaptic dysfunction as predicted by changes in several SNARE-associated proteins from PV-CIBOP.

3. Line 396-397. What AAV serotype is being used? It appears from Figure 6A that the virus is injected directly into the brain, rather than via retro-orbital injection as before. It is known that many cell type-specific enhancers such as E2 do not retain cell type specificity when injected

directly into the brain and confirming this to be the case would be important.

Response: Overall, we certainly appreciate the reviewer's points here- indeed we agree that often, direct injection of AAVs with apparently specific promoters or enhancers can result in off-target expression. However, in our group, we have simply not seen 'off-target' expression using the AAV(PHP.eB).E2.XFP vector when injected locally (with one exception being for Cre expression). Our criteria for 'on-target' expression following stereotactic injection in S1 cortex (where experiments in Figure 6 are performed) is based solely on physiology from fluorescent-guided patch clamp experiments (i.e., fluorescent-targeted recordings). The physiological phenotype of all the recordings we make in S1 cortex following stereotactic injections in this way have a fast-spiking parvalbumin interneuron signature with very rapid (<0.4ms) APs, very fast and non-accommodating firing rates (often ~300Hz) (e.g., see ~40 whole cell recordings here in supp. Fig 1 panels I & J which include standard deviations, also Olah, Rowan et al. eLife, 2022). Furthermore, our optogenetically derived PV synaptic results were indistinguishable when cell-specific expression was regulated either by Cre (in PV-Cre mice) or the E2 enhancer (in WT mice).

However, it is important to note that we have not extensively characterized the targeting-specificity of E2 following stereotaxic injection in other regions of cortex or when packaged into serotypes other than PHP.eB. Lastly, what is also important is that retro-orbital injection of E2 (with the PHP.eB serotype) was found to be specific for PV expressing/fast-spiking interneurons throughout the cortex (this study) using both patch clamp and post-hoc IHC methods. We address these points:

- We now clarify in the methods section and Figure 6 legend that the PHP.eB serotype was also used for stereotactic injections, and now further emphasize our selection of serotype-enhancer combinations (p. 26, 28). We also now added now citations from our group and others in support of the specificity of the enhancer approach to PV interneurons.

4. Measurements of PPR using optogenetics are confounded by the fact that ChR2 is permeable to calcium. This is clear simply from the fact that data from multiple labs have shown that the PV-IN to pyramidal cell synapse has a PPR of about 0.65, whereas this optogenetic-induced response is much higher. The authors allude to this issue; but, minimally, the authors should discuss this issue more directly and may not want to refer to this response as PPR per se. If there is a lower release probability at PV-IN to pyramidal cell synapses and fewer PV-INs, then wouldn't one expect to see a decreased frequency of spontaneous IPSCs in pyramidal cells (Figure 6I)? Prior work from multiple labs including that of the author (Rowan) has shown that axonal Kv3 channels regulate action potential width. Might the difference in "PPR" be due to increased rate of failures of synaptic release? If Kv3 channels keep spikes brief and limit spike-evoked calcium influx at the PV-IN terminal, then shouldn't less Kv3 protein in FxFAD mice (as shown by the authors) lead to an increased PPR? Alternatively, might any difference in PPR be attributable to decreased PNN (see Wingert and Worg 2021 Front Synaptic Neurosci for a review and discussion of this issue) or A-beta accumulation in pyramidal cells? Finally, a PPR of 0.9 might suggest that a mixture of PV-INs and other interneurons are labeled via this approach (see previous point re: specificity of E2). The authors do temper their claims somewhat, but the limitations of this approach could be discussed further. Please clarify.

Response: We thank the reviewer for bringing up these important points- to address these we looked to our data, the prior literature, and performed new experiments:

It is true that many labs have seen a lower PPR (more depressing) at PV-PC synapses than reported in our studies. This is likely due to the fact that we used a 'near-physiological' external Ca^{2+} concentration (1.5mM) in our experiments, whereas others often use 2mM or higher. We know from prior work that P_r , and its proxy PPR, are very sensitive to changes in external Ca^{2+} ; this is also true at PV synapses. Indeed, the presynaptic Ca^{2+} sensor in PV cells is likely not saturated until around 4mM external Ca^{2+} [8] (Figure 5C) and that PPR can shift from near-facilitating to depressing when external Ca^{2+} is changed from 1mM to 2mM externally (Figure 7M [9]). Also see (Nilssen et al. 2018)[10] (Figure 7A) and (Kanigowski et al. 2023)[11] (Figure 7A), wherein facilitating responses were seen using 1.6mM and 2.0mM external Ca^{2+} , respectively. However, to examine this in the context of our optogenetic experiments, we increased the external Ca^{2+} and now found (in agreement with previous studies) a now strongly depressing PPR and much larger IPSC amplitude (1.5mM Ca^{2+} is 58.18 +/- 12.79 pA; whereas 3 mM is 204.3 +/- 86.41).

- We now mention this in the results section (p. 15) and include these new data in Supp. Fig. Figure 11 (previously Supp. Fig. 7).

In our original manuscript, the reviewer is correct- we did not consider whether the rapid light pulses used for optogenetic experiments would result in Ca^{2+} influx directly through the ChR2 pore, and thus, alter the purely AP-evoked IPSC. We are not aware of any literature directly examining if such short widefield LED pulses (~0.5-5ms) result in appreciable presynaptic Ca^{2+} entry, such that presynaptic release would be modified. Thus, to address this point, we examined postsynaptic responses to light stimulation both before and after TTX (to eliminate APs and isolate any direct ChR2 effects) using high external Ca^{2+} . We found no synaptic response to light pulses after TTX was applied, indicating that direct Ca^{2+} entry through the ChR2 channel did not influence our synaptic data. These findings do not mean that absolutely no Ca^{2+} influx occurs presynaptically with this perturbation. Indeed, caution should be applied especially at more facilitating synapses where residual Ca^{2+} is expected to have a more prominent effect (i.e., Synaptotagmin-7 expressing synapses).

- We now mention these findings in the results (p. 15-16) and include these new data in Supp. Fig. 11.

We agree that modification of Kv3 availability (i.e., conductance density) could affect presynaptic Ca^{2+} by modifying the presynaptic AP waveform. However, in our previous work in 5xFAD we did not find a change in Kv3 surface expression (at the soma) but rather, biophysical shifts in its voltage of activation/kinetics which resulted in a faster AP waveform (at the soma). Further proteomic analysis of Kv3 levels in PV-INs (now reported in the results section (p. 12)) and in this document (see above) in WT and 5xFAD mice also agrees with these earlier findings.

While it is tempting to map out changes in the soma to mechanisms at axonal sites, we prefer to exercise caution, due to the fact that Kv3 channels exert highly local control on the AP waveform [12, 13] and because we have not measured the AP presynaptically in this study. Furthermore, in distinct optogenetic approaches using the same light power in WT and 5xFAD mice, we did not observe trends that would suggest AP failures were occurring. Lastly, we did not see a change in PV-IN somatic or PV-IN bouton density at 3 months of age using IHC (reported in the original manuscript; Figure 4E & Figure S5). With this in mind, the fact that we did not see a change in spontaneous IPSCs may be due to several factors, which we admittedly failed to fully explore in the original manuscript. This includes the fact that Sst interneurons contribute to these integrated network measurements. Furthermore, changes observed following evoked release (here with optogenetics) are not necessarily coupled to changes in stochastic, relatively low-frequency events recorded spontaneously in PV interneurons[14].

This is potentially mechanistically salient, as presynaptic molecular ensembles responsible for spontaneous (stochastic) and evoked release may differ (i.e., distinct pools) [15]. In the original manuscript, we reported changes in several presynaptic proteins in 5xFAD (p.17) as a rationale to perform the synaptic physiology. To now explore this further, we returned to our PV-CIBOP proteomic dataset to compare potential changes across all SNARE-associated proteins from WT and 5xFAD mice now more explicitly. Of the SNARE-associated proteins identified in PV-INs, only the Cplx 1,2 proteins were reduced in 5xFAD- and quite dramatically. Interestingly, previous work has found that the loss of Cplx1 and 2 can alter evoked release *without* affecting spontaneous release (analogous to our findings in 5xFAD mice) [16]. Although we did observe a significant reduction in PNN density at 3 months of age, alterations in PNNs *in vivo* are not likely to affect short-term synaptic plasticity [16].

- We now mention these points in context within the results (p. 16) and include these new proteomic analyses in Figure 6.

5. A main issue encountered in the study is determining if observed changes in PV-INs are cause, effect, or epiphenomenon. The idea that the observed patterns of proteomic changes in PV-INs in 5xFAD mice might reflect increases in mitochondrial biogenesis to meet increased energy demands of PV-INs to “maintain circuit homeostasis” still seems speculative. Is this because of seizures/epilepsy? Is there evidence that PV-INs require more ATP or fire at higher rates early in AD pathogenesis? The slice physiology presented here shows that sIPSC frequency and amplitude is unchanged.

Response: This is a fair point. While there is 1) evidence in the literature of associations between increased metabolic stress and AD pathology (which we discuss in the manuscript), and that 2) oxygen consumption/mitochondrial protein expression is associated with PV-interneuron related circuit activity [17], the proteomic changes we see here may not necessarily reflect a causal response to changes in circuit activity *in vivo*. Thus, have now tempered and/or broadened the language throughout the relevant parts of the results and discussion such that the significant alterations in the PV proteome may arise due a combination of cell intrinsic or extrinsic changes in the network, or due to direct interactions of certain proteins (mitochondrial or otherwise) with A β .

6. Minor: on Line 311-312, “chosen capture” should perhaps be “chosen to capture.”

Response: This error has been corrected.

Reviewer #3 (Remarks to the Author):

The author attempted to identify significant protein targets related to AD pathogenesis in parvalbumin interneurons (PV-INs) using cell-type-specific in-vivo biotinylation of proteins (CIBOP) coupled with mass spectrometry. They proposed that PV-IN in early amyloid disease had increased mitochondrial protein and metabolism, synaptic and cytoskeletal disruption, and reduced Akt/mTOR signaling. Furthermore, a presynaptic impairment in PV INs-to-excitatory neurotransmission was discovered. The study is intriguing and significant, and the author showed the important role of PV INs in early AD through several tests and analysis. Here are some queries for the author.

1. The author's rationale for selecting PV-INs for this study appears to be inadequate, despite the author's suggestion that PV-INs are significant to cognitive impairment. There are many other cell types that are relevant for cognitive impairment and AD pathogenesis, and they must explain why they chose PV INs and provide background information on previous investigations of cell type specific proteomes.

Response: We certainly appreciate this perspective. Ultimately, we agree that many distinct cell types will be related to cognitive dysfunction in MCI, and finally, dementia in late-stage AD. In terms of neuron-type-specific dysfunction in early-stage AD, extensive literature indicates that inhibitory interneurons appear to be one of the earliest cell types showing alterations in their intrinsic functionality. Of this extensive literature, PV interneurons are certainly the most prominently studied inhibitory cell studied to date. Indeed, early hyperexcitability in AD pathology models is thought to arise in large part due to a reduction in network inhibition from PV interneurons [18-21]. Furthermore, encouraging recent papers now demonstrate that direct chemogenetic enhancement of PV interneuron activity can improve network hyperexcitability and short-term memory deficits in *in vivo* mouse models of AD pathology [22, 23].

- However, we agree that this literature should be more prominently described and cited in the manuscript introduction, and thus have now done so.

In terms of other *in vivo* cell-type-specific proteomics in the brain, it is our understanding that this manuscript represents one of the first to ever be resolved in this way (our group developed the TurboID mouse model just recently). It is true that the proteomes of other interneuron types, such as SST interneurons, are also likely affected by AD pathology [18]. Indeed, our findings suggest that cell-type-specific proteomic alterations may be a rule, not an exception, in neurodegenerative disease. Thus, we hope others and our groups will employ the integrative techniques developed in the current study to other cell types in the near future. Novel enhancer-AAVs will allow for this in many cell types soon- including in oligodendrocytes, specific excitatory neuron populations, and more.

2. PV-INs account for less than 10-20% of all neurons in the brain, therefore I'm curious how much impact this cell-type-specific proteome alteration could have during early AD pathology, compared to the universal change caused by amyloid in the majority of neurons that could be observed in the bulk brain proteome.

Response: Despite the fact that PV-IN somas are a minority of somas, these cells make extensive, dense local axonal projections which reach essentially every excitatory neuron in the local circuit. Alterations in PV interneuron function have been shown to contribute to circuit dysfunction in early-stage AD pathology models (see point (6) above for details) which is also predicted to accelerate AD pathology further (i.e., circuit hyperexcitability leading to increased

deposition of both amyloid and tau; for example see [24, 25].

3. The author compares PV-INs to Camk2a in figure 2F and suggests that PV INs enriched more neurodegeneration-relevant proteins, AD risk genes, and pro-resilience proteins than Camk2a, implying that PV INs are vulnerable or significant during AD pathogenesis. However, it is difficult to assess the significance based on the number of elevated proteins until they demonstrate level change or malfunction in the context of AD. To demonstrate the significance of these proteins, the author must compare DEP of PV-INs to DEP of Camk2a using the wt and AD models, respectively.

Response: We agree with the reviewer, that direct verification of PV-IN vs. excitatory neuron proteomic differences, patterns of enrichment of AD-risk and resilience proteins, and how each neuronal proteome changes in the context of AD pathology, need to be carefully examined using neuron class-specific approaches. Ideally, the same labeling approach (eg. TurboID), using equivalent Cre-delivery strategies (AAV vs transgenic), and regional precision, should be applied. While Camk2a-Cre-ert2 mice represent a good starting point to probe excitatory neuronal proteomes using CIBOP, we recognize that there are likely to be sub-class-specific AD-related changes within excitatory neurons that need more precise approaches, which can overcome problems related to inheritance heterogeneity within Excitatory neurons, and technical concerns related to Camk2a-CIBOP. We present our arguments below.

[REDACTED]

me(i) Neuronal heterogeneity: The degree of heterogeneity within PV-INs is far less than the diversity among the larger group of Camk2a excitatory neurons. In the Allen Brain atlas of single cell/nuclear RNAseq neuronal transcriptomes from adult mouse brain which identified 387 distinct clusters of cells and sub-types, 240 pan-excitatory (Slc17a7/Camk2a+) and 122 pan-inhibitory (Gad1/2+) neuronal clusters were identified. Among the inhibitory clusters, 15 clusters represented Pvalb+ interneurons (**Figure R1, for reviewer only**). Therefore, we expect the Camk2a-CIBOP approach to average across all excitatory classes, leading to under-appreciation of any distinct changes within various classes of excitatory neurons. Differential vulnerability to AD pathology is likely to exist within the broader group of pan-excitatory neurons, as published recently [26]. While using Camk2a-CIBOP may be an intuitive approach to obtain a cellular contrast to compare/benchmark against PV-IN proteomic changes in AD pathology

models, the better controlled and rigorous experimental design should contrast PV-INs with specific excitatory neuronal classes rather than all Camk2a excitatory neurons grouped into a large heterogenous group. Comparing a relatively homogenous PV-IN proteomic group with a highly heterogenous Camk2a-excitatory group, is therefore likely to be problematic, and will most likely yield high-level of discordance between PV-IN and Camk2a proteomes as we already observed when comparing PV-IN changes with bulk brain changes. This is because PV-INs account for approximately <20% of cortical neurons while Camk2a excitatory neurons account for the vast majority of cortical neurons. In **Figure 4C** of the manuscript, we estimated abundances of pan-excitatory neuronal proteins as one large group (using a collated list of 562 pan-excitatory neuron markers), as well as pan-inhibitory (148 markers) as well as interneuron subclasses (eg. 53 markers for PV-INs). The estimates of these neuronal classes showed distinct patterns of change across age, and genotype. In general, both pan-excitatory and pan-inhibitory proteins showed a general decrease with aging. However, excitatory protein abundances were generally higher while inhibitory proteins were generally lower in 5xFAD brain as compared to WT brain. Interestingly, within inhibitory interneurons, genotype-specific effects were only evidence in PV-INs but not in SST or VIP INs, justifying our focus in PV-INs using CIBOP. This also means that the broader group of Camk2a excitatory neurons needs to be interrogated at the proteomic level with more granularity rather than as one large neuronal group. Recognizing neuronal heterogeneity within excitatory and inhibitory neurons, and the emergent evidence for distinct interneuron class vulnerabilities in AD, we targeted PV-INs rather than all inhibitory neurons (using a broader Gad-Cre approach), and re-emphasize that the focus of this paper is PV-INs.

(ii) Technical concerns with Camk2a-CIBOP for excitatory neuronal proteomics in AD models: Another important technical reason not to use Camk2a-CIBOP for disease-specific changes is that low level Camk2a expression is seen across inhibitory neurons (**Figure R1**) [27]. Therefore, the pan-excitatory neuronal CIBOP proteome will be partly contaminated by less-abundant inhibitory neuronal proteins as well.

(iii) Feasibility of alternative strategy and scope of this paper: In principle, we agree with the reviewer's line of thought, and are performing a fair assessment of excitatory neuronal subclass-specific (L2/3, 4/5, Layer 5/6) and interneuron (PV, SST, VIP INs) proteomic changes in AD models, using several class-specific enhancer AAVs to direct Cre to these neurons in TurboID/WT and TurboID x 5xFAD mice. While the PV-IN AAV approach has been verified, many of the other AAV-Cre approaches need to be validated using electrophysiological and immunohistochemical approaches first, before we can perform CIBOP. These validation studies must happen before extension to disease models as well. This approach, if successful, will also give us more precision with regards to brain regions. These studies are proposed within the scope of a 5-year R01 proposal and the ongoing validation of AAV-Cre strategies will establish the feasibility of executing the proposed experiments in a future grant. As can be seen from the scope of these studies, these are expected to take several years to complete. We sincerely hope that the Reviewer considers our response to this comment.

4. The author conducted a bulk proteomic investigation on 5XFAD mice of various ages because human post-mortem brain proteomics analysis has limitations in studying protein changes in aging and illness progression (Figure 3, 4). To justify the extension of human proteome analysis to the mouse model, the author should present a comparison or correlation analysis of proteomics data between the human post mortem study and the late stage of AD animal model.

Response: To address this comment, we present a specific analysis comparing 5xFAD vs. WT mouse brain proteomic changes at age 10 months, and 14.4 months, with human brain proteomic changes (post-mortem dorsolateral pre-frontal cortex, comparing AD+dementia (AD), asymptomatic AD (AsymAD) and Control [28], using a list of 6,681 proteins quantified in both species (**Supp Fig S8**). As suggested by the reviewer, we correlated effect sizes (log₂ FC) observed in human brain (AD vs. Control, AsymAD vs. Control) with those in mice (10 mo 5xFAD vs. WT, 14 mo 5xFAD vs. WT). Proteome-wide (6,681 proteins), changes occurring in AsymAD (vs. Control) and AD (vs. Control) in humans, moderately and positively correlated with changes occurring in mouse brain, particularly at 14.4 months of age (**Figure S8A**). We then focused our analysis specifically on differentially enriched proteins (DEPs) in AsymAD (n=450) and AD (n=2,577) in humans, which were also identified as DEPs in the mouse 5xFAD vs. WT proteomes (**Figure S8B-E**). Using these DEPs from both humans and mice, we estimated the level of concordance in effect sizes (log₂FC) across species. 109 proteins were DEPs in the AsymAD vs. control (human) comparison and in 5xFAD vs. WT (14 mo) comparisons, and amongst these, 70% of changes were concordant. Similarly, 68.4% of 693 proteins identified as DEPs in both humans (AD vs control) and mice (14 mo 5xFAD vs WT) showed concordant

changes (204 increased in both, 270 decreased in both). Ontologies related to splicing, proteasome, actin/cytoskeleton, innate immunity and MAPK signaling, were increased with AD pathology in both species (**Figure S8D**). In contrast, mitochondrial respiratory and TCA cycle proteins, mitochondrial translation, membrane trafficking and clathrin-mediated endocytic mechanisms were decreased in both species (**Figure S8E**). These concordant changes represent AD mechanisms in humans that are modeled in 5xFAD mice at 14 months of age. Our results are also in agreement with proteomic studies of other A β models, which highlight conserved mechanisms across species [29]. In summary, despite well-known species differences, our analyses suggest that it is reasonable to use amyloid beta (A β) pathology models to understand AD-relevant mechanisms. Furthermore, since changes occurring in late-stages of 5xFAD pathology overlap with changes occurring in post-mortem brain changes in both AsymAD and AD with dementia, we conclude that interrogating very early changes that precede A β pathology in 3-month-old 5xFAD is reasonably-well justified. In lieu of recent evidence for significant pathological and modest cognitive benefits of A β -targeted monoclonal antibodies (eg. lecanemab, aducanumab) in human trials, understanding early effects of human-relevant A β pathology in mouse models becomes even more relevant. These ancillary analyses are presented as a supplemental figure (**Supp Figure S8**) and a brief summary has been included in the Results section.

In Figures 4C, D, and E, it is unclear why there is a disparity in results between PV protein change, PV IN number change, and PNN-positive PV INs by age and genotype, despite the author's attempt to argue that the lower proportion of PNN-positive PV-INs in 5XFAD is a health concern. Furthermore, the number of PV INs appears to be altered by phenotype, particularly in the later stages of disease (fig 3C), despite the author's suggestion that PV INs are influenced in the early stages of amyloid beta pathology.

Response: We thank the reviewer for bringing this up and we present our response below.

Figure 4C shows that pattern of change of 53 PV-IN markers collectively (measured by mass spectrometry), across age and genotype (5xFAD vs. WT) in mouse brain. We show that PV-IN markers increase with age in WT mice although this pattern is blunted in 5xFAD mice. Pvalb itself (Figure 4B), measured by mass spectrometry, also showed a decrease at 3 and 6 months in 5xFAD mice. Since this is bulk proteomic data, this finding could reflect either changes in protein levels themselves *without cell loss*, OR, a change in cell numbers. While neuronal and synaptic molecular changes are seen in 5xFAD mice, overt neuronal loss is not typically apparent until after 9 months of age [30]. Therefore, we suspect that these changes occurring at 3-6 months likely reflect protein-level changes without neuronal loss/death while changes >9 months may reflect a combination of protein-level changes and neurodegeneration.

Figure 4D shows that PV protein (Pvalb) abundance, using IHC, is not altered in 5xFAD brain as compared to WT brain, at both ages 3 or 6 months. Pvalb protein is one marker of PV-INs. Pvalb protein is known to undergo rapid down regulation in response to stress or injury (eg. single dose of LPS [31] [32], suggesting that this protein likely has rapid turnover, and its expression may be tightly coupled with neuronal function/activity. Therefore, a change in Pvalb protein itself should not be interpreted as changes in cell numbers.

The apparent discordance here is that PV-IN proteins measured by mass spectrometry show a decreasing trend in the brain proteome in 5xFAD mice at 3 and 6 months, but the IHC data suggests that Pvalb expression in neurons does not change at 3 or 6 months. The most likely

explanation is that IHC is semi-quantitative for a comparison of Pvalb expression levels, while mass spectrometry is quantitative allowing for direct comparisons. Accordingly, MS is able to quantify small differences in total Pvalb abundance (20% decrease measured by MS), which cannot be resolved by IHC.

The reviewer also brought up Figure 3C as representing decrease in number of PV INs in human post-mortem brain from AD cases. Figure 3C presents module M33 (enriched in PV-IN markers) which decreases in abundance when AD brain is compared to AsymAD and control brains. As these are bulk proteomic analyses, these patterns could indicate cell loss (due to neurodegeneration and cell death), OR, decreased expression/abundance of the protein itself without cell loss. It is not possible to resolve between these two possibilities based on bulk brain proteomics data. Unlike mouse 5xFAD brain where overt neuronal loss is not very apparent, neuronal loss is apparent in late-stage human post-mortem brain. Given the challenges with deducing cell numbers from bulk proteomic data, and the known differences in levels of neuronal loss in late-stage human AD (where tau pathology coincides with neuronal and synaptic loss) as compared to Abeta mouse models (no tau pathology and minimal neuronal loss at 3-6 months), we humbly argue that any conclusions about changes in PV-IN cell numbers are limited.

Our PV-CIBOP studies clearly show that early molecular changes do occur in early stages of Abeta pathology in 5xFAD mice, a stage at which several PV-IN markers decrease in mouse brain, although the number of PV-positive neurons does not change. Together, the interpretation is that early molecular changes in PV-INs do not need to cause neuronal loss. In fact, neuronal loss often occurs decades after onset of Abeta pathology in patients as they progress from asymptomatic stages to onset of cognitive impairment [33].

5. It would be more plausible if the results in figure 5JP DEP list were confirmed or integrated using the postmortem bulk proteome data given in figure 3.

Response: We thank the reviewer for this comment. **Figure 5** represents results from differential abundance analyses of PV-IN-specific proteomic changes occurring at a pre-plaque stage of AD pathology in 5xFAD mice. **Figure 3** on the other hand, represents analyses of bulk post-mortem human brain proteomic data (which is not cell type specific). We performed the detailed analyses of human brain data in **Figure 3** (despite being bulk data) to determine whether there is any evidence to support the idea of neuronal class-specific changes that may be occurring in the human brain. We indeed found that PV-IN protein abundances, using module-based analyses as well as in-silico cell type abundance estimates, are associated with cognitive decline, and decrease as AD pathology progresses. However, these results are supportive, and cannot be used as direct evidence for cell type-specific dysfunction in AD. Since majority of neuronal proteins are shared across sub-classes, only a small proportion of proteins have class-specific-enrichment patterns. Due to this inherent limitation of bulk tissue proteomics, the indirect evidence from human brain data serves as the rationale to use cell type-specific methods such as PV-IN-CIBOP in a relevant AD pathology model.

We also emphasize that given the rarity of PV-INs (<20% of cortical cells) and their proteins in the bulk brain proteome, PV-IN-specific changes will more than likely be diluted or even missed when analyzing bulk omics data. Consistent with this prediction, our analyses of bulk brain data from mouse brain (the same animals from which the PV-IN proteomes were

derived) clearly show that 5xFAD vs. WT changes occurring in the bulk brain proteome minimally overlap with PV-IN specific changes (**Supp Fig S10**).

As requested by the reviewer, we provide an ancillary analysis (**for Reviewers only**) which affirms our prediction: that PV-IN-specific changes in mice are not consistent with changes in bulk brain proteomic changes, neither in mouse brain or in human brain. We correlated effect sizes (Log2FC) in mouse brain (5xFAD vs. WT in PV-IN proteomes, 5xFAD vs. WT in bulk brain mouse proteomes) with those in human bulk brain proteomes (AD vs. control, AsymAD vs. control). These analyses were done first using all 1,724 proteins that were identified across PV-IN, mouse bulk and human brain proteomes (**Table R1**) and then restricted only to proteins that were DEPs in the PV-IN proteome (5xFAD vs. WT) and also identified in mouse bulk and human bulk proteomes (n=213 proteins) (**Table R2**). Using 1,724 proteins (proteome-wide analysis), no significant correlation was observed between PV-IN vs mouse bulk or PV-IN vs human bulk proteomic changes (**Table R1**). Restricted to PV-IN DEPs, again, no significant correlation was observed between PV-IN vs mouse bulk or PV-IN vs human bulk proteomic changes (**Table R2**). Taken together, we conclude that PV-IN proteomic changes occurring in early AD pathology in mouse brain, cannot be captured by bulk proteomic analyses of brain tissue. This is most likely because early molecular changes are unique to PV-IN, which are then diluted at the bulk level.

Table R1.

Log2FC Pearson Correlations: All PV-IN proteins vs Human brain proteins (N=1,724)				
	PV_5xFAD vs WT	Bulk_5xFAD vs WT	Hum_AD vs Con	Hum_AsymAD vs Con
PV_5xFAD vs WT		0.014	-0.006	-0.004
p value		0.555	0.81	0.882
Bulk_5xFAD vs WT	0.014		0.029	0.064
p value	0.555		0.24	0.008
Hum_AD vs Con	-0.006	0.029		0.77
p value	0.81	0.24		<.001
Hum_AsymAD vs Con	-0.004	0.064	0.77	
p value	0.882	0.008	<.001	

Table R2.

Log2FC Pearson Correlations: PV-IN DEPs vs Human brain proteins (N=213)				
	PV_5xFAD vs WT	Bulk_5xFAD vs WT	Hum_AD vs Con	Hum_AsymAD vs Con
PV_5xFAD vs WT		0.06	0.056	-0.008
p value		0.353	0.415	0.908
Bulk_5xFAD vs WT	0.06		0.032	0.029
p value	0.353		0.648	0.675
Hum_AD vs Con	0.056	0.032		0.744
p value	0.415	0.648		<.001
Hum_AsymAD vs Con	-0.008	0.029	0.744	
p value	0.908	0.675	<.001	

6. In Figures 5 and 6, how does early amyloid disease specifically affect presynaptic dysfunction of the PV-pyramidal synapse? What would the mechanism be?

Response: See point (4 of Reviewer #2's concerns) above for more details. Exploring all the molecular factors (i.e., the direct or indirect effectors and the presynaptic targets) underlying

changes in presynaptic function in our AD pathology models is outside the reasonable scope of this study. However, we now have more explicitly looked at the potential mechanisms from the presynaptic proteomics in light of the physiological alterations (see point **(4 of Reviewer #2's concerns)** and in updated Figure 6 with more data on SNARE proteins changes in revised manuscript). We hope these findings inspire several new directions for future work, to explore potential causality between presynaptic molecular vulnerabilities in inhibitory interneurons and downstream AD pathology.

7. In figure 6, the author introduced hAPP-AAV as a 5XFAD compatible model for functional studies. However, there is no proof that the hAPP-AAV model (expressed for 3 weeks) has similar PV INs proteome features to the 3 months 5XFAD. The authors must clarify how these two Alzheimer's disease models are equivalent.

Response: In the original manuscript, we did not argue that these two models were equivalent- indeed they were meant to compare the effects of two distinct yet analogous models of early APP/Abeta pathology on PV synaptic function. Ultimately, AD is highly complex with both shared and unique trajectories in both familial (early-onset) and late-onset models of the disease. In the manuscript we demonstrate that similar synaptic phenotypes can arise using distinct yet analogous models, thereby increasing confidence that the effect is not confined to one model but more generalizable. These two approaches are similar in that they both result in an increase in soluble Abeta but not significant plaque formations at the stage/age of observation (see [20, 34] and Figure 4D). One major difference is the fact that 5xFAD mice express hAPP/Abeta throughout development, whereas hAPP-AAV induces expression at an adult stage. Nonetheless we found the very similar synaptic deficits in the 5xFAD and hAPP-AAV approaches (albeit perhaps not surprisingly a stronger effect size in the 5xFAD) would be useful for readers and for future proteomic and mechanistic studies, which we believe will inform future studies on circuit dysfunction and/or the appearance of AD pathology in early-stage models.

- However, we now include a more descriptive rationale (based on the above description) for looking at synaptic effects of hAPP-AAV in the results.

8. The authors propose that the proteome modification of PV INs in early amyloid disease does not extend to functional alteration and that the lack of network disruption is due to an early homeostatic response to keep the circuit functional as resilience. The authors proposed metabolic stress and mitochondrial protein change as reasons for this; however, mitochondrial proteins may be modified by amyloid not only in PV INs but also in other excitatory neurons, microglia, astrocytes, and so on. Furthermore, compensating mechanisms for early circuit disruption caused by amyloid disease may occur in a variety of brain cell types. How could the author argue that PV INs mitochondrial protein change or compensatory process is the fundamental mechanism of resilience in early amyloid disease without first verifying it in other cell types like Camk2a?

Response: We appreciate this perspective and have modified language around homeostatic responses and causality in general throughout the manuscript.

9. The findings of this study imply that maintaining PV-IN function is protective in AD and that PV-INs are associated with cognitive resilience and vulnerability in AD (page 20). However, this AD vulnerability of PV INs appears to be ApoE genotype-independent, and I'm curious about the author's explanation for this.

Response: We appreciate the Reviewer's concern. From the human post-mortem brain proteomic analysis in Figure 3, it is indeed intriguing that the pyramidal module M1 is associated

with APOE genotype (APOE4 allele status is associated with decreased M1 abundance), while the PV-IN module M33 does not have a relationship with APOE genotype. This could suggest that perhaps the relationship between PV-IN proteins and AD pathology (or vulnerability) is APOE-independent. However, this is contradicted by mouse data, where APOE4 expression impacts gamma oscillatory activity (a surrogate of PV-IN function) [35, 36]. Given the bulk proteomic nature of the human post-mortem study, it is possible that relationships between APOE and PV-IN vulnerability are under-estimated. Furthermore, it is also possible that APOE4 mouse models are at a very immature stage of the disease with respect to humans expressing APOE4, for example, no Abeta or Tau pathology is seen in even very old mice with the APOE4 genotype [35]. Whether APOE genotype impacts PV-IN vulnerability independent of AD pathology, cannot be examined in the human post-mortem cohort because APOE4 genotype and AD status are highly collinear.

Together, the direct experimental assessment of how APOE genotype (using KI mouse models with and without AD pathology) impacts PV-IN physiology and molecular properties using PV-CIBOP and electrophysiology need to be performed, representing very exciting future directions of our work.

REFERENCES USED HERE, NOW INCLUDED AND/OR FURTHER EMPHASIZED IN THE MANUSCRIPT

1. Yu, L., et al., *Cortical Proteins Associated With Cognitive Resilience in Community-Dwelling Older Persons*. JAMA Psychiatry, 2020. **77**(11): p. 1172-1180.
2. Hurst, C., et al., *Integrated Proteomics to Understand the Role of Neuritin (NRN1) as a Mediator of Cognitive Resilience to Alzheimer's Disease*. Mol Cell Proteomics, 2023. **22**(5): p. 100542.
3. Rath, S., et al., *MitoCarta3.0: an updated mitochondrial proteome now with sub-organelle localization and pathway annotations*. Nucleic Acids Res, 2021. **49**(D1): p. D1541-D1547.
4. Sunna, S., et al., *Cellular Proteomic Profiling Using Proximity Labeling by TurboID-NES in Microglial and Neuronal Cell Lines*. Mol Cell Proteomics, 2023. **22**(6): p. 100546.
5. Glock, C., et al., *The translome of neuronal cell bodies, dendrites, and axons*. Proc Natl Acad Sci U S A, 2021. **118**(43).
6. Fornasiero, E.F., et al., *Precisely measured protein lifetimes in the mouse brain reveal differences across tissues and subcellular fractions*. Nat Commun, 2018. **9**(1): p. 4230.
7. van Oostrum, M., et al., *The proteomic landscape of synaptic diversity across brain regions and cell types*. Cell, 2023. **186**(24): p. 5411-5427 e23.
8. Bucurenciu, I., et al., *Nanodomain coupling between Ca²⁺ channels and Ca²⁺ sensors promotes fast and efficient transmitter release at a cortical GABAergic synapse*. Neuron, 2008. **57**(4): p. 536-45.
9. Guan, W., et al., *Eye opening differentially modulates inhibitory synaptic transmission in the developing visual cortex*. Elife, 2017. **6**.
10. Nilssen, E.S., et al., *Inhibitory Connectivity Dominates the Fan Cell Network in Layer II of Lateral Entorhinal Cortex*. J Neurosci, 2018. **38**(45): p. 9712-9727.
11. Kanigowski, D., et al., *Somatostatin-expressing interneurons modulate neocortical network through GABA_B receptors in a synapse-specific manner*. Sci Rep, 2023. **13**(1): p. 8780.
12. Rowan, M.J., et al., *Synapse-Level Determination of Action Potential Duration by K(+) Channel Clustering in Axons*. Neuron, 2016. **91**(2): p. 370-83.

13. Rowan, M.J., E. Tranquil, and J.M. Christie, *Distinct Kv channel subtypes contribute to differences in spike signaling properties in the axon initial segment and presynaptic boutons of cerebellar interneurons*. J Neurosci, 2014. **34**(19): p. 6611-23.
14. Yee, A.X. and L. Chen, *Differential regulation of spontaneous and evoked inhibitory synaptic transmission in somatosensory cortex by retinoic acid*. Synapse, 2016. **70**(11): p. 445-52.
15. Kavalali, E.T., *The mechanisms and functions of spontaneous neurotransmitter release*. Nat Rev Neurosci, 2015. **16**(1): p. 5-16.
16. Uzay, B. and E.T. Kavalali, *Genetic disorders of neurotransmitter release machinery*. Front Synaptic Neurosci, 2023. **15**: p. 1148957.
17. Kann, O., et al., *Gamma oscillations in the hippocampus require high complex I gene expression and strong functional performance of mitochondria*. Brain, 2011. **134**(Pt 2): p. 345-58.
18. Algamal, M., et al., *Reduced excitatory neuron activity and interneuron-type-specific deficits in a mouse model of Alzheimer's disease*. Commun Biol, 2022. **5**(1): p. 1323.
19. Chen, L., et al., *Novel Quantitative Analyses of Spontaneous Synaptic Events in Cortical Pyramidal Cells Reveal Subtle Parvalbumin-Expressing Interneuron Dysfunction in a Knock-In Mouse Model of Alzheimer's Disease*. eNeuro, 2018. **5**(4).
20. Goettemoeller, A.M., et al., *Entorhinal cortex vulnerability to human APP expression promotes hyperexcitability and tau pathology*. Res Sq, 2023.
21. Petrache, A.L., et al., *Aberrant Excitatory-Inhibitory Synaptic Mechanisms in Entorhinal Cortex Microcircuits During the Pathogenesis of Alzheimer's Disease*. Cereb Cortex, 2019. **29**(4): p. 1834-1850.
22. Shu, S., et al., *Prefrontal parvalbumin interneurons deficits mediate early emotional dysfunction in Alzheimer's disease*. Neuropsychopharmacology, 2023. **48**(2): p. 391-401.
23. Terstege, D.J. and J.R. Epp, *Parvalbumin as a sex-specific target in Alzheimer's disease research - A mini-review*. Neurosci Biobehav Rev, 2023. **153**: p. 105370.
24. Rodriguez, G.A., et al., *Chemogenetic attenuation of neuronal activity in the entorhinal cortex reduces Abeta and tau pathology in the hippocampus*. PLoS Biol, 2020. **18**(8): p. e3000851.
25. Wu, J.W., et al., *Neuronal activity enhances tau propagation and tau pathology in vivo*. Nat Neurosci, 2016. **19**(8): p. 1085-92.
26. Leng, K., et al., *Molecular characterization of selectively vulnerable neurons in Alzheimer's disease*. Nat Neurosci, 2021. **24**(2): p. 276-287.
27. Achterberg, K.G., et al., *Temporal and region-specific requirements of alphaCaMKII in spatial and contextual learning*. J Neurosci, 2014. **34**(34): p. 11180-7.
28. Johnson, E.C.B., et al., *Large-scale deep multi-layer analysis of Alzheimer's disease brain reveals strong proteomic disease-related changes not observed at the RNA level*. Nat Neurosci, 2022. **25**(2): p. 213-225.
29. Levites, Y., et al., *A β Amyloid Scaffolds the Accumulation of Matrisome and Additional Proteins in Alzheimer's Disease*. bioRxiv, 2023: p. 2023.11.29.568318.
30. Zhang, L., J. Li, and A. Lin, *Assessment of neurodegeneration and neuronal loss in aged 5XFAD mice*. STAR Protoc, 2021. **2**(4): p. 100915.
31. Rezaei, S., et al., *LPS-Induced Inflammation Reduces GABAergic Interneuron markers and Brain-derived Neurotrophic Factor in Mouse Prefrontal Cortex and Hippocampus*. bioRxiv, 2023: p. 2023.11.15.567229.
32. Mao, M., et al., *The dysfunction of parvalbumin interneurons mediated by microglia contributes to cognitive impairment induced by lipopolysaccharide challenge*. Neurosci Lett, 2021. **762**: p. 136133.
33. Jack, C.R., Jr., et al., *Tracking pathophysiological processes in Alzheimer's disease: an updated hypothetical model of dynamic biomarkers*. Lancet Neurol, 2013. **12**(2): p. 207-16.

34. Mucke, L., et al., *High-level neuronal expression of abeta 1-42 in wild-type human amyloid protein precursor transgenic mice: synaptotoxicity without plaque formation*. J Neurosci, 2000. **20**(11): p. 4050-8.
35. Nuriel, T., et al., *Neuronal hyperactivity due to loss of inhibitory tone in APOE4 mice lacking Alzheimer's disease-like pathology*. Nat Commun, 2017. **8**(1): p. 1464.
36. Palop, J.J. and L. Mucke, *Network abnormalities and interneuron dysfunction in Alzheimer disease*. Nat Rev Neurosci, 2016. **17**(12): p. 777-792.

REVIEWERS' COMMENTS

Reviewer #1 (Remarks to the Author):

This is a revised submission. The authors have addressed nearly all of my concerns. I thought it was a terrific paper before - and still think it is a very important contribution. The challenge is that it is just a tremendous amount of data and a bit hard to take it all in. The authors are to be commended for their careful attention to the reviewers' comments and their efforts to address concerns. I think this is an excellent and important paper. Furthermore, it illustrates how one might go about moving from transcriptomics/translatomics to cell-type specific proteomics.

Reviewer #2 (Remarks to the Author):

This manuscript provides a rich dataset of differentially expressed proteins in parvalbumin-positive GABAergic interneurons (PV-INs) vs. bulk tissue and pyramidal cells, in wild-type mice and in a mouse model of Alzheimer's disease (FxFAD mice), revealing PV-IN specific signatures that are hidden via other approaches such as sc/snRNAseq and proteomics of bulk tissue. Of note, the authors find evidence for a compensatory or homeostatic signature in PV-INs in FxFAD mice that might reflect a cell-type specific response to increased metabolic demand in these cells. Overall, this is an ambitious project; the approach is rigorous and the amount of data produced is vast. Novelty is deemed to be high. Results will be of interest to scientists in a range of fields including proteomics, neuroscience, neurology, and aging.

The authors are now highly responsive to review of the manuscript and have markedly improved the clarity of the message and the experimental rigor of the work. This is a highly detailed and professional response to review and the authors should be commended for this.

The authors have a very logical explanation to the question of somatic vs. axosynaptic protein expression and support this with additional analysis and new data (Supp Figure 4).

The authors have a very clear answer to point 2 of Reviewer #2 and include new data in the Results section to address this.

Reviewer #2 accepts the author's response to point 3.

The authors have a detailed and sophisticated explanation to the question raised in point 4 and the response is deemed acceptable. This explanation is supported by new proteomic analyses and mentioned in the Results section.

The authors have appropriately tempered certain claims made in the previous version of the manuscript.

Overall, in the opinion of Reviewer #2, this is an impressive piece of work.

Reviewer #3 (Remarks to the Author):

The author has made an effort to provide sincere and thorough responses to comments, incorporating additional supple figures based on data analysis and references. Lack of fully satisfying answers to some questions is acceptable, considering experimental assessments for these questions may take several years to complete. I hope that the author continues addressing these questions in the future research, as mentioned by the author.

There is a specific aspect that the author may need to discuss. It seems the author has highlighted a crucial observation regarding the inconsistency between cell type specific proteomic changes and bulk brain proteomic changes, both in mouse and human brains. This raises important questions about the interpretation and integration of these diverse sets of data for meaningful utilization. To address this issue, the author may need to engage in a thoughtful discussion in discussion section.

REVIEWERS' COMMENTS

Reviewer #3 (Remarks to the Author):

The author has made an effort to provide sincere and thorough responses to comments, incorporating additional supplementary figures based on data analysis and references. Lack of fully satisfying answers to some questions is acceptable, considering experimental assessments for these questions may take several years to complete. I hope that the author continues addressing these questions in the future research, as mentioned by the author.

There is a specific aspect that the author may need to discuss. It seems the author has highlighted a crucial observation regarding the inconsistency between cell type specific proteomic changes and bulk brain proteomic changes, both in mouse and human brains. This raises important questions about the interpretation and integration of these diverse sets of data for meaningful utilization. To address this issue, the author may need to engage in a thoughtful discussion in discussion section.

Response: There are indeed discrepancies or discordances between bulk brain proteomics data and cell type-specific proteomics data from PV-INs, from mouse brain. Majority of this disagreement is probably because PV-INs are small fraction of the cells that contribute to the bulk brain proteome, therefore these PV-IN signatures are diluted or under-sampled when the whole brain is analyzed at the bulk level. Another explanation is that the CIBOP approach, using TuboID-mediated biotinylation of the cytosolic proteome, introduces its own biases that under or over-sample specific compartments. The discrepancies between bulk brain proteomic data from mouse vs. human is most likely related to inter-species differences, differences in how aging impacts brain, and also how comorbidities and post-mortem intervals impact the brain proteome. We clarify that cell type-specific proteomics using CIBOP is only feasible in mice, but not in humans, therefore a comparison of cell type-specific proteomes from human vs. mouse is not possible based on our work or existing datasets.

We have added to the discussion section to highlight potential reasons for these discrepancies, and how one must consider these variables when interpreting findings. Minor change to the results section has also been made related to human vs. mouse concordant pathological changes.